# Intense upconverted ultraviolet emission of Er$^{3+}$ through confined energy transfer in Yb$^{3+}$/Er$^{3+}$ co-doped Rb$_3$InCl$_6$

Wen Zhang [1,2,3], Wei Zheng [1,2,3] ✉, Ping Huang [1,2,3] ✉, Dengfeng Yang[1,3], Zhiqing Shao[1,3], Wei Zhang[1], Hao Zhang[1], Zhi Xie[4], Jin Xu[1,2,3] & Xueyuan Chen [1,2,3] ✉

Yb$^{3+}$/Er$^{3+}$ activated upconversion (UC) materials have been widely applied in many advanced technologies owing to their high UC efficiency in the visible region. However, it is challenging to achieve efficient ultraviolet (UV) UC luminescence (UCL) in Yb$^{3+}$/Er$^{3+}$ system, due to the dense energy levels of Er$^{3+}$ that impose deleterious nonradiative relaxation. Herein, we report a strategy to liberate the UV-UCL of Er$^{3+}$ based on the confined energy transfer in Yb$^{3+}$/Er$^{3+}$ co-doped 0D Rb$_3$InCl$_6$ with a low phonon energy and a large interionic distance. This facilitates the population of Er$^{3+}$ at the $^4G_{11/2}$ state, which yields intense upconverted UV emission at 384 nm, with a much higher UV-to-green ratio ($I_{384}/I_{554} = 0.864$) than that of traditional UC materials ( < 0.1). By leveraging the intense upconverted UV emission of Er$^{3+}$, we demonstrate the application of Rb$_3$InCl$_6$:Yb$^{3+}$/Er$^{3+}$ nanocrystals as a NIR-to-UV transducer for NIR-triggered anion exchange of CsPbX$_3$ perovskite nanocrystals with high efficiency and good controllability. These findings offer an approach for the exploration of novel UC materials via energy transfer and crystal lattice engineering towards versatile applications.

Lanthanide (Ln$^{3+}$)-doped upconversion (UC) materials, owing to their ability of converting low-energy near-infrared (NIR) photons into high-energy ultraviolet (UV) and visible photons, have evoked tremendous interest in diverse fields covering from optoelectronics, photovoltaics and photocatalysis to biomedicine[1–4]. Specifically, the UV UC luminescence (UCL) of Ln$^{3+}$ has great prospects in many frontier applications such as UV sterilization, cancer therapy, photochemical reaction, and aerospace, because of the high-energy and spatial resolution of UV light triggered by the NIR laser with good penetration depth, convenient remote controllability, and little photodamage to the targeted samples[5–7]. Generally, UV-UCL is achieved through a four- or five-photon energy transfer (ET) UC (ETU) process in Yb$^{3+}$/Tm$^{3+}$ co-doped system upon 980-nm excitation[8–12]. Compared with Yb$^{3+}$/Tm$^{3+}$, the

Yb$^{3+}$/Er$^{3+}$ couple as a famous UC engine has been demonstrated to be more effective in UCL, typically in the green (≈540 nm) and red (≈650 nm) regions, due to the ladder-like energy levels of Er$^{3+}$ that facilitate efficient ET from Yb$^{3+}$ to Er$^{3+}$ with minimal energy mismatch[13–15]. Specifically, the upconverted UV emission of Yb$^{3+}$/Er$^{3+}$ may offer a higher theoretical efficiency than that of Yb$^{3+}$/Tm$^{3+}$, as it can be generated from $^4G_{11/2}$ (≈380 nm) of Er$^{3+}$ through a three-photon ETU process under 980-nm excitation. However, it is challenging to realize efficient UV-UCL in Yb$^{3+}$/Er$^{3+}$ system, because of the dense energy levels of Er$^{3+}$ that aggravate the nonradiative energy losses through cross relaxation between adjacent Er$^{3+}$, back ET from Er$^{3+}$ to Yb$^{3+}$, and energy migration (EM) among Yb$^{3+}$ to the surface and lattice defects[16,17]. In this context, it is imperative to develop a strategy to

[1]State Key Laboratory of Structural Chemistry and Fujian Key Laboratory of Nanomaterials, Fujian Institute of Research on the Structure of Matter, Chinese Academy of Sciences, Fuzhou, China. [2]Fujian Science & Technology Innovation Laboratory for Optoelectronic Information of China, Fuzhou, China. [3]University of Chinese Academy of Sciences, Beijing, China. [4]College of Mechanical and Electronic Engineering, Fujian Agriculture and Forestry University, Fuzhou, China. ✉e-mail: zhengwei@fjirsm.ac.cn; huangping09@fjirsm.ac.cn; xchen@fjirsm.ac.cn

liberate the upconverted UV emission of Er$^{3+}$, which is not only important for the NIR-to-UV utilization but also fundamentally significant for UC materials innovation based on ET engineering between Yb$^{3+}$/Er$^{3+}$ and other Ln$^{3+}$ emitters.

To unlock the UV-UCL of Er$^{3+}$, the search for new host materials with low phonon energies that can mitigate the nonradiative multiphonon relaxation (MPR) from $^4G_{11/2}$ to $^2H_{9/2}$ of Er$^{3+}$ is of utmost importance[18,19]. In this regard, all-inorganic lead-free metal halides with cutoff phonon energies smaller than 300 cm$^{-1}$ could be ideal candidates for this purpose. By doping with transition-metal, lanthanide, and/or main-group s-electron ions, a plethora of lead-free luminescent metal halides with excellent optical properties have been developed and explored as alternatives to lead halide perovskites for various optoelectronic applications[20–26]. Among these candidates, zero-dimensional (0D) In-based metal halides such as Rb$_3$InCl$_6$ stand out owing to their good structural stability, suitable bandgap and coordination environment for Ln$^{3+}$ dopants, and spatially confined 0D structure that may alleviate the nonradiative energy losses associated with EM[27–29]. Note that the term "0D" herein is used to define the crystal structure with isolated structural units at the molecular level rather than the spherical nanoparticles with small sizes. These features make 0D Rb$_3$InCl$_6$ appealing as a distinctive host material for Ln$^{3+}$ doping to achieve desirable UCL properties.

Herein, we report strong UV-UCL of Er$^{3+}$ at 384 nm in Yb$^{3+}$/Er$^{3+}$ co-doped Rb$_3$InCl$_6$ microcrystals (MCs) and nanocrystals (NCs) under 980-nm excitation. The mechanism for the unusual upconverted UV emission of Er$^{3+}$ in Rb$_3$InCl$_6$ is investigated in detail through the structural and spectroscopic analyses combined with the theoretical calculations as well as the comparison with traditional UC phosphors. By taking advantage of the intense upconverted UV emission of Er$^{3+}$, we demonstrate the application of Rb$_3$InCl$_6$:Yb$^{3+}$/Er$^{3+}$ NCs as a UV generator for NIR-triggered anion exchange of CsPbX$_3$ (X = Cl, Br, and

I) perovskite NCs (PeNCs) in haloalkanes with high efficiency and remote controllability, thus unlocking the potential of Yb$^{3+}$/Er$^{3+}$ couple for NIR-to-UV utilization.

## Results
### Structural characterization of Rb$_3$InCl$_6$:Yb$^{3+}$/Er$^{3+}$

Rb$_3$InCl$_6$ crystallizes with the Rb$_3$YCl$_6$ structure type (space group C2/c), which can be derived from the double perovskite structure by non-cooperative tilting of isolated [InCl$_6$]$^{3-}$ octahedra[30]. Each [InCl$_6$]$^{3-}$ octahedron is surrounded by Rb$^+$ cations, forming a spatially confined 0D structure at the molecular level (Fig. 1a). Yb$^{3+}$ and Er$^{3+}$ dopants are supposed to occupy the octahedral In$^{3+}$ site with a symmetry of $C_{2h}$[29], resulting in bright UCL under 980 nm excitation (Fig. 1b). Yb$^{3+}$ (or Er$^{3+}$) singly-doped and Yb$^{3+}$/Er$^{3+}$ co-doped Rb$_3$InCl$_6$ MCs were synthesized via a solvothermal method by using InCl$_3$, YbCl$_3$, and ErCl$_3$ as the metal precursors and methyl alcohol as the solvent. The doping concentrations of Yb$^{3+}$ and Er$^{3+}$ were controlled by varying the feeding ratios of the metal precursors between In, Yb, and Er, and identified by inductively coupled plasma-atomic emission spectroscopy (ICP-AES), showing the actual doping or alloying concentrations of Yb$^{3+}$ and Er$^{3+}$ ranging from 6.2 and 0.9 mol% to 36.9 and 5.9 mol%, respectively (Supplementary Tables 1 and 2), slightly lower than their feeding concentrations. For convenience, we used the feeding concentrations in the following discussion.

Powder XRD patterns of the MCs displayed intense diffraction peaks that can be well indexed into monoclinic Rb$_3$InCl$_6$ (CCDC No. 2018909) without any discernible impurities, revealing high crystallinity and phase purity of the resulting MCs (Fig. 1c). Moreover, the diffraction peaks of the MCs shifted towards smaller angles with an increase in Yb$^{3+}$ or Er$^{3+}$ concentration (Supplementary Figs. 1 and 2), indicating lattice expansion of the MCs, as confirmed by Rietveld refinement of the XRD patterns (Supplementary Figs. 3 and 4),

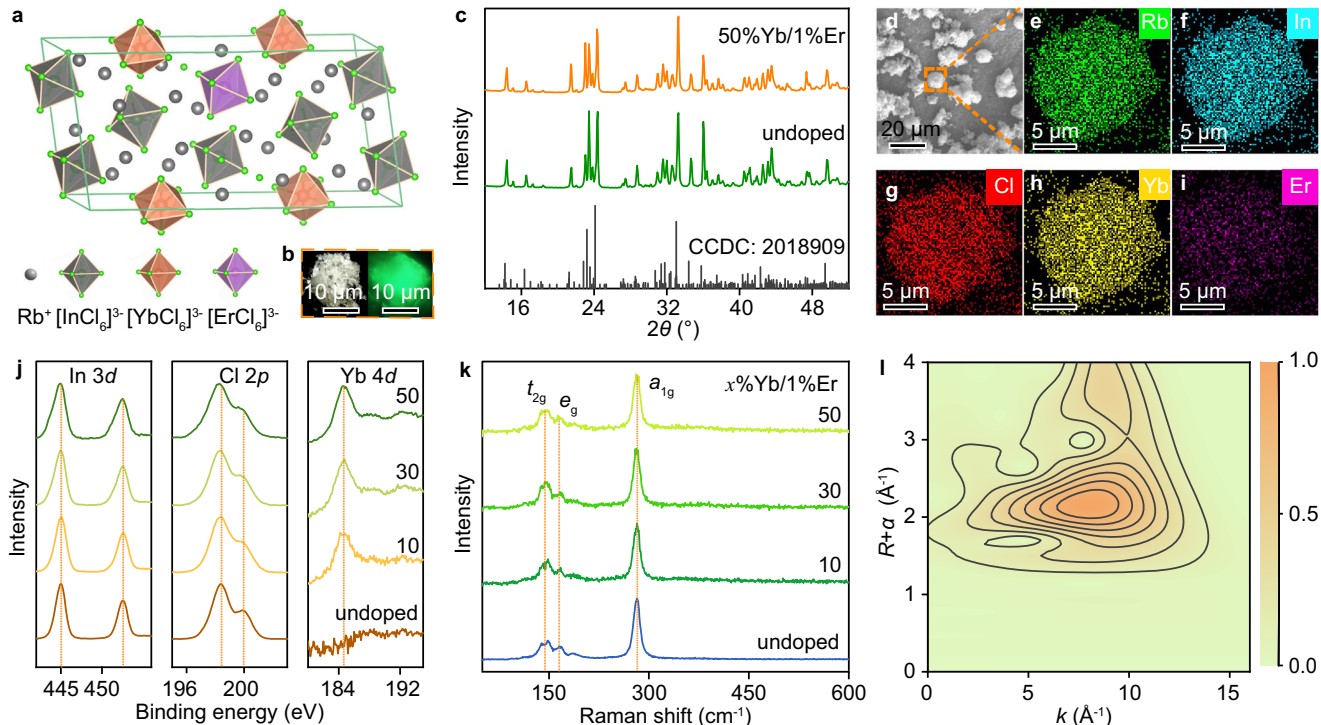

**Fig. 1 | Structural characterization of Rb$_3$InCl$_6$:Yb$^{3+}$/Er$^{3+}$. a** Schematic of the crystal structure of monoclinic Rb$_3$InCl$_6$ and the crystallographic site for Yb$^{3+}$ and Er$^{3+}$ dopants. **b** Photographs of Rb$_3$InCl$_6$: 50%Yb$^{3+}$/1%Er$^{3+}$ MCs under daylight (left) and 980-nm NIR laser irradiation (right). **c** Powder XRD patterns of undoped and 50 mol% Yb$^{3+}$ and 1 mol% Er$^{3+}$ co-doped Rb$_3$InCl$_6$ MCs. The bottom lines represent the standard XRD pattern of monoclinic Rb$_3$InCl$_6$ (CCDC No. 2018909). **d** SEM image and **e–i** EDS elemental mappings (Rb, In, Cl, Yb, and Er) of Rb$_3$InCl$_6$: 50%Yb$^{3+}$/4%Er$^{3+}$ MCs. **j** XPS and **k** Raman spectra of undoped Rb$_3$InCl$_6$ and Rb$_3$InCl$_6$: x%Yb$^{3+}$/1%Er$^{3+}$ MCs with different Yb$^{3+}$ concentrations. **l** Contour plot of the EXAFS in k/R-space of Yb in Rb$_3$InCl$_6$: 50%Yb$^{3+}$/1%Er$^{3+}$ MCs.

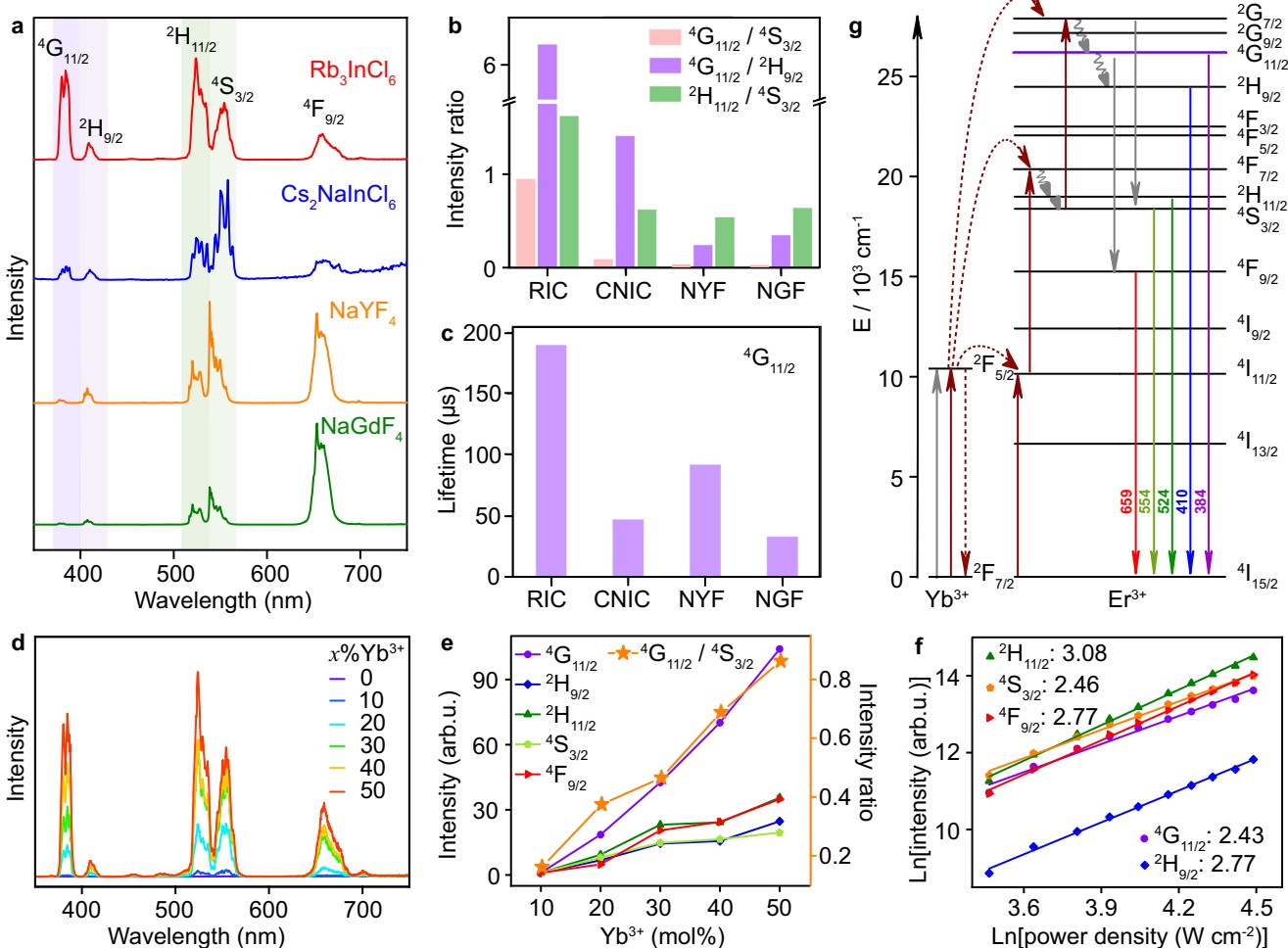

**Fig. 2 | Intense upconverted UV emission of Er³⁺ in 0D Rb₃InCl₆:Yb³⁺/Er³⁺. a** UCL spectra and **b** intensity ratios between the UV and green emissions from ⁴G₁₁/₂ and ⁴S₃/₂ of Er³⁺ (⁴G₁₁/₂/⁴S₃/₂) and between the emissions from the thermally coupled energy levels of ⁴G₁₁/₂/²H₉/₂ and ²H₁₁/₂/⁴S₃/₂ of Er³⁺ in Yb³⁺/Er³⁺ co-doped Rb₃InCl₆ (RIC), Cs₂NaInCl₆ (CNIC), NaYF₄ (NYF), and NaGdF₄ (NGF) MCs under excitation at 980 nm with a power density of 60 W cm⁻². **c** UCL lifetimes of ⁴G₁₁/₂ of Er³⁺ (λ_em: ≈384 nm) in Yb³⁺/Er³⁺ co-doped RIC, CNIC, NYF, and NGF. **d** UCL spectra of Rb₃InCl₆: x%Yb³⁺/1%Er³⁺ MCs with different Yb³⁺ concentrations under 980-nm excitation at a power density of 60 W cm⁻². **e** UCL intensity ratio of ⁴G₁₁/₂/⁴S₃/₂ of Er³⁺ and intensities of the upconverted emissions from ⁴G₁₁/₂ (384 nm), ²H₉/₂ (409 nm), ²H₁₁/₂ (524 nm), ⁴S₃/₂ (554 nm), and ⁴F₉/₂ (659 nm) of Er³⁺ in Rb₃InCl₆: x%Yb³⁺/1%Er³⁺ MCs as a function of the Yb³⁺ concentration. **f** Power dependence of the UCL for the emissions from ⁴G₁₁/₂, ²H₉/₂, ²H₁₁/₂, ⁴S₃/₂, and ⁴F₉/₂ of Er³⁺ in Rb₃InCl₆: 50%Yb³⁺/1%Er³⁺ MCs. **g** Simplified energy-level scheme of Rb₃InCl₆:Yb³⁺/Er³⁺ MCs indicating the major UC processes. The colored solid lines and gray curves represent the radiative and nonradiative transitions, respectively.

whereby the cell volume of the MCs was calculated to increase from 2451.3 Å³ in undoped Rb₃InCl₆ to 2460.5 Å³ in Rb₃InCl₆: 50%Yb³⁺/1%Er³⁺ (Supplementary Tables 3, 4). Such dopant-induced lattice expansion suggests that Yb³⁺ and Er³⁺ have been successfully incorporated into Rb₃InCl₆ lattice and substituted the In³⁺ site bearing the smaller ionic radius ($r_{Yb3+}$ = 0.86 Å, $r_{Er3+}$ = 0.88 Å, $r_{In3+}$ = 0.81 Å, CN = 6). This can be further verified by scanning electron microscopy (SEM), energy-dispersive X-ray spectroscopy (EDS), X-ray photoelectron spectroscopy (XPS), Raman spectroscopy, and extended X-ray absorption fine structure (EXAFS) analyses. SEM images showed that the MCs had an irregular morphology with a broad size distribution in the range of 5–20 μm (Fig. 1d), and EDS elemental mappings revealed the homogeneous distribution of the Yb³⁺ and Er³⁺ dopants in Rb₃InCl₆ lattice (Fig. 1e–i, Supplementary Table 5). XPS spectra exhibited typical Yb 4d and Er 4d signals in Yb³⁺/Er³⁺ co-doped Rb₃InCl₆ MCs, accompanied by a slight shift of the In 3d, Rb 3d and Cl 2p peaks towards lower energies as compared to those observed in undoped Rb₃InCl₆ (Fig. 1j, Supplementary Figs. 5 and 6). Raman spectra of the MCs displayed vibrational peaks at 142, 169, and 282 cm⁻¹, corresponding to the $t_{2g}$, $e_g$, and $a_{1g}$ vibrational modes of the [InCl₆]³⁻ octahedra, respectively (Fig. 1k)[31]. The introduction of Yb³⁺ led to a decrease in intensities and a

broadening in the bandwidth of the Raman peaks, due to the decreased content of [InCl₆]³⁻ octahedra as well as the lattice distortion resulting from the substitution of [InCl₆]³⁻ with [YbCl₆]³⁻ octahedra. The EXAFS analysis showed that the average coordination number for Yb³⁺ is ≈6.2 in Rb₃InCl₆: 50%Yb³⁺/1%Er³⁺ MCs (Fig. 1l, Supplementary Fig. 7, Supplementary Table 6), which is close to the coordination number (CN = 6) of In³⁺ in Rb₃InCl₆ lattice. These observations confirmed that Yb³⁺ substituted the octahedral In³⁺ site in Rb₃InCl₆ lattice.

## UCL properties of Rb₃InCl₆:Yb³⁺/Er³⁺

Figure 2a presents the representative UCL spectrum of Rb₃InCl₆: 50% Yb³⁺/1%Er³⁺ MCs upon NIR excitation with a 980-nm diode laser. The MCs exhibited a set of sharp and intense emission peaks of Er³⁺ at 384 nm (⁴G₁₁/₂ → ⁴I₁₅/₂), 409 nm (²H₉/₂ → ⁴I₁₅/₂), 524 nm (²H₁₁/₂ → ⁴I₁₅/₂), 554 nm (⁴S₃/₂ → ⁴I₁₅/₂), and 659 nm (⁴F₉/₂ → ⁴I₁₅/₂). Strikingly, we found that the UV emission (384 nm) of Er³⁺ was unusually strong and comparable to the green emission (554 nm) under excitation with power densities in a wide range from 0.5 to 150 W cm⁻² (Supplementary Fig. 8). The intensity ratio between the UV and green emissions of Er³⁺ ($I_{384}/I_{554}$) in Rb₃InCl₆: 50%Yb³⁺/1%Er³⁺ MCs was calculated to be 0.864 under 980-nm excitation at a power density of 60 W cm⁻², which is about 1 − 2 orders of

magnitude larger than those obtained in conventional UC phosphors such as NaYF$_4$:Yb$^{3+}$/Er$^{3+}$ (0.025) and NaGdF$_4$:Yb$^{3+}$/Er$^{3+}$ (0.027) and double perovskite Cs$_2$NaInCl$_6$:Yb$^{3+}$/Er$^{3+}$ MCs (0.083) (Fig. 2b, Supplementary Figs. 9–11). The corresponding UCL lifetime of $^4G_{11/2}$ of Er$^{3+}$ (190 µs) in Rb$_3$InCl$_6$:Yb$^{3+}$/Er$^{3+}$ was also much longer than those in NaYF$_4$:Yb$^{3+}$/Er$^{3+}$ (57 µs), NaGdF$_4$:Yb$^{3+}$/Er$^{3+}$ (18 µs), and Cs$_2$NaInCl$_6$:Yb$^{3+}$/Er$^{3+}$ (47 µs) (Fig. 2c, Supplementary Fig. 12), revealing the long-lived $^4G_{11/2}$ state of Er$^{3+}$ in Rb$_3$InCl$_6$:Yb$^{3+}$/Er$^{3+}$, which supports the intense UV emission. It is worth mentioning that, Cs$_2$NaInCl$_6$:Yb$^{3+}$/Er$^{3+}$ MCs exhibited much weaker UCL with a short decay time for $^4G_{11/2}$ of Er$^{3+}$ in comparison with that of Rb$_3$InCl$_6$:Yb$^{3+}$/Er$^{3+}$ MCs, due to the high symmetry of Yb$^{3+}$ and Er$^{3+}$ ($O_h$) and the inefficient ETU processes from Yb$^{3+}$ to Er$^{3+}$ in Cs$_2$NaInCl$_6$:Yb$^{3+}$/Er$^{3+}$[32,33], though they have similarly low phonon energies (Supplementary Fig. 13). Additionally, because of the low phonon energies (<282 cm$^{-1}$) of Rb$_3$InCl$_6$ that alleviate the MPR processes, the intensity ratios between the emissions from the thermally coupled energy levels of $^4G_{11/2}$/$^2H_{9/2}$ ($I_{384/409}$) and $^2H_{11/2}$/$^4S_{3/2}$ ($I_{524/554}$) of Er$^{3+}$ were significantly improved in Rb$_3$InCl$_6$:Yb$^{3+}$/Er$^{3+}$ in comparison with those in the other UC phosphors (Supplementary Table 7). The suppressed MPR of Ln$^{3+}$ in Rb$_3$InCl$_6$ can also be evidenced by the negligibly weak UCL in Rb$_3$InCl$_6$:Yb$^{3+}$/Tm$^{3+}$ and Rb$_3$InCl$_6$:Yb$^{3+}$/Ho$^{3+}$ MCs (Supplementary Fig. 14), wherein the energy gap between $^3H_5$ of Tm$^{3+}$ (or $^5I_6$ of Ho$^{3+}$) and $^2F_{5/2}$ of Yb$^{3+}$ cannot be bridged by the lattice phonons. The intriguing UCL properties of Rb$_3$InCl$_6$:Yb$^{3+}$/Er$^{3+}$ MCs cannot be ascribed to the laser heating effect, as the temperature rise due to laser heating (11.3 °C, 60 W cm$^{-2}$) is not high enough to cause a significant change in the population of Er$^{3+}$ among different energy levels (Supplementary Figs. 15 and 16).

To gain deep insights into the unusual upconverted UV emission ($^4G_{11/2} \rightarrow {}^4I_{15/2}$) of Er$^{3+}$ in Rb$_3$InCl$_6$:Yb$^{3+}$/Er$^{3+}$ MCs, we investigated the effects of the Yb$^{3+}$ concentration on the UCL properties of the MCs. As shown in Fig. 2d, the integrated UCL intensity of the MCs increased steadily with the increasing Yb$^{3+}$ concentration from 10 to 50 mol%, indicating the absence of concentration quenching effect of Yb$^{3+}$ in Rb$_3$InCl$_6$: $x$%Yb$^{3+}$/1%Er$^{3+}$ MCs. Higher doping concentrations of Yb$^{3+}$ beyond 50 mol% led to the RbCl impurity. Concurrently, the intensity ratio of UV-to-green ($I_{384/554}$) of Er$^{3+}$ was remarkably enhanced from 0.182 (10 mol% of Yb$^{3+}$) to 0.864 (50 mol% of Yb$^{3+}$) (Fig. 2e). This observation is in stark contrast to previous findings in other Yb$^{3+}$/Er$^{3+}$ co-doped UC phosphors such as NaYF$_4$:Yb$^{3+}$/Er$^{3+}$, where the intensity ratio of $^4G_{11/2}$/$^4S_{3/2}$ of Er$^{3+}$ underwent a decrease with the increasing Yb$^{3+}$ concentration, due to the enhanced back ET from Er$^{3+}$ to Yb$^{3+}$ (Er$^{3+}$: $^4G_{11/2}$ + Yb$^{3+}$: $^2F_{7/2} \rightarrow$ Er$^{3+}$: $^4F_{9/2}$ + Yb$^{3+}$: $^2F_{5/2}$) at higher Yb$^{3+}$ concentrations[34–36]. Correspondingly, the UCL lifetimes of Er$^{3+}$ exhibited a decrease upon increasing the Yb$^{3+}$ concentration (Supplementary Fig. 17, Supplementary Table 8), as a result of improved ETU processes that accelerated the exhaustion of the excitation energy from Yb$^{3+}$ and consequently the depopulation of Er$^{3+}$ from the emitting levels[37].

Figure 2f shows the double logarithmic plots of the UCL intensities ($I$) of Rb$_3$InCl$_6$: 50%Yb$^{3+}$/1%Er$^{3+}$ MCs versus the excitation power density ($P$), whereby the number of pump photons required to populate each emitting level of Er$^{3+}$ can be derived from the slope ($I \propto P^s$)[38]. By linear fitting to the plots, the slopes ($s$) for the upconverted emissions from $^4G_{11/2}$, $^2H_{9/2}$, $^2H_{11/2}$, $^4S_{3/2}$, and $^4F_{9/2}$ of Er$^{3+}$ were determined to be 2.43, 2.77, 3.08, 2.46, and 2.77, respectively. Similar nonlinear slopes were also obtained in Rb$_3$InCl$_6$: $x$%Yb$^{3+}$/1%Er$^{3+}$ MCs with different Yb$^{3+}$ concentrations (Supplementary Fig. 18). According to Pollnau et al[38]., this may be interpreted as a three-photon process, wherein ETU appears to be the dominant mechanism. Generally, the upconverted green emissions from $^2H_{11/2}$ and $^4S_{3/2}$ of Er$^{3+}$ (524 and 554 nm) in Yb$^{3+}$/Er$^{3+}$ co-doped systems are governed by a two-photon ETU process, characterized by a nonlinear slope $s < 2$[39–41]. Therefore, the observation of nonlinear slopes greater than 2 for the green emissions of Er$^{3+}$ in Rb$_3$InCl$_6$:Yb$^{3+}$/Er$^{3+}$ indicates a significant

contribution of three-photon ETU processes for the population of $^2H_{11/2}$ and $^4S_{3/2}$ of Er$^{3+}$. As schematic illustration in Fig. 2g, the $^4G_{11/2}$ and $^2H_{9/2}$ levels of Er$^{3+}$ are primarily populated through nonradiative relaxation from $^2G_{7/2}$ with a three-photon ETU process, while the $^2H_{11/2}$ and $^4S_{3/2}$ levels can be populated through nonradiative relaxation from $^4F_{7/2}$ with a two-photon ETU process. Concurrently, the excited Er$^{3+}$ at $^2G_{7/2}$ and $^4G_{11/2}$ may also transfer the excitation energy to Yb$^{3+}$, leaving themselves at $^2H_{11/2}$/$^4S_{3/2}$ and $^4F_{9/2}$, respectively[42]. The back ET from Er$^{3+}$ to Yb$^{3+}$ can be evidenced by the decreased intensity ratio of UV-to-green ($I_{384}/I_{554}$) of Er$^{3+}$ and markedly reduced UCL lifetime of $^4G_{11/2}$ with the increasing Er$^{3+}$ concentration in Rb$_3$InCl$_6$: 50%Yb$^{3+}$/$y$%Er$^{3+}$ MCs (Supplementary Figs. 19 and 20, Supplementary Table 9). Such ET cycles between Yb$^{3+}$ and Er$^{3+}$ were also documented by Liu et al. in KYb$_2$F$_7$: Er$^{3+}$ NCs, which led to the improved violet emission from $^2H_{9/2}$ of Er$^{3+}$ at 407 nm[43]. Moreover, owing to the low phonon energies (<282 cm$^{-1}$) of the matrix, the MPR between the levels of Er$^{3+}$ with energy gaps larger than 1400 cm$^{-1}$ is inefficient in Rb$_3$InCl$_6$:Yb$^{3+}$/Er$^{3+}$, as a rule of thumb[44–46]. This property enables the high-order UCL of Rb$_3$InCl$_6$:Yb$^{3+}$/Er$^{3+}$, whereby the four-photon UCL from $^2K_{3/2}$ (307 nm) and $^2P_{3/2}$ (320 nm) of Er$^{3+}$ can be explicitly observed (Supplementary Fig. 21). These results verify that the unusual upconverted UV emission of Er$^{3+}$ in Rb$_3$InCl$_6$:Yb$^{3+}$/Er$^{3+}$ is associated with the low phonon energies of the matrix and the confined ET between Yb$^{3+}$ and Er$^{3+}$ in the peculiar 0D structure of Rb$_3$InCl$_6$.

## Confined ET between Yb$^{3+}$ and Er$^{3+}$ in 0D Rb$_3$InCl$_6$:Yb$^{3+}$/Er$^{3+}$

To shed more light on the ET processes between Yb$^{3+}$ and Er$^{3+}$ in Rb$_3$InCl$_6$:Yb$^{3+}$/Er$^{3+}$ MCs, we calculated the interionic distance in Rb$_3$InCl$_6$: $x$%Yb$^{3+}$/1%Er$^{3+}$ MCs with different Yb$^{3+}$ concentrations based on the density functional theory. As shown in Fig. 3a, the nearest distance between In$^{3+}$ was determined to be 7.39 Å in Rb$_3$InCl$_6$, which is much longer than that between Y$^{3+}$ (3.64 Å) in NaYF$_4$. As a result, the Yb–Yb and Yb–Er distances (>7.14 Å) in Rb$_3$InCl$_6$:Yb$^{3+}$/Er$^{3+}$ are significantly larger than those (<3.50 Å) in NaYF$_4$:Yb$^{3+}$/Er$^{3+}$ when Yb$^{3+}$ and Er$^{3+}$ substitute the In$^{3+}$ or Y$^{3+}$ sites in these two matrices (Supplementary Fig. 22)[36,47]. It is well known that the probability (or rate) of ET between Ln$^{3+}$ ions depends strongly on their distance, and a short interionic distance favors the successive ET, namely, EM among identical ions (e.g., Yb$^{3+}$), while an increase in the interionic distance will slow down the EM rate[48,49]. Regarding the large Yb–Yb and Yb–Er distances in Rb$_3$InCl$_6$:Yb$^{3+}$/Er$^{3+}$, we speculate that the long-range EM process among Yb$^{3+}$ is suppressed, which accounts for the absence of concentration quenching effect of Yb$^{3+}$ in this system. Instead, the excitation energy is deduced to be confined in the [YbCl$_6$]$^{3-}$ and [ErCl$_6$]$^{3-}$ octahedra within a short range, resulting in the ET cycles between Yb$^{3+}$ and Er$^{3+}$ (Fig. 3b).

The confined ET and suppressed EM processes in 0D Rb$_3$InCl$_6$:Yb$^{3+}$/Er$^{3+}$ can be further validated by measuring the photoluminescence (PL) decay curves of Rb$_3$InCl$_6$: $x$%Yb$^{3+}$ MCs with different Yb$^{3+}$ concentrations, whereby the rate of EM among Yb$^{3+}$ can be derived on the basis of the energy diffusion model. According to G. Blasse et al[50]., there are three modes of ET between Ln$^{3+}$ ions when the back ET is negligible, namely, direct ET without diffusion, diffusion-limited, and fast-diffusion modes, which can be distinguished from the shapes of the decay curves of their excited states (Supplementary Fig. 23). In the case of direct ET without diffusion ($P_{DD} = 0$), the decay curve is characterized by an initially non-exponential portion which reflects the transfer to acceptors located at various distances from the donors, followed by an exponential portion with a decay rate equal to the radiative rate. In the case of fast diffusion, the rate of ET among donors (namely, EM) is much larger than that of ET to acceptors ($P_{DD} > P_{DA}$), and the decay curve is exponential because the rapid migration has the effect of averaging the environments of the donors. In the case of diffusion-limited migration, the rate of ET among donors is lower than that from donors to acceptors ($P_{DD} < P_{DA}$). In this situation, the decay curve of the donors is characterized by an initially fast

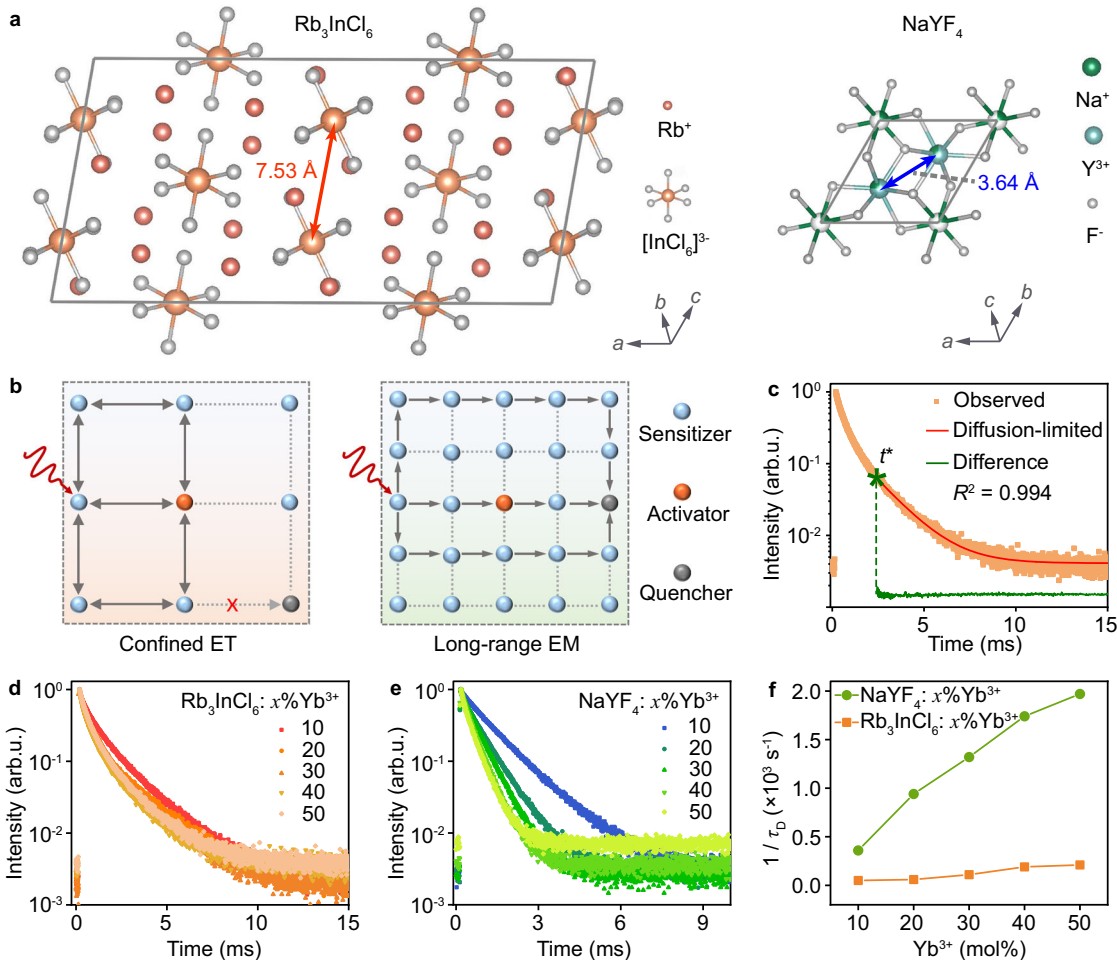

**Fig. 3 | Confined ET between Yb³⁺ and Er³⁺ in 0D Rb₃InCl₆:Yb³⁺/Er³⁺.** **a** Crystal structures of monoclinic Rb₃InCl₆ and hexagonal NaYF₄ viewed along the *c* and *b* axes, respectively, showing the shortest In−In and Y−Y distances. **b** Schematic of the confined ET and long-range EM among the sensitizers and activators with long and short interionic distances, respectively. **c** PL decay curve of Yb³⁺ ($\lambda_{em}$ = 994 nm) in Rb₃InCl₆: 50%Yb³⁺ MCs, which can be fitted well with the diffusion-limited mode. **d** PL decay curves of Yb³⁺ in d Rb₃InCl₆: *x*%Yb³⁺ and **e** NaYF₄: *x*%Yb³⁺ MCs with different Yb³⁺ concentrations by monitoring the Yb³⁺ emission at 994 nm. **f** EM rates among Yb³⁺ in Rb₃InCl₆: *x*%Yb³⁺ and NaYF₄: *x*%Yb³⁺ MCs with different Yb³⁺ concentrations.

non-exponential portion followed by a slow exponential portion with the decay time ($\tau$) shorter than the radiative decay time ($\tau_0$) due to the influence of EM. The boundary between the non-exponential and exponential decay portions can be marked by a characteristic time *t**. Based on these definitions, it can be deduced that the decay of ²F₅/₂ of Yb³⁺ in Rb₃InCl₆: *x*%Yb³⁺ MCs obeyed the diffusion-limited migration mode (Fig. 3c). According to M. Yokota and O. Tanimoto[51], the decay rate due to migration ($1/\tau_D$) can thus be obtained by the following expression:

$$1/\tau = 1/\tau_D + 1/\tau_0 \quad (1)$$

where $1/\tau$ is the observed decay rate derived from the exponential portion of the decay curve and $1/\tau_0$ is the radiative decay rate obtained from the low-doping (1 mol% Yb³⁺) sample. The results showed that the observed decay time ($\tau$) of ²F₅/₂ of Yb³⁺ in Rb₃InCl₆: *x*%Yb³⁺ MCs exhibited only a slight decline from 1.84 to 1.43 ms as the Yb³⁺ concentration increased from 10 to 50 mol% (Fig. 3d, Supplementary Fig. 24), due to the long Yb−Yb distance that mitigated the nonradiative relaxation of Yb³⁺ via EM. This is in marked contrast to the observation in NaYF₄: *x*%Yb³⁺ MCs, wherein the decay time ($\tau$) of Yb³⁺ reduced drastically from 1.12 ms (10 mol%) to 0.37 ms (50 mol%) (Fig. 3e, Supplementary Fig. 25). Specifically, the EM rate ($1/\tau_D$) of Yb³⁺ in Rb₃InCl₆:Yb³⁺ MCs ($2.10 \times 10^2$ s⁻¹ for 50%Yb³⁺) was determined to be

about one order of magnitude slower than that in NaYF₄:Yb³⁺ MCs ($2.17 \times 10^3$ s⁻¹ for 50%Yb³⁺) (Fig. 3f and Supplementary Table 10), confirming the inhibition of EM in Rb₃InCl₆:Yb³⁺ MCs. Therefore, after co-doping with Er³⁺ ions, ET cycles occurred mainly between Yb³⁺ and Er³⁺ because Er³⁺ ions are prone to locate in close proximity to Yb³⁺ in Rb₃InCl₆:Yb³⁺/Er³⁺ (Supplementary Fig. 26), which led to the unusual upconverted UV emission of Er³⁺ in this system. It is worthy of mentioning that the "shell model" developed by F. Rabouw and A. Meijerink et al[52]. instead of the Yokota-Tanimoto model could be more appropriate for elucidating the dynamics of ET from Yb³⁺ to Er³⁺, considering the complex interplay between the donors and acceptors. Benefiting from the confined ET, we achieved a quantum yield (QY) of 0.12% for the upconverted UV emission (384 nm) of Er³⁺ in Rb₃InCl₆: 50%Yb³⁺/1%Er³⁺ MCs upon 980-nm excitation at a power density of 60 W cm⁻², which is much higher than that of the benchmark NaYF₄: 18%Yb³⁺/2%Er³⁺ MCs (0.04%), although the overall UCQY of Rb₃InCl₆:Yb³⁺/Er³⁺ (0.42%) is lower than that of NaYF₄:Yb³⁺/Er³⁺ (1.38%).

**NIR-triggered anion exchange of CsPbX₃ PeNCs**
Furthermore, we synthesized Rb₃InCl₆:Yb³⁺/Er³⁺ NCs with different Yb³⁺ and Er³⁺ concentrations to investigate the effect of particle size on the upconverted UV emission of Er³⁺. Structural and compositional analyses through XRD, transmission electron microscopy (TEM), EDS, ICP-AES, XPS, and Raman spectra confirmed the pure phase and high crystallinity

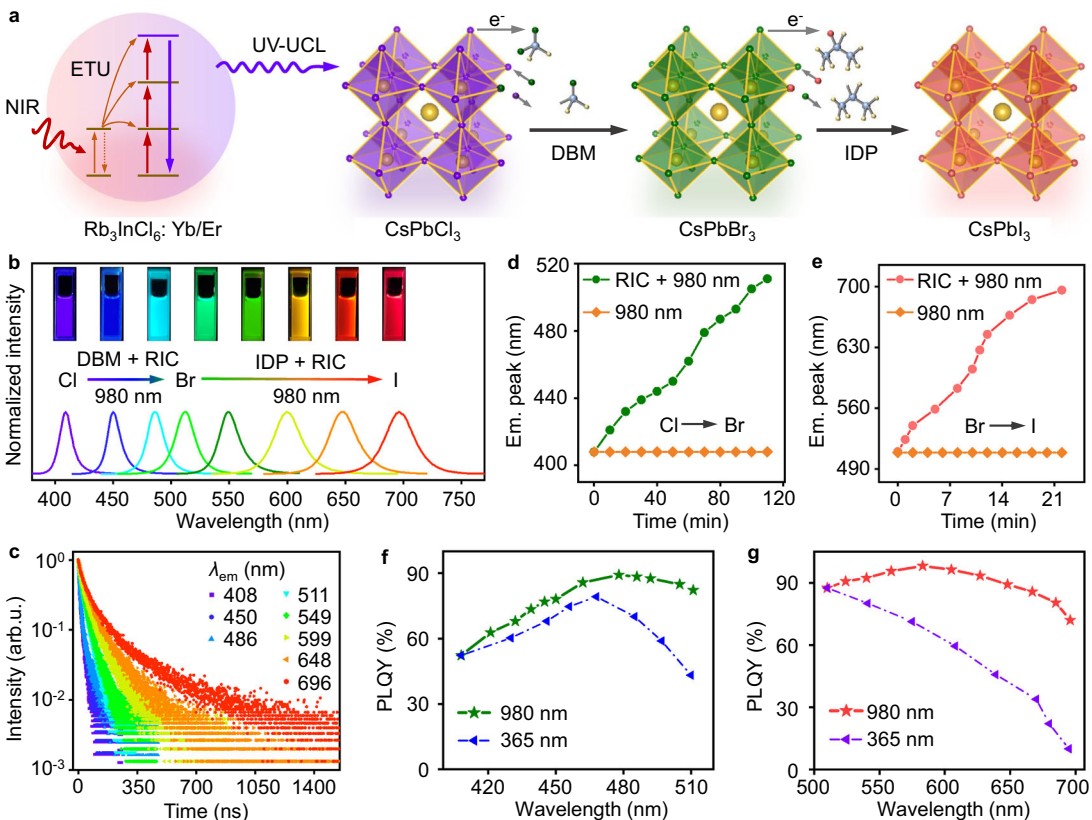

**Fig. 4 | NIR-triggered anion exchange of CsPbX₃ PeNCs using Rb₃InCl₆:Yb³⁺/Er³⁺.** **a** Schematic of NIR-triggered anion exchange of CsPbX₃ PeNCs in haloalkanes by using Rb₃InCl₆:Yb³⁺/Er³⁺ NCs as the NIR-to-UV transducer. **b** PL emission spectra ($\lambda_{ex}$ = 365 nm) and **c** PL decay curves of CsPbX₃ PeNCs with different halide compositions, derived from anion exchange based on the mixtures of RIC/DBM/CsPbCl₃ and RIC/IDP/CsPbBr₃ upon 980-nm NIR laser exposure. The insets in (**b**) show the PL photographs of the resulting PeNCs under 365-nm irradiation. Time dependence of the PL emission peaks ($\lambda_{ex}$ = 365 nm) for **d** CsPbCl₃ → CsPbBr₃ in DBM/CsPbCl₃ and RIC/DBM/CsPbCl₃ and **e** CsPbBr₃ → CsPbI₃ in IDP/CsPbBr₃ and RIC/IDP/CsPbBr₃ upon 980-nm irradiation at a power density of 60 W cm⁻². PLQYs for PeNCs derived from the photoinduced anion exchange **f** from CsPbCl₃ to CsPbBr₃ in DBM/CsPbCl₃ ($\lambda_{ex}$ = 365 nm) and RIC/DBM/CsPbCl₃ ($\lambda_{ex}$ = 980 nm) and **g** from CsPbBr₃ to CsPbI₃ in IDP/CsPbBr₃ ($\lambda_{ex}$ = 365 nm) and RIC/IDP/CsPbBr₃ ($\lambda_{ex}$ = 980 nm), respectively.

of the resulting Rb₃InCl₆:Yb³⁺/Er³⁺ NCs with particle sizes in the range of 48.8–52.8 nm (Supplementary Figs. 27–29, Supplementary Tables 11, 12). Similar to their MC counterparts, Rb₃InCl₆:Yb³⁺/Er³⁺ NCs displayed strong UV emission of Er³⁺ at 384 nm ($^4G_{11/2} \rightarrow {}^4I_{15/2}$), with the UV-to-green ratio ($I_{384}/I_{554}$) of up to 0.702 for Rb₃InCl₆: 50%Yb³⁺/1%Er³⁺ NCs, upon 980-nm excitation at a power density of 60 W cm⁻² (Supplementary Fig. 30). The corresponding UC QYs for the UV emission (≈384 nm) and overall emissions (360–720 nm) of Er³⁺ in Rb₃InCl₆: 50%Yb³⁺/1%Er³⁺ NCs were determined to be 0.08% and 0.37%, respectively, slightly lower than those of their MC counterparts (0.12% and 0.42%). Concentration-dependent UCL measurements showed the increased UCL intensity and $I_{384}/I_{554}$ ratio with the increasing Yb³⁺ concentration from 10 to 50 mol%, concurrent with the decreased UCL lifetimes (Supplementary Fig. 31, Supplementary Table 13), confirming the absence of concentration quenching effect of Yb³⁺ in Rb₃InCl₆:Yb³⁺/Er³⁺ NCs. Further increasing the Er³⁺ concentration on the basis of Rb₃InCl₆: 50%Yb³⁺/1%Er³⁺ NCs shortened the interionic distance between Yb³⁺ and Er³⁺, resulting in the enhanced back ET from Er³⁺ to Yb³⁺ and the decreased $I_{384}/I_{554}$ ratio of Er³⁺ with reduced UCL lifetime of $^4G_{11/2}$ (Supplementary Figs. 32 and 33, Supplementary Table 14). These results are generally consistent with those observed in Rb₃InCl₆:Yb³⁺/Er³⁺ MCs, unraveling that the intense upconverted UV emission of Er³⁺ is the intrinsic properties of Rb₃InCl₆:Yb³⁺/Er³⁺ and independent of the particle size. Our findings may pave the way for exploring efficient UV-UCL from the Yb³⁺/Er³⁺ couple based on the host materials with low photon energies and large interionic distances, which favor the multiphoton UC processes with a long lifetime of $^4G_{11/2}$ of Er³⁺.

Considering the fact that the intense upconverted UV emission of Er³⁺ at 384 nm matches well with the absorption of CsPbX₃ (X = Cl, Br, and I) PeNCs, we explored Rb₃InCl₆:Yb³⁺/Er³⁺ as a NIR-to-UV transducer for photoinduced anion exchange of CsPbX₃ PeNCs in haloalkanes. CsPbX₃ PeNCs are emerging as a new generation of semiconductor materials for various optoelectronic and photovoltaic applications owing their outstanding optical properties[53–55]. Upon 980-nm excitation, the upconverted UV emission of Er³⁺ from Rb₃InCl₆:Yb³⁺/Er³⁺ can be effectively absorbed by CsPbX₃ to trigger the interfacial electron transfer from PeNCs to haloalkanes, resulting in the breakage of the covalent C–X bonds of haloalkanes with released halide ions for anion exchange with PeNCs (Fig. 4a)[56–58]. For this purpose, dibromomethane (DBM) and 2-iodopropane (IDP) were selected as the Br⁻ and I⁻ sources and mixed with cyclohexane (1:1 in volume), respectively. Then, CsPbCl₃ and CsPbBr₃ PeNCs were dispersed in the solvent of DBM and IDP, respectively, along with Rb₃InCl₆:Yb³⁺/Er³⁺ (RIC) NCs at the molar ratio of ≈1:1. Additionally, a trace amount of trioctylphosphine (≈20 μL) was added to improve the PL efficiency of the resulting CsPbX₃ PeNCs[59,60]. The solution was then subjected to photoirradiation with a 980-nm diode laser, and the reaction was tracked by extracting aliquots of the reaction mixture at different time intervals, followed by the measurement of the optical absorption and PL spectra.

As expected, the mixtures of RIC/DBM/CsPbCl₃ and RIC/IDP/CsPbBr₃ underwent an obvious PL color change, respectively, from violet to blue and green and from green to yellow and red with the increasing irradiation time upon 980-nm exposure (Fig. 4b). For comparison, the emission color of the mixtures remained unchanged

in the dark. The corresponding optical absorption and emission spectra of the PeNCs after NIR irradiation showed tunable bandgap (1.8–3.0 eV) and PL emission covering the entire visible spectral region from 408 to 696 nm (Supplementary Fig. 34), indicating the successful NIR-triggered halide exchange of $CsPbX_3$ in haloalkanes. PL decay curves showed an increased PL lifetime from 9.8 to 84.0 ns, as the halide composition of PeNCs evolved from $Cl^-$ to $I^-$ with the increasing irradiation time (Fig. 4c), which is consistent with our previous findings[60]. Structural and morphological characterizations through XRD and TEM showed the cubic phase of PeNCs with mean sizes in the range of 18.0–21.9 nm (Supplementary Figs. 35 and 36), unveiling that the process of photoinduced anion exchange had no noticeable influence on the size and morphology of the resulting PeNCs.

The strategy of NIR-triggered anion exchange of $CsPbX_3$ PeNCs provides significant advantages as compared to that using UV light directly, because it can not only avoid the photodamage of PeNCs upon prolonged UV exposure, but also offer a high degree of remote control by using the NIR laser as the excitation source. As shown in Fig. 4d and Supplementary Fig. 37, the emission peak of PeNCs derived from the mixture of RIC/DBM/$CsPbCl_3$ experienced a continuous red-shift from 408 to 511 nm upon 980-nm excitation for 110 min, with PLQYs in the range of 52.3%–89.2% (Supplementary Table 15). Similarly, the emission peak of PeNCs derived from the mixture of RIC/IDP/$CsPbBr_3$ shifted to red gradually from 510 to 696 nm upon 980-nm excitation for 22 min (Fig. 4e, Supplementary Fig. 38), with PLQYs in the range of 72.0%–98.1% (Supplementary Table 16). The faster anion exchange rate of $CsPbBr_3$ in IDP than that of $CsPbCl_3$ in DBM is ascribed to the weaker C–I bond than C–Br bond, which makes IDP more vulnerable to photoreduction[58]. By contrast, the emission peaks of PeNCs derived from DBM/$CsPbCl_3$ and IDP/$CsPbBr_3$ without RIC remained essentially unchanged upon 980-nm excitation for 3 h (Supplementary Fig. 39), underscoring the key role of NIR-to-UV UC processes in NIR-triggered anion exchange of $CsPbX_3$ PeNCs. This feature enables the remote control of PeNCs with desired PL emissions by switching off the laser on demand at a specific exposure time (Supplementary Fig. 40). Moreover, the extent of halide exchange can also be controlled by tuning the power density of laser, whereby high-power excitation promoted the exchange rate (Supplementary Fig. 41). Note that the laser-induced heating effect may also contribute to the enhanced anion exchange rate due to the increased motion of the anions with the temperature rise (Supplementary Fig. 42), but it is not the cause of anion exchange as it cannot lead to the cleavage of the C–X bonds of DBM and IDP (Supplementary Fig. 43). For comparison, we also performed UV-triggered anion exchange of $CsPbX_3$ PeNCs in haloalkanes without RIC NCs by using a 365-nm light-emitting diode (LED) as the excitation source under otherwise identical conditions. The results showed that 365-nm UV-LED can trigger the anion exchange of $CsPbX_3$ PeNCs with an accelerated reaction rate (Supplementary Fig. 44), but it is detrimental to the PLQYs (decreased to 10.0% for $CsPbI_3$) due to the serious photodamage of PeNCs upon direct UV exposure (Fig. 4f, g, Supplementary Tables 17, 18). Besides, although conventional UC materials such as $NaYF_4$:$Yb^{3+}$/$Er^{3+}$ may also be applied for the NIR-triggered anion exchange of $CsPbX_3$ PeNCs (Supplementary Fig. 45, Supplementary Tables 19 and 20), they are less efficient with a significantly reduced reaction rate, due to the much weaker UV-UCL than that of $Rb_3InCl_6$:$Yb^{3+}$/$Er^{3+}$. These results reveal the great potential of $Rb_3InCl_6$:$Yb^{3+}$/$Er^{3+}$ in NIR-to-UV utilization, and also offer a new way for the post-synthesis modification of PeNCs through NIR-triggered halide exchange towards various optoelectronic applications such as perovskite-based color conversion and patterning.

## Discussion

In summary, we have achieved abnormally strong UV-UCL of $Er^{3+}$ in $Yb^{3+}$/$Er^{3+}$ co-doped 0D $Rb_3InCl_6$ under 980-nm excitation. Mechanistic investigation through concentration- and power-dependent UCL spectroscopic analyses unraveled that the unusual upconverted UV emission of $Er^{3+}$ is dictated by the spatially confined 0D structure of $Rb_3InCl_6$ with a large interionic distance and low phonon energies, which facilitated the confined ET between $Yb^{3+}$ and $Er^{3+}$ within a short range while suppressing the energy losses through the MPR and long-range EM processes. This promoted the population of $Er^{3+}$ at the $^4G_{11/2}$ state, resulting in the intense UV emission at 384 nm with a large UV-to-green ratio ($I_{384}/I_{554}$) of 0.864 and a QY of 0.12% upon 980-nm excitation at a power density of 60 W cm$^{-2}$, which are significantly higher than those of conventional UC materials. Furthermore, we demonstrated the application of $Rb_3InCl_6$:$Yb^{3+}$/$Er^{3+}$ NCs as an efficient NIR-to-UV transducer for NIR-triggered anion exchange of $CsPbX_3$ PeNCs in haloalkanes with a high efficiency and remote controllability. These findings provide fundamental insights into the effect of ET and crystal lattice engineering on the UC dynamics of $Ln^{3+}$, which lay a foundation for the UC materials innovation based on the confined ET between different $Ln^{3+}$ emitters towards versatile applications such as NIR-to-UV utilization.

## Methods

### Materials

RbCl (99.99%), $Rb_2CO_3$ (99.8%), $Cs_2CO_3$ (99.99%), $InCl_3$ (99.99%), $In(CH_3COO)_3$ (99.99%), $YbCl_3 \cdot 6H_2O$ (99.99%), $Yb(CH_3COO)_3 \cdot 4H_2O$ (99.99%), $Yb(NO_3)_3 \cdot 6H_2O$ (99.9%), $ErCl_3 \cdot 6H_2O$ (99.99%), $Er(CH_3COO)_3 \cdot 4H_2O$ (99.99%), $Er(NO_3)_3 \cdot 6H_2O$ (99.9%), $TmCl_3 \cdot 6H_2O$ (99.99%), $HoCl_3 \cdot 6H_2O$ (99.99%), $Y(NO_3)_3 \cdot 6H_2O$ (99.9%), $Gd(NO_3)_3 \cdot 6H_2O$ (99.9%), oleic acid (OA, 90%), oleylamine (OAm, 90%), 1-octadecene (ODE, 90%), and trioctylphosphine (TOP, 98%) were purchased from Sigma-Aldrich (Shanghai, China). CsCl (99.999%), $Pb(CH_3COO)_3 \cdot 3H_2O$ (99.99%), benzoyl chloride (98.0%), isopropyl alcohol (99.9%), dibromomethane (DBM, 99%), and 2-iodopropane (IDP, 99%) were purchased from Aladdin (Shanghai, China). NaCl (99.5%), NaOH (96.0%), NaF (99.8%), hydrochloric acid (HCl, 37%), methanol ($CH_3OH$, 99.9%), and ethanol ($C_2H_5OH$, 99.9%) were purchased from Sinopharm Chemical Reagent Co. (Shanghai, China). All chemical reagents were used as received without further purification.

### Synthesis of $Rb_3InCl_6$:$Yb^{3+}$/$Er^{3+}$ MCs

$Yb^{3+}$ and $Er^{3+}$ singly-doped and $Yb^{3+}$/$Er^{3+}$ co-doped $Rb_3InCl_6$ MCs were synthesized via a solvothermal method by using $InCl_3$, $YbCl_3$, and $ErCl_3$ as the metal precursors and methanol as the solvent. In a typical synthesis of $Rb_3InCl_6$: 50%$Yb^{3+}$/1%$Er^{3+}$ MCs, 1.5 mmol of RbCl, 0.245 mmol of $InCl_3$, 0.25 mmol of $YbCl_3 \cdot 6H_2O$ and 0.005 mmol of $ErCl_3 \cdot 6H_2O$ were dissolved in 8.0 mL of methanol in a 20 mL Teflon autoclave. Thereafter, the solution was heated at 180 °C for 12 h in a stainless-steel autoclave. The mixture was then cooled down to 30 °C at a speed of 3 °C h$^{-1}$. Finally, the MCs were filtered out, washed three times with ethanol, and dried in an oven at 60 °C. For synthesizing $Rb_3In \cdot Cl_6$: $x$%$Yb^{3+}$/$y$%$Er^{3+}$ MCs with different $Yb^{3+}$ and $Er^{3+}$ concentrations, $(0.5 \times x)$% mmol of $YbCl_3 \cdot 6H_2O$, $(0.5 \times y)$% mmol of $ErCl_3 \cdot 6H_2O$, and $(0.5 \times (1 - x\% - y\%))$ mmol of $InCl_3$ were used under otherwise identical conditions.

### Synthesis of $Rb_3InCl_6$:$Yb^{3+}$/$Er^{3+}$ NCs

$Rb_3InCl_6$:$Yb^{3+}$/$Er^{3+}$ NCs were synthesized through a modified hot-injection method by using benzoyl chloride as the halide source. In a typical process of $Rb_3InCl_6$: 50%$Yb^{3+}$/1%$Er^{3+}$ NCs, 0.45 mmol of $Rb_2CO_3$, 0.147 mmol of $In(CH_3COO)_3 \cdot 4H_2O$, 0.15 mmol of $Yb(CH_3COO)_3 \cdot 4H_2O$, and 0.003 mmol of $Er(CH_3COO)_3 \cdot 4H_2O$ were added in a 100 mL two-neck flask containing 4 mL of OA, 1 mL of OAm, and 12 mL of ODE. The mixture was then heated to 120 °C under an $N_2$ flow with constant stirring for 1 h to dissolve the powder and remove the moisture from the raw materials. After the solution became clear, the temperature was raised to 200 °C and stabilized for 10 min, followed by rapid injection of 1.8 mmol of benzoyl chloride into the hot solution. After

30 s of reaction, the mixture was cooled down to room temperature (RT) by an ice-water bath. The NCs were precipitated by centrifugation of the mixture at $4000 \times g$ for 5 min, washed with 2 mL of cyclohexane, and collected by centrifugation again at $12,000 \times g$ for 5 min. For synthesizing $Rb_3InCl_6$: $x\%Yb^{3+}/y\%Er^{3+}$ NCs with different $Yb^{3+}$ and $Er^{3+}$ concentrations, $(0.3 \times x\%)$ mmol of $Yb(CH_3COO)_3 \cdot 4H_2O$, $(0.3 \times y\%)$ mmol of $Er(CH_3COO)_3 \cdot 4H_2O$, and $(0.3 \times (1 - x\% - y\%))$ mmol of $In(CH_3COO)_3 \cdot 4H_2O$ were used under otherwise identical conditions.

### Synthesis of $Cs_2NaInCl_6$:$Yb^{3+}$/$Er^{3+}$ MCs
$Yb^{3+}/Er^{3+}$ co-doped $Cs_2NaInCl_6$ double perovskite MCs were synthesized via a modified solvothermal method. In a typical synthesis of $Cs_2NaInCl_6$: $50\%Yb^{3+}/1\%Er^{3+}$ MCs, 0.8 mmol of CsCl, 0.4 mmol of NaCl, 0.4 mmol of $In(CH_3COO)_3 \cdot 4H_2O$, 0.2 mmol of $Yb(CH_3COO)_3 \cdot 4H_2O$, and 0.004 mmol of $Er(CH_3COO)_3 \cdot 4H_2O$ were dissolved in 8.0 mL of HCl in a 20-mL Teflon autoclave. Thereafter, the solution was heated at 180 °C for 12 h in a stainless-steel autoclave. The solution was then cooled down to 30 °C at a speed of 3 °C $h^{-1}$. Finally, the MCs were filtered out, washed three times with isopropyl alcohol, and dried in an oven at 60 °C.

### Synthesis of $NaYF_4$:$Yb^{3+}$/$Er^{3+}$ and $NaGdF_4$:$Yb^{3+}$/$Er^{3+}$ MCs
$NaYF_4$: $18\%Yb^{3+}/2\%Er^{3+}$ MCs were synthesized via a solvothermal method. In a typical synthesis, 0.6 g NaOH was dissolved in 9 mL of distilled water, and then 20 mL of OA and 9 mL of ethanol was added. After vigorous stirring for 30 min, 2 mL of aqueous solution containing 0.8 mmol of $Y(NO_3)_3 \cdot 6H_2O$, 0.18 mmol of $Yb(NO_3)_3 \cdot 6H_2O$, and 0.02 mmol of $Er(NO_3)_3 \cdot 6H_2O$ was added, followed by addition of 4.0 mL of aqueous solution containing 4 mmol of NaF. The mixture was stirred for 30 min and then transferred to a 50 mL Teflon-lined autoclave and hydrothermally treated at 200 °C for 24 h. After the autoclave was cooled to RT, the precipitates were separated by centrifugation, washed with ethanol and distilled water for three times, and then dried at 60 °C. For synthesizing $NaGdF_4$: $18\%Yb^{3+}/2\%Er^{3+}$ MCs, $Gd(CH_3COO)_3 \cdot 6H_2O$ instead of $Y(CH_3COO)_3 \cdot 6H_2O$ was used under otherwise identical conditions.

### Synthesis of $CsPbX_3$ PeNCs
$CsPbX_3$ (X = Cl, Br, and I) PeNCs were synthesized via a modified hot-injection method by using HX as the halide source to precipitate the PeNCs. In a typical synthesis of $CsPbCl_3$ PeNCs, 0.5 mmol of $Pb(CH_3COO)_3 \cdot 3H_2O$ and 0.1 mmol of $Cs_2CO_3$ were mixed with 1 mL of OA, 1 mL of OAm, 1 mL of TOP, and 6 mL of ODE in a 50 mL three-neck round-bottom flask. The resulting mixture was heated to 120 °C under a $N_2$ flow with constant stirring for 1 h to form a clear solution. The temperature was then raised up to 180 °C and stabilized for 10 min, followed by rapid injection of 1.5 mmol of HCl into the hot solution. After 10 s, the reaction mixture was cooled down to RT by an ice-water bath and centrifuged at 12,000 rpm for 5 min to collect the PeNCs. The precipitate was then dispersed in 1 mL of cyclohexane and centrifuged again at $12,000 \times g$ for 5 min. After centrifugation, the supernatant was discarded and the PeNCs were redispersed in 30 mL of cyclohexane. Finally, 20 μL of TOP was added and ultrasonicated for 1 min to improve the PLQYs and long-term stability of the PeNCs. For synthesizing $CsPbBr_3$ PeNCs, 1.5 mmol of HBr was injected into the hot solution under otherwise identical conditions.

### NIR-triggered anion exchange of $CsPbX_3$ PeNCs
We demonstrated the NIR-triggered anion exchange of $CsPbX_3$ PeNCs by using DBM and IDP as the $Br^-$ and $I^-$ sources, respectively. For the anion exchange of $CsPbCl_3$ to $CsPbBr_3$, 0.05 mmol of $CsPbCl_3$ PeNCs and 0.05 mmol of $Rb_3InCl_6$: $50\%Yb^{3+}/1\%Er^{3+}$ (RIC) NCs were dispersed in the mixed solution of DBM (1 mL) and cyclohexane (1 mL) with a volume of 1:1 in in a 3 mL quartz cuvette at RT. To improve the PL efficiency of the resulting $CsPbX_3$ PeNCs, a trace amount of TOP (≈10 μL) was added. Then, the mixture was subjected to

photoirradiation with a 980-nm diode laser, and the reaction was tracked by extracting aliquots of the reaction mixture at different time intervals, followed by the measurement of the optical absorption and PL spectra. It is worth mentioning that, we extracted the PeNCs with PL emission at 450 nm through centrifugation, which were used for further halide exchange to improve the exchange rate. For the anion exchange of $CsPbBr_3$ to $CsPbI_3$, 0.05 mmol of $CsPbBr_3$ PeNCs and 0.05 mmol of $Rb_3InCl_6$: $50\%Yb^{3+}/1\%Er^{3+}$ NCs were dispersed in the mixed solution of IDP (1 mL) and cyclohexane (1 mL) with a volume of 1:1. A trace amount of TOP (≈10 μL) was added to improve the PL efficiency of the PeNCs. Then, the mixture was subjected to photoirradiation with a 980-nm diode laser, and the reaction was tracked by extracting aliquots of the reaction mixture at different time intervals, followed by the measurement of the optical absorption and PL spectra.

**Synchrotron X-ray absorption fine spectroscopy (XAFS) analyses.** The XAFS analyses including the X-ray absorption near-edge structure (XANES) and EXAFS studies were conducted on the easy XAFS 300+ (energy 5–12 keV, Bragg Angle 55°–85°, resolution ratio 0.5–1.5 eV, luminous flux 7–9 keV) at Fujian Science & Technology Innovation Laboratory for Optoelectronic Information of China. The obtained XAFS data was processed in Athena (version 0.9.26) for background, pre-edge line, and post-edge line calibrations. Then, Fourier transformed fitting was carried out in Artemis. The $k^2$ weighting, $k$-range of 2–10 $Å^{-2}$, and $R$ range of 1–2 Å were used to fit $Rb_3InCl_6$: $50\%Yb^{3+}/1\%Er^{3+}$ MCs.

### DFT calculations
We employed the first principles to perform the DFT calculations within the generalized gradient approximation (GGA) using the Perdew-Burke-Ernzerhof (PBE) formulation. The projected augmented wave (PAW) potentials were used to describe the ionic cores and the valence electrons were taken into account using a plane wave basis set with a kinetic energy cutoff of 400 eV. Partial occupancies of the Kohn−Sham orbitals were allowed using the Gaussian smearing method with a width of 0.05 eV. The electronic energy was considered self-consistent when the energy change was smaller than $10^{-6}$ eV. A geometry optimization was considered convergent when the energy change was smaller than 0.03 eV $Å^{-1}$. The Brillouin zone integration is performed using $1 \times 1 \times 1$ Monkhorst-Pack $k$-point sampling for $3 \times 3 \times 2$ supercell structure and $1 \times 4 \times 3$ mesh for unit cell.

### Characterization
Powder XRD patterns were collected with an X-ray diffractometer (MiniFlex2, Rigaku) using Cu $K_{\alpha 1}$ radiation ($\lambda = 0.154187$ nm). ICP-AES analyses were conducted on an ICP-AES spectrometer (Ultima2, Jobin Yvon). SEM measurements were performed by using a JSM-6700F SEM equipped with EDS. TEM measurements were performed by using a TECNAI G2 F20 TEM. XPS was carried out on a Thermo Fisher ESCALAB 250Xi using Al $K_\alpha$ (1486.6 eV) and He $I_\alpha$ (21.2 eV) as the sources of radiation. Raman spectra were recorded using a micro-Raman spectrometer (Labram HR Evolution, Horiba Scientific) upon laser excitation at 785 nm. Optical absorption spectra were translated from the UV-vis diffuse reflectance spectra, which were acquired with a Perkin-Elmer Lambda 950 UV/Vis/NIR spectrometer by using $BaSO_4$ as a reference. UCL and PL spectra were measured on an FLS980 spectrometer (Edinburgh), and a 980-nm continuous-wave diode laser (2 W) was used the excitation source for the UCL measurements. UCL and PL lifetimes were measured on an FLS980 spectrometer (Edinburgh) equipped with a tunable midband Optical Parametric Oscillator (OPO) pulsed laser as the excitation source (410–2400 nm, 10 Hz, pulse width ≤5 ns, Vibrant 355II, OPO-TEK). The absolute PL and UC QYs of the samples were obtained by employing a standard barium sulfate coated integrating sphere (150 nm in diameter, Edinburgh) as the sample chamber that was

mounted on the FLS980 spectrometer with the entry and output port of the sphere located in 90° geometry from each other in the plane of the spectrometer. A standard tungsten lamp was used to correct the optical response of the instrument. UCL and PL photographs were taken by using a Huawei P30Pro cell phone with a 750 nm short-wave pass filter. All the spectral data were recorded at RT unless otherwise noted.

## Data availability

The data that support the findings of this study are available from the corresponding authors upon request.

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

## Acknowledgements
This work was supported by the National Key R&D Program of China (2022YFB3503700), the National Natural Science Foundation of China (Nos. U22A20398, 22135008, 12474418, 12174391, and 12374389), and the Natural Science Foundation of Fujian Province (2024J010038, and 2024I0040).

## Author contributions
Wen Z. conducted the experiments and measurements, analyzed the data, and prepared the manuscript. Z.X. performed the DFT calculations. D.Y. and Z.S. helped with the materials synthesis and characterizations. W. Zhang, H.Z., and J.X. helped with the UCL measurements. W. Zheng, P.H., and X.C. conceived the project and revised the paper. All authors contributed to the general discussion and analysis of the manuscript.

## Competing interests
The authors declare no competing interests.
