## [Transparent Peer Review file · Nature Communications]

Intense upconverted ultraviolet emission of Er³⁺ through confined energy transfer in Yb³⁺/Er³⁺ co-doped Rb₃InCl₆

Corresponding Author: Professor Xueyuan Chen

Version 0:

Reviewer comments:

Reviewer #1

(Remarks to the Author)

The authors present an interesting candidate for upconversion using the now established Er³⁺/Yb³⁺ couple. Upon irradiation with 980 nm (NIR radiation) of Yb³⁺, not only green, but even NUV upconverted luminescence based on a 4G_{11/2} → 4I_{15/2} transition is observed. For that purpose, they use a halide-based host compound with low cutoff phonon energy (< 300 cm⁻¹), Rb₃InCl₆. Based on its structure (which I will also discuss in one my comments below), energy migration among the incorporated lanthanoid ions is highly limited that allows higher activator fractions without significant loss in emission intensity. Next to the fundamental upconversion studies, its use as latent UV light source for the activation of C-X bonds (X = halogen atom) in organic solvents was probed, which was followed by an anion exchange of additionally present perovskite nanocrystals. The authors could demonstrate that this concept does work without significant sacrifice in photoluminescence quantum yields of the perovskite nanocrystals.

The work is appealing in using established concepts of the optical properties of lanthanoid ions (control of nonradiative decay) and the work was performed with the care that is well-known from the authors. My overall impression is positive. However, several issues remain that need to be carefully addressed prior to final publication at Nature Communications.

1) One motivation/add-on of this work was to demonstrate that the NIR-to-UV UC with Rb₃InCl₆:Er³⁺/Yb³⁺ could allow light-induced anion exchange of perovskite nanocrystals in organic solvents. A critical question could be why no immediate UV irradiation with e.g. 375 nm is a suited alternative. For that wavelength range, cheap semiconductor LEDs with sufficient power densities are available. The key motivation could be clarified a bit more.

2) In the results section, the authors state that Rb₃InCl₆ crystallizes in a clinopyroxene structure type. That cannot be true. (Clino-)pyroxenes are inosilicates with linear chains of two vertex-sharing [SiO₄]⁴⁻ tetrahedra that result in [Si₂O₆]⁴⁻ units. Examples include enstatite (MgSiO₃ = Mg₂[Si₂O₆]) or wollastonite (CaSiO₃ = Ca₂[Si₂O₆]). Among the pyroxenes, the clinopyroxenes distort towards a monoclinic crystal system with space group type C_{2/c} (no. 15) (this is the case in wollastonite, for example).

Rb₃[InCl₆] in turn does not contain any vertex-connected tetrahedra, which makes it difficult to find an analogy to clinopyroxenes. Instead, it is basically a monoclinically distorted double perovskite crystallizing in a Rb₃[YCl₆] structure type, which can be generated if half of the octahedra are basically tilted by ~ 45°. A good example for that is found in the work of Woodward et al., DOI: 10.1021/acs.inorgchem.0c02248. It will be good to be precise here.

3) An interesting finding is that the UC decay time of the 4G_{11/2} level is higher than that found in Cs₂NaInCl₆:Er³⁺, Yb³⁺ microcrystals despite structural similarities and their similarly low phonon energies (Cs₂NaInCl₆ should have a cutoff phonon energy of around 280 cm⁻¹). In fact, the respective decay time is even lower than in nanocrystalline β-NaYF₄:Er³⁺, Yb³⁺. Did the authors check vibrational spectra to exclude any impact of surface-attached ligands?

4) In contrast to the other low cutoff phonon energy hosts, the intensity ratio I(2H_{11/2})/I(4S_{3/2}) in Rb₃[InCl₆] at room temperature is unusually large (> 1 according to Fig. 2a). To what extent is this intensity ratio caused by laser-induced heating with the 980 nm light source? Connected to that is the question what is the UC quantum yield for the different microcrystalline samples given a reference power density and similar absorption strength (i.e. concentration of Yb³⁺) in the two regarded samples? I fear that this point also plays a dominant role for the overall intensity of the 4G_{11/2}-based emission in the UV range.

It may be good to independently calibrate Er³⁺ as a luminescent thermometer with low incident power densities to extract the local heating effects.

5) On p. 11, the authors use the term "density function theory" but probably mean density functional theory.

6) In Fig. 3c, the authors discuss a diffusion-limited model for the luminescence decay with a fast decay component characterizing the energy migration process, while the slower decay represents the decay of the collective, non-interacting ensemble. While the general concept is correct, this is not clearly visible from the decay curve. Neither a specific model nor a residual plot is depicted that could justify the validity of this anticipated model. That should be clarified for the general reader.

I hope that the authors carefully reflect and revise the manuscript.

Reviewer #2

(Remarks to the Author)

This manuscript describes a relevant finding in the field of luminescence up-conversion materials, such as the observation of an unprecedentedly efficient conversion of near infrared (NIR, $\lambda=980$ nm) to near ultraviolet (UV, $\lambda=384$ nm) light. To the best of my knowledge, there are no precedents of a result like this in the related literature. This achievement is possible by co-doping with Yb³⁺ and Er³⁺ cations a 0D dimensional network of Rb₃InCl₆, rationally chosen as a host matrix due to its low phonon energies, which in turn allows foreseeing a mitigated non-radiative multiphonon relaxation. The effect is adequately quantified by comparing the UV up-conversion process described to others well studied, like the NIR-to-green light conversion both in the targeted matrix and others commonly employed in the field (such as NaYF₄, for instance). The authors also carefully addressed the analysis of the potential mechanisms involved in the observed effect, identifying the much longer lifetime of the 4G_{11/2} state and, most interestingly, the much larger Yb³⁺ interionic distance (both compared to the values reported for other standard matrices used in the field of up-conversion) as the key factors responsible for its magnitude. The discussion on the inhibition of long range energy migration processes in the proposed 0-dimensional matrix, as opposed to what has been previously observed in other hosts, is particularly clarifying of the surprising absence of quenching of the UV emission even at very high Yb³⁺ concentrations (50%).

In what follows, the authors will find some specific questions or comments that should be addressed in a revised version of the manuscript:

1) I agree with the authors when they expect the reported effect to have an impact in the field of biomedicine, as a significant conversion of light within the near infrared transparency window of biological tissue into potentially therapeutic higher energy radiation in targeted regions might be of great interest for certain tumor treatments. In this context, my only remark would be that the application of this effect to the anion exchange between lead halide perovskite nanocrystals, given at the end of the manuscript as a kind of bonus, may be better suited to a separate analysis and publication, as it is too specific and lacks the general relevance of the rest of the information provided in the paper.

2) 1. 0D materials may also refer to nanomaterials, typically spheres, with all dimensions below 100 nm. A sentence may be added in introduction to avoid any misunderstanding.

3) Knowing the stringent effect of doping content on UV UC property, the real concentrations of dopants in NCs should be measured from ICP-AES, just as the authors did with the MCs, to allow a more accurate comparison between MCs and NCs. In general I miss some characterizations of the NCs (SEM, EDS, XPS...).

4) Authors give a complete analysis of UV UC lifetime measurements to explain the abnormal high efficiency of UV UC in their material. For instance, they observe that increasing Er³⁺ content decreases the 4G_{11/2} UCL lifetime as a result of back ET (= bad thing) while increasing Yb³⁺ content decreases the 4G_{11/2} UCL lifetime as a result of improved ETU (= good thing). As the UV UC efficiency depends on several competing processes (EM, ET, back ET...), I miss a general conclusion as a rule of thumb here on the UV UC lifetime. Can the obtained results pave the way on how to choose the best UV UC phosphor based on its UV UC lifetime ?

5) Results demonstrate that a close vicinity between Yb³⁺-Er³⁺ is necessary while long Yb³⁺-Yb³⁺ distances are favorable to decrease the EM. Can authors comment on the accuracy of the VASP calculations compared to real materials? Especially in the context of the 2 different synthesis methods used. Indeed MCs synthesis may be thermodynamically driven (12h at 180 °C and slow cooling 3 °C h⁻¹) while the synthesis of NCs may be kinetically driven (30 s at 200 °C with instant temperature quenching).

Reviewer #3

(Remarks to the Author)

Reviewer #4

(Remarks to the Author)

The authors display a work in which lanthanide ions are incorporated into an oxide perovskite matrix. An overall

enhancement of the UV emission of Er is observed, which is interesting, as typically the green and red emission is stronger. Despite this being an interesting result, this work seems like just another article of lanthanide doping in an inert matrix. It is not very clear why and how this particular matrix was chosen and why it has this impact on the UV Er emission. The rationale behind it as well as the mechanism is not elaborated. I am doubtful about the results on the NIR triggered halide cation exchange. The term abnormal in the title is not scientific. It is a nice article, well done, but in my view not bringing sufficient new insight for the readership of Nature Com.

Some more detailed points below:

- 1) It is not clear which is the novelty of this specific approach of implementing Rb₃InCl₆ as a matrix for the lanthanide doping. It is not clear if the authors were the first to display lanthanide doping in perovskite oxides, if so the question arises about the rationale behind this design. If they are not the first then the literature review needs to become more exhaustive.
- 2) The authors use the term abnormal which is not very scientific. What exactly does it mean. If something is abnormal then I expect something else to be normal. Which is the normal part that makes this PL abnormal?
- 3) I understand that the PL in the UV is increased with respect to other examples of Er doped nanocrystals. The authors try to give an explanation of the mechanism, but it is very briefly discussed only. The mechanistic insight in my view however is very important for a journal of this level and the authors would have needed to show this more in detail and with a more exhaustive discussion.
- 4) The application of using NIR light to trigger ion exchange in perovskite nanocrystal is not really convincing. First of all: what is the advantage of UV light. It's way more accessible than IR. Did the authors compare the efficiency with respect to pure NIR light? What about multi photon processes or heat as a result of the NIR excitation of the halide perovskites as a reason for the enhanced ion exchange. The reference experiments of a solution of perovskite nanocrystals without up-converting nanocrystals but with the exposure to NIR light needs to be performed.
- 5) What is the difference between the micro and the nano-crystals. The authors seem to switch between one or the other depending on the experiment they are performing. But is this valid? What is the difference between them, how does their size impact the specific property they are looking at? More insights into this must be given.

Version 1:

Reviewer comments:

Reviewer #1

(Remarks to the Author)

The authors have carefully addressed the referees' concerns and the level of this manuscript has improved a lot. Nonetheless, I fear there are still issues to be discussed.

1) The authors used a Yokota-Tanimoto diffusion model to describe the energy transfer mechanistically. However, this does not take into account the discrete nature of the crystal structure with pre-defined cation-cation distances. A shell model-type approach seems more suited to me, see e.g. the works by Rabouw and Meijerink 2014, <https://doi.org/10.1038/ncomms4610>.

2) The authors tested the anion exchange in the perovskite NCs upon NIR irradiation and in the absence of the Rb₃InCl₆:Er, Yb NCs, which is a very important experiment. The incident power density was, however, 60 W cm⁻² (see Fig. 4). At such a high local power density, I would definitely expect local heating effects. How can the authors be sure that the anion exchange is not a consequence of local heating and thus, thermally stimulated reactivity? It may be good to have a control setup here by e.g. performing an experiment under varying temperatures and cross-correlate the results.

Reviewer #2

(Remarks to the Author)

I have reviewed the response to the referees' comments and I am satisfied with it. They have satisfactorily addressed my concerns and provided a significant amount of new evidence to support their claims. I have no further suggestions, I believe this work is now suitable for Nature Communications standards and broad audience.

Reviewer #4

(Remarks to the Author)

The authors have responded to my concerns.

Version 2:

Reviewer comments:

Reviewer #1

(Remarks to the Author)

The authors have clarified my remaining concerns and the manuscript appears publishable to me now. It is good to see how the (already initially high!) quality even improved in each step and I am sure that the study will be beneficial for the community.

Reply to the Comments of Reviewer #1

General Comment:

The authors present an interesting candidate for upconversion using the now established Er^{3+}/Yb^{3+} couple. Upon irradiation with 980 nm (NIR radiation) of Yb^{3+} , not only green, but even NUV upconverted luminescence based on a ${}^4G_{11/2} \rightarrow {}^4I_{15/2}$ transition is observed. For that purpose, they use a halide-based host compound with low cutoff phonon energy ($< 300 \text{ cm}^{-1}$), Rb_3InCl_6 . Based on its structure (which I will also discuss in one my comments below), energy migration among the incorporated lanthanoid ions is highly limited that allows higher activator fractions without significant loss in emission intensity. Next to the fundamental upconversion studies, its use as latent UV light source for the activation of C-X bonds ($X = \text{halogen atom}$) in organic solvents was probed, which was followed by an anion exchange of additionally present perovskite nanocrystals. The authors could demonstrate that this concept does work without significant sacrifice in photoluminescence quantum yields of the perovskite nanocrystals.

The work is appealing in using established concepts of the optical properties of lanthanoid ions (control of nonradiative decay) and the work was performed with the care that is well-known from the authors. My overall impression is positive. However, several issues remain that need to be carefully addressed prior to final publication at Nature Communications.

Response:

We greatly appreciate the reviewer for his/her positive comments on our manuscript and the efforts to improve the quality of our manuscript. We have carefully looked into all the helpful suggestions by the reviewer and made all the requested changes that have been reflected either in the text or in the Supplementary Information. A point-by-point response is noted below.

Comment #1:

One motivation/add-on of this work was to demonstrate that the NIR-to-UV UC with $Rb_3InCl_6: Er^{3+}/Yb^{3+}$ could allow light-induced anion exchange of perovskite nanocrystals in organic solvents. A critical question could be why no immediate UV irradiation with e.g. 375 nm is a suited alternative. For that wavelength range, cheap semiconductor LEDs with sufficient power densities are available. The key motivation could be clarified a bit more.

Response:

Many thanks for this valuable suggestion. The strategy of near-infrared (NIR)-triggered anion exchange of $CsPbX_3$ perovskite nanocrystals (PeNCs) provides significant advantages as compared to that using ultraviolet (UV) light directly, because it can not only avoid the photodamage of PeNCs upon prolonged UV exposure, but also offer a high degree of remote control by using the NIR laser as the excitation source. As per your suggestion, we have supplemented additional control experiments by using a 365-nm UV light-emitting diode (LED) (3 W) instead of the 980-nm NIR laser as the excitation source for

the photoinduced anion exchange of CsPbX₃ PeNCs in haloalkanes without Rb₃InCl₆: Yb³⁺/Er³⁺ (RIC) NCs under otherwise identical conditions.

The results showed that 365-nm UV-LED can trigger the anion exchange of CsPbX₃ PeNCs with an accelerated reaction rate as compared to that upon 980-nm NIR laser excitation, because of the higher power and larger area of UV light acting on the reaction system upon direct UV illumination. The reaction times required for completing the anion exchange from CsPbCl₃ to CsPbBr₃ and from CsPbBr₃ to CsPbI₃ were 17 and 8 minutes, respectively, under UV illumination (Supplementary Fig. 43), much shorter than those upon NIR irradiation (110 and 22 minutes) (Figs. 4d and e). However, the photoluminescence (PL) quantum yields (QYs) of PeNCs after UV-triggered anion exchange decreased significantly from 50.8% for CsPbCl₃ to 43.2% for CsPbBr₃ and from 88.7% for CsPbBr₃ to 10.0% for CsPbI₃ (Figs. 4f and g and Supplementary Tables 17 and 18), due to the serious photodamage of PeNCs especially CsPbI₃ upon prolonged UV exposure.

Such UV-induced photodamage of PeNCs can be avoided by the strategy of NIR-triggered anion exchange, yielding significantly improved PLQYs in the range of 52.3–98.1% (Supplementary Tables 15 and 16). Additionally, the use of NIR laser enables the remote control of PeNCs with desired PL emissions by tuning the on/off state and the power density of laser as well as the irradiation time (Supplementary Figs. 40 and 41). These results demonstrate the remarkable advantages of our strategy of NIR-triggered anion exchange of CsPbX₃ PeNCs, which offers a new way for the post-synthesis modification of PeNCs towards various optoelectronic applications. See also text discussion newly added in lines 1-3 from the bottom of page 18 and lines 1-4 of page 19 and Fig. 4 in the revised manuscript, and experimental data newly added in Supplementary Tables 17 and 18 and Supplementary Fig. 43 in the revised Supplementary Information.

Text discussion newly added in the main manuscript:

Lines 1-3 from the bottom of page 18 and lines 1-4 of page 19: “For comparison, we also performed UV-triggered anion exchange of CsPbX₃ PeNCs in haloalkanes without RIC NCs by using a 365-nm light-emitting diode (LED) as the excitation source under otherwise identical conditions. The results showed that 365-nm UV-LED can trigger the anion exchange of CsPbX₃ PeNCs with an accelerated reaction rate (Supplementary Fig. 43), but it is detrimental to the PLQYs (decreased to 10.0% for CsPbI₃) due to the serious photodamage of PeNCs upon direct UV exposure (Figs. 4f and g and Supplementary Tables 17 and 18).”

Revised Fig. 4 in the main manuscript:

Fig. 4. NIR-triggered anion exchange of CsPbX₃ PeNCs using Rb₃InCl₆: Yb³⁺/Er³⁺. (a) Schematic of NIR-triggered anion exchange of CsPbX₃ PeNCs in haloalkanes by using Rb₃InCl₆: Yb³⁺/Er³⁺ NCs as the NIR-to-UV transducer. (b) PL emission spectra ($\lambda_{\text{ex}} = 365 \text{ nm}$) and (c) PL decay curves of CsPbX₃ PeNCs with different halide compositions, derived from anion exchange based on the mixtures of RIC/DBM/CsPbCl₃ and RIC/IDP/CsPbBr₃ upon 980-nm NIR laser exposure. The insets in (b) show the PL photographs of the resulting PeNCs under 365-nm irradiation. Time dependence of the PL emission peaks ($\lambda_{\text{ex}} = 365 \text{ nm}$) for (d) CsPbCl₃ \rightarrow CsPbBr₃ in DBM/CsPbCl₃ and RIC/DBM/CsPbCl₃ and (e) CsPbBr₃ \rightarrow CsPbI₃ in IDP/CsPbBr₃ and RIC/IDP/CsPbBr₃ upon 980-nm irradiation at a power density of 60 W cm^{-2} . PLQYs for PeNCs derived from the photoinduced anion exchange (f) from CsPbCl₃ to CsPbBr₃ in DBM/CsPbCl₃ ($\lambda_{\text{ex}} = 365 \text{ nm}$) and RIC/DBM/CsPbCl₃ ($\lambda_{\text{ex}} = 980 \text{ nm}$) and (g) from CsPbBr₃ to CsPbI₃ in IDP/CsPbBr₃ ($\lambda_{\text{ex}} = 365 \text{ nm}$) and RIC/IDP/CsPbBr₃ ($\lambda_{\text{ex}} = 980 \text{ nm}$), respectively.

Experimental data newly added in the Supplementary Information:

Supplementary Table 17. Time dependence of the PL emission peak (λ_{em}) and PLQY for PeNCs derived from the mixture of DBM/CsPbCl₃ upon 365-nm UV-LED (3 W) irradiation.

Time (min)	λ_{em} (nm)	PLQY (%)
------------	----------------------------	----------

0	406	50.8
1	430	60.3
2	446	68.0
3	456	74.7
5	468	79.2
8	485	70.1
12	497	58.9
17	510	43.2

Supplementary Table 18. Time dependence of the PL emission peak (λ_{em}) and PLQY for PeNCs derived from the mixture of IDP/CsPbBr₃ upon 365-nm UV-LED (3 W) irradiation.

Time (min)	λ_{em} (nm)	PLQY (%)
0	512	88.7
1	541	80.1
2	575	71.3
3	608	59.5
5	639	45.8
6	670	33.9
7	680	22.1
8	695	10.0

Supplementary Figure 43. Time-dependent PL emission spectra ($\lambda_{\text{ex}} = 365 \text{ nm}$) for (a) $\text{CsPbCl}_3 \rightarrow \text{CsPbBr}_3$ in DBM/ CsPbCl_3 and (c) $\text{CsPbBr}_3 \rightarrow \text{CsPbI}_3$ in IDP/ CsPbBr_3 upon 365-nm UV-LED (3 W) irradiation. Time dependence of the PL emission peaks ($\lambda_{\text{ex}} = 365 \text{ nm}$) for (b) $\text{CsPbCl}_3 \rightarrow \text{CsPbBr}_3$ in DBM/ CsPbCl_3 and (d) $\text{CsPbBr}_3 \rightarrow \text{CsPbI}_3$ in IDP/ CsPbBr_3 upon 365-nm UV-LED (3 W) irradiation. The insets in (b) and (d) show the corresponding PLQYs of PeNCs. The reaction times required for completing the anion exchange from CsPbCl_3 to CsPbBr_3 and from CsPbBr_3 to CsPbI_3 were 17 and 8 minutes, respectively, under UV illumination, much shorter than those upon NIR irradiation using $\text{Rb}_3\text{InCl}_6: \text{Yb}^{3+}/\text{Er}^{3+}$ (110 and 22 minutes). However, the PLQYs of PeNCs after UV-triggered anion exchange decreased significantly from 50.8% for CsPbCl_3 to 43.2% for CsPbBr_3 and from 88.7% for CsPbBr_3 to 10.0% for CsPbI_3 , due to the serious photodamage of PeNCs especially CsPbI_3 upon prolonged UV exposure.

Comment #2:

In the results section, the authors state that Rb_3InCl_6 crystallizes in a clinopyroxene structure type. That cannot be true. (Clino-)pyroxenes are inosilicates with linear chains of two vertex-sharing $[\text{SiO}_4]^{4-}$ tetrahedra that result in $[\text{Si}_2\text{O}_6]^{4-}$ units. Examples include enstatite ($\text{MgSiO}_3 = \text{Mg}_2[\text{Si}_2\text{O}_6]$) or wollastonite ($\text{CaSiO}_3 = \text{Ca}_2[\text{Si}_2\text{O}_6]$). Among the pyroxenes, the clinopyroxenes distort towards a monoclinic crystal system with space group type $C2/c$ (no. 15) (this is the case in wollastonite, for example).

$\text{Rb}_3[\text{InCl}_6]$ in turn does not contain any vertex-connected tetrahedra, which makes it difficult to find an analogy to clinopyroxenes. Instead, it is basically a monoclinically distorted double perovskite

crystallizing in a $Rb_3[YCl_6]$ structure type, which can be generated if half of the octahedra are basically tilted by $\sim 45^\circ$. A good example for that is found in the work of Woodward et al., DOI: 10.1021/acs.inorgchem.0c02248. It will be good to be precise here.

Response:

Thank you for pointing out this inaccuracy. We have read through the excellent work by Woodward *et al.* (*Inorg. Chem.* **2020**, *59*, 14478–14485) about the crystal structure of Rb_3InCl_6 . According to Woodward *et al.*, Rb_3InCl_6 crystallizes with the Rb_3YCl_6 structure type (space group $C2/c$), which can be derived from the double perovskite structure by noncooperative tilting of isolated $[InCl_6]^{3-}$ octahedra. Each $[InCl_6]^{3-}$ octahedron is surrounded by Rb^+ cations, forming a spatially confined zero-dimensional (0D) structure at the molecular level. As per your request, we have corrected the description about the crystal structure of Rb_3InCl_6 to make it more precise. See also text discussion newly added in lines 14-17 of page 4 and Ref. 30 newly added in the revised manuscript.

Text discussion newly added in the main manuscript:

Lines 14-17 of page 4: “ Rb_3InCl_6 crystallizes with the Rb_3YCl_6 structure type (space group $C2/c$), which can be derived from the double perovskite structure by noncooperative tilting of isolated $[InCl_6]^{3-}$ octahedra.³⁰ Each $[InCl_6]^{3-}$ octahedron is surrounded by Rb^+ cations, forming a spatially confined 0D structure at the molecular level (Fig. 1a).”

Reference newly added in the main manuscript:

30 Majher, J. D., Gray, M. B., Liu, T. Y., Holzapfel, N. P. & Woodward, P. M. Rb_3InCl_6 : A monoclinic double perovskite derivative with bright Sb^{3+} -activated photoluminescence. *Inorg. Chem.* **59**, 14478-14485 (2020).

Comment #3:

An interesting finding is that the UC decay time of the $^4G_{11/2}$ level is higher than that found in $Cs_2NaInCl_6:Er^{3+}$, Yb^{3+} microcrystals despite structural similarities and their similarly low phonon energies ($Cs_2NaInCl_6$ should have a cutoff phonon energy of around 280 cm^{-1}). In fact, the respective decay time is even lower than in nanocrystalline $\beta\text{-NaYF}_4:Er^{3+}$, Yb^{3+} . Did the authors check vibrational spectra to exclude any impact of surface-attached ligands?

Response:

Yes, we have measured the Raman and Fourier transform infrared (FTIR) spectra of $Rb_3InCl_6:Yb^{3+}/Er^{3+}$ and $Cs_2NaInCl_6:Yb^{3+}/Er^{3+}$ microcrystals (MCs) to check their vibrational energies. Raman spectrum showed that $Cs_2NaInCl_6:Yb^{3+}/Er^{3+}$ MCs have a cutoff phonon energy of around 295 cm^{-1} (Supplementary Fig. 13), slightly larger than that of $Rb_3InCl_6:Yb^{3+}/Er^{3+}$ MCs (282 cm^{-1}). FTIR spectra showed negligible vibrational peaks in the range of $400\text{--}4000\text{ cm}^{-1}$ for both $Rb_3InCl_6:Yb^{3+}/Er^{3+}$ and

$\text{Cs}_2\text{NaInCl}_6:\text{Yb}^{3+}/\text{Er}^{3+}$ MCs (Fig. R1), indicating the absence of organic ligands attached on their surface. Since $\text{Rb}_3\text{InCl}_6:\text{Yb}^{3+}/\text{Er}^{3+}$ and $\text{Cs}_2\text{NaInCl}_6:\text{Yb}^{3+}/\text{Er}^{3+}$ MCs were synthesized through a similar solvothermal method without using any organic ligands, the influence of surface-attached ligands on the upconversion (UC) decay time of the $^4\text{G}_{11/2}$ level of Er^{3+} in these two MCs can be ruled out.

Figure R1. FTIR spectra of $\text{Rb}_3\text{InCl}_6:\text{Yb}^{3+}/\text{Er}^{3+}$ and $\text{Cs}_2\text{NaInCl}_6:\text{Yb}^{3+}/\text{Er}^{3+}$ MCs, showing negligible vibrational peaks in the range of 400–4000 cm^{-1} .

The UC decay times of Er^{3+} in $\text{Yb}^{3+}/\text{Er}^{3+}$ co-doped UC materials are influenced by many factors such as the crystal structure, the phonon energy, the crystal-field (CF) strength and local site symmetry of Er^{3+} , the energy transfer (ET) between Yb^{3+} and Er^{3+} , and the surface structure (*e.g.*, surface ligands) (*Chem. Soc. Rev.* **2015**, *44*, 1635-1652; *Chem. Soc. Rev.* **2015**, *44*, 1379-1415; *ACS Nano* **2018**, *12*, 4812-4823; *Light-Sci. Appl.* **2022**, *11*, 256). Considering the similarities in the phonon energies of the host materials, we deduced that the significantly longer decay time of $^4\text{G}_{11/2}$ of Er^{3+} observed in $\text{Rb}_3\text{InCl}_6:\text{Yb}^{3+}/\text{Er}^{3+}$ (190 μs) than in $\text{Cs}_2\text{NaInCl}_6:\text{Yb}^{3+}/\text{Er}^{3+}$ (47 μs) is dictated by the difference in the CF strength and local site symmetry of Er^{3+} as well as the ET between Yb^{3+} and Er^{3+} . Actually, $\text{Cs}_2\text{NaInCl}_6:\text{Yb}^{3+}/\text{Er}^{3+}$ MCs exhibited very weak UC luminescence (UCL) under 980-nm excitation (Supplementary Fig. 11), due to the high symmetry of Yb^{3+} and Er^{3+} (O_h) in $\text{Cs}_2\text{NaInCl}_6$, the inefficient ET from Yb^{3+} to Er^{3+} , and the nonradiative relaxation of Yb^{3+} through energy migration (EM) to the surface and lattice defects (*Adv. Sci.* **2022**, *9*, 2203735; *Adv. Sci.* **2023**, *10*, 2207571; *Adv. Opt. Mater.* **2023**, *11*, 2202704).

In comparison with $\text{Cs}_2\text{NaInCl}_6:\text{Yb}^{3+}/\text{Er}^{3+}$, $\text{Rb}_3\text{InCl}_6:\text{Yb}^{3+}/\text{Er}^{3+}$ have a lower site symmetry of C_{2h} for Yb^{3+} and Er^{3+} dopants and thus display much more intense UCL. Specifically, owing to the confined ET between Yb^{3+} and Er^{3+} in 0D Rb_3InCl_6 with a low phonon energy and a large interionic distance, the long-range EM among Yb^{3+} is effectively suppressed in $\text{Rb}_3\text{InCl}_6:\text{Yb}^{3+}/\text{Er}^{3+}$, resulting in the unusually strong UV-UCL from $^4\text{G}_{11/2}$ of Er^{3+} at 384 nm, along with the long decay time of 190 μs . For better clarity, we have added more text discussion about the difference in the UC decay times between $\text{Rb}_3\text{InCl}_6:\text{Yb}^{3+}/\text{Er}^{3+}$ and $\text{Cs}_2\text{NaInCl}_6:\text{Yb}^{3+}/\text{Er}^{3+}$ MCs. See also text discussion newly added in lines 15-19 of page

7 in the revised manuscript and experimental data newly added in Supplementary Fig. 13 in the revised Supplementary Information.

Text discussion newly added in the main manuscript:

Lines 15-19 of page 7: “It is worth mentioning that, $\text{Cs}_2\text{NaInCl}_6: \text{Yb}^{3+}/\text{Er}^{3+}$ MCs exhibited much weaker UCL with a short decay time for $^4\text{G}_{11/2}$ of Er^{3+} in comparison with that of $\text{Rb}_3\text{InCl}_6: \text{Yb}^{3+}/\text{Er}^{3+}$ MCs, due to the high symmetry of Yb^{3+} and Er^{3+} (O_h) and the inefficient ETU processes from Yb^{3+} to Er^{3+} in $\text{Cs}_2\text{NaInCl}_6: \text{Yb}^{3+}/\text{Er}^{3+}$,^{32, 33} though they have similarly low phonon energies (Supplementary Fig. 13).”

Reference newly added in the main manuscript:

- 32 Han, S. *et al.* Unveiling local electronic structure of lanthanide-doped $\text{Cs}_2\text{NaInCl}_6$ double perovskites for realizing efficient near-infrared luminescence. *Adv. Sci.* **9**, 2203735 (2022).
- 33 Shi, Y., Zhang, X., Wang, X. & Zhang, Y. Pure green upconversion from a multicolor downshifting perovskite crystal. *Adv. Opt. Mater.* **11**, 2202704 (2023).

Experimental data newly added in the Supplementary Information:

Supplementary Figure 13. Raman spectra of $\text{Cs}_2\text{NaInCl}_6: 50\%\text{Yb}^{3+}/1\%\text{Er}^{3+}$ MCs, showing the cutoff phonon energy of around 295 cm^{-1} . $\text{Cs}_2\text{NaInCl}_6: 50\%\text{Yb}^{3+}/1\%\text{Er}^{3+}$ MCs exhibit two intense vibrational peaks, corresponding to the breathing vibrations of Cl atoms (t_{2g} : 141 cm^{-1}) and the symmetric stretching vibrations of In–Cl bonds (a_{1g} : 295 cm^{-1}) in the $[\text{InCl}_6]^{5-}$ octahedra, respectively.

Comment #4:

In contrast to the other low cutoff phonon energy hosts, the intensity ratio $I(^2H_{11/2})/I(^4S_{3/2})$ in $Rb_3[InCl_6]$ at room temperature is unusually large (> 1 according to Fig. 2a). To what extent is this intensity ratio caused by laser-induced heating with the 980 nm light source? Connected to that is the question what is the UC quantum yield for the different microcrystalline samples given a reference power density and similar absorption strength (i.e. concentration of Yb^{3+}) in the two regarded samples? I fear that this point also plays a dominant role for the overall intensity of the $^4G_{11/2}$ -based emission in the UV range. It may be good to independently calibrate Er^{3+} as a luminescent thermometer with low incident power densities to extract the local heating effects.

Response:

Many thanks for this valuable suggestion. $Rb_3InCl_6: 50\%Yb^{3+}/1\%Er^{3+}$ MCs exhibited intense UCL with an UCQY (360–720 nm) of 0.42% upon 980-nm excitation at a power density of 60 W cm^{-2} . For comparison, $Cs_2NaInCl_6: Yb^{3+}/Er^{3+}$ MCs with optimal Yb^{3+} (50 mol%) and Er^{3+} (1 mol%) concentrations displayed very weak UCL under 980-nm excitation (Supplementary Fig. 11), due to the high symmetry of Yb^{3+} and Er^{3+} (O_h) and the inefficient ETU from Yb^{3+} to Er^{3+} in $Cs_2NaInCl_6: Yb^{3+}/Er^{3+}$ (Adv. Sci. 2022, 9, 2203735; Adv. Sci. 2023, 10, 2207571; Adv. Opt. Mater. 2023, 11, 2202704). As such, we are not able to measure the UCQY of $Cs_2NaInCl_6: 50\%Yb^{3+}/1\%Er^{3+}$ MCs, because it is beyond the detection limit of our instrument ($< 0.01\%$).

To evaluate the laser-induced heating effect on the intensity ratio of $I(^2H_{11/2})/I(^4S_{3/2})$ ($I_{524/554}$) and the UV emission (384 nm, $^4G_{11/2} \rightarrow ^4I_{15/2}$) of Er^{3+} in $Rb_3InCl_6: 50\%Yb^{3+}/1\%Er^{3+}$ MCs, we have measured the power- and temperature-dependent UCL spectra of the MCs. The laser-induced temperature rise during the UCL measurements was also recorded by using an infrared thermographic camera (H21Pro, HIKMICRO). The temperature of $Rb_3InCl_6: 50\%Yb^{3+}/1\%Er^{3+}$ MCs (powder sample) increased from room temperature (25.0 °C) to 25.3, 26.1, 27.2, 30.2, and 36.3 °C upon 980-nm laser excitation with a power density of 1, 5, 10, 30, and 60 W cm^{-2} for 2 min (Supplementary Fig. 15), during which the UCL measurement had been accomplished.

Power-dependent UCL measurements showed that the intensity ratios between the UV and green emissions (I_{384}/I_{554}) and between the emissions from the thermally coupled energy levels of $^4G_{11/2}/^2H_{9/2}$ (I_{384}/I_{409}) and $^2H_{11/2}/^4S_{3/2}$ (I_{524}/I_{554}) of Er^{3+} increased from 0.441, 3.173, and 0.639 to 1.286, 6.077, and 2.095, respectively, upon 980-nm excitation with the increasing power density from 0.5 to 150 W cm^{-2} (Supplementary Fig. 8). This indicates that the intensity ratios of I_{384}/I_{554} , I_{384}/I_{409} , and I_{524}/I_{554} depend strongly on the excitation power density, which is not unexpected since the nonlinear UCL is power dependent. Specifically, the intensity of the UV emission of Er^{3+} is comparable to that of the green emission ($I_{384}/I_{554} = 0.471$) even under excitation with a low power density of 1 W cm^{-2} , wherein the laser-induced heating effect (0.3 °C) is negligible.

Supplementary Fig. 16 shows the temperature-dependent UCL spectra (77–377 K) of $Rb_3InCl_6: 50\%Yb^{3+}/1\%Er^{3+}$ MCs under 980-nm excitation with a low power density of 10 W cm^{-2} . The overall UCL intensity of the MCs decreased gradually with the temperature rise from 77 K to 377 K, as a result of accelerated nonradiative relaxation of Yb^{3+} and Er^{3+} at higher temperatures (Nanoscale 2017, 9, 6521;

J. Mater. Chem. C **2021**, *9*, 5148). Concurrently, the intensity ratios of ${}^4G_{11/2}/{}^4S_{3/2}$ ($I_{384/554}$), ${}^4G_{11/2}/{}^2H_{9/2}$ ($I_{384/409}$), and ${}^2H_{11/2}/{}^4S_{3/2}$ ($I_{524/554}$) of Er^{3+} increased from 0.408, 7.791 and 0.357 at 77 K to 0.718, 15.34 and 1.726 at 377 K, respectively, due to the thermal-enhanced population of Er^{3+} at higher states (${}^4G_{11/2}$ and ${}^2H_{11/2}$) (*Adv. Mater.* **2013**, *25*, 4868-4874; *Mater. Horiz.* **2023**, *10*, 1816). Notably, the intensity ratio of ${}^4G_{11/2}/{}^4S_{3/2}$ ($I_{384/554}$) remained abnormally large (0.408) at 77 K, as compared to that of conventional UC materials. These results demonstrate that the unusually intense UV emission (384 nm) of Er^{3+} observed in $Rb_3InCl_6: Yb^{3+}/Er^{3+}$ MCs is not dictated by the laser-induced heating effect, since the temperature rise due to laser heating ($2.2\text{ }^\circ\text{C}$, 10 W cm^{-2}) is not high enough to cause a significant change in the population of Er^{3+} among different energy levels. Instead, it is the intrinsic properties of $Rb_3InCl_6: Yb^{3+}/Er^{3+}$ governed by the spatially confined 0D structure of the host with a low phonon energy and a large interionic distance. See also text discussion newly added in lines 3-6 of page 8 in the revised manuscript and experimental data newly added in Supplementary Figs. 8, 15, and 16 in the revised Supplementary Information.

Text discussion newly added in the main manuscript:

Lines 3-6 of page 8: “The intriguing UCL properties of $Rb_3InCl_6: Yb^{3+}/Er^{3+}$ MCs cannot be ascribed to the laser heating effect, as the temperature rise due to laser heating ($11.3\text{ }^\circ\text{C}$, 60 W cm^{-2}) is not high enough to cause a significant change in the population of Er^{3+} among different energy levels (Supplementary Figs. 15 and 16).”

Experimental data newly added in the Supplementary Information:

Supplementary Figure 8. UCL spectra of $Rb_3InCl_6: 50\%Yb^{3+}/1\%Er^{3+}$ MCs upon NIR excitation at 980 nm with power densities in the range of (a) 0.5–7 and (b) 10–150 $W\text{ cm}^{-2}$, showing unusually strong upconverted UV emission of Er^{3+} at 384 nm. (c) Intensity ratios between the UV and green emissions from ${}^4G_{11/2}$ and ${}^4S_{3/2}$ of Er^{3+} (${}^4G_{11/2}/{}^4S_{3/2}$) and between the emissions from the thermally coupled energy levels of ${}^4G_{11/2}/{}^2H_{9/2}$ and ${}^2H_{11/2}/{}^4S_{3/2}$ of Er^{3+} in $Rb_3InCl_6: 50\%Yb^{3+}/1\%Er^{3+}$ MCs as a function of excitation power density. The intensity ratios of ${}^4G_{11/2}/{}^4S_{3/2}$ ($I_{384/554}$), ${}^4G_{11/2}/{}^2H_{9/2}$ ($I_{384/409}$), and ${}^2H_{11/2}/{}^4S_{3/2}$ ($I_{524/554}$) of Er^{3+} increased from 0.441, 3.173, and 0.639 to 1.286, 6.077, and 2.095, respectively, with the increasing power density from 0.5 to 150 $W\text{ cm}^{-2}$. This indicates that the intensity ratios of $I_{384/554}$, $I_{384/409}$, and $I_{524/554}$ depend strongly on the excitation power density, which is not unexpected since the nonlinear UCL is power dependent. Specifically, the intensity of UV emission of Er^{3+} is comparable to that of the

green emission ($I_{384}/I_{554} = 0.471$) even under excitation with a low power density of 1 W cm^{-2} , wherein the laser-induced heating effect is negligible.

Supplementary Figure 15. Thermographs for the powder sample of $\text{Rb}_3\text{InCl}_6: 50\%\text{Yb}^{3+}/1\%\text{Er}^{3+}$ MCs upon 980-nm excitation at power densities of 1, 5, 10, 30, and 60 W cm^{-2} for 2 min, recorded by an infrared thermographic camera. Owing to the laser heating effect, the temperature of the sample increased from room temperature ($25.0 \text{ }^\circ\text{C}$) to 25.3, 26.1, 27.2, 30.2, and $36.3 \text{ }^\circ\text{C}$, respectively. Note that the UCL measurement was accompanied in 2 min.

Supplementary Figure 16. (a) Temperature-dependent UCL spectra of $\text{Rb}_3\text{InCl}_6: 50\%\text{Yb}^{3+}/1\%\text{Er}^{3+}$ MCs upon 980-nm excitation at a power density of 10 W cm^{-2} . (b) Intensity ratios between the UV and green emissions from ${}^4\text{G}_{11/2}$ and ${}^4\text{S}_{3/2}$ of Er^{3+} (${}^4\text{G}_{11/2}/{}^4\text{S}_{3/2}$) and between the emissions from the thermally coupled energy levels of ${}^4\text{G}_{11/2}/{}^2\text{H}_{9/2}$ and ${}^2\text{H}_{11/2}/{}^4\text{S}_{3/2}$ of Er^{3+} in $\text{Rb}_3\text{InCl}_6: 50\%\text{Yb}^{3+}/1\%\text{Er}^{3+}$ MCs as a function of temperature. The overall UCL intensity of the MCs decreased gradually with the temperature rise from 77 K to 377 K, as a result of accelerated nonradiative relaxation of Yb^{3+} and Er^{3+} at higher temperatures. Concurrently, the intensity ratios of ${}^4\text{G}_{11/2}/{}^4\text{S}_{3/2}$ (I_{384}/I_{554}), ${}^4\text{G}_{11/2}/{}^2\text{H}_{9/2}$ (I_{384}/I_{409}), and ${}^2\text{H}_{11/2}/{}^4\text{S}_{3/2}$ (I_{524}/I_{554}) of Er^{3+} increased from 0.408, 7.791 and 0.357 at 77 K to 0.718, 15.34 and 1.726 at 377 K, respectively, due to the thermal-enhanced population of Er^{3+} at higher states (${}^4\text{G}_{11/2}$ and ${}^2\text{H}_{11/2}$). Notably, the intensity ratio of ${}^4\text{G}_{11/2}/{}^4\text{S}_{3/2}$ (I_{384}/I_{554}) remained abnormally large (0.408) at 77 K, as compared to that of conventional UC materials.

Comment #5:

On p. 11, the authors use the term "density function theory" but probably mean density functional theory.

Response:

Thank you for your kind reminding. We have corrected the term “density function theory” as “density functional theory” on page 11 in the revised manuscript.

Comment #6:

In Fig. 3c, the authors discuss a diffusion-limited model for the luminescence decay with a fast decay component characterizing the energy migration process, while the slower decay represents the decay of the collective, non-interacting ensemble. While the general concept is correct, this is not clearly visible from the decay curve. Neither a specific model nor a residual plot is depicted that could justify the validity of this anticipated model. That should be clarified for the general reader.

Response:

Thank you very much for this valuable suggestion. To justify the validity of the anticipated model, we have added the residual plots of the fitting in Fig. 3c in the revised manuscript and Supplementary Figs. 24 and 25 in the revised Supplementary Information. The goodness-of-fit parameters R^2 are higher than 0.99, confirming the reliability of the fitting.

Particularly, for better clarity to the general reader, we have added more text discussion about the EM model. According to G. Blasse *et al.* (*J. Chem. Phys.* **1981**, 75, 561-571), there are three modes of ET between Ln^{3+} ions when the back ET is negligible, namely, direct ET without diffusion (or migration), diffusion-limited, and fast-diffusion modes, which can be distinguished from the shapes of the decay curves of their excited states (Supplementary Fig. 23). In the case of direct ET without diffusion ($P_{\text{DD}} = 0$), the decay curve is characterized by an initially non-exponential portion which reflects the transfer to acceptors located at various distances from the donors, followed by an exponential portion with a decay rate equal to the radiative rate. In the case of fast diffusion, the rate of ET among donors (namely, EM) is much larger than that of ET to acceptors ($P_{\text{DD}} > P_{\text{DA}}$), and the decay curve is exponential because the rapid migration has the effect of averaging the environments of the donors. In the case of diffusion-limited migration, the rate of ET among donors is lower than that from donors to acceptors ($P_{\text{DD}} < P_{\text{DA}}$). In this situation, the decay curve of the donors is characterized by an initially fast non-exponential portion followed by a slow exponential portion with the decay time (τ) shorter than the radiative decay time (τ_0) due to the influence of EM. The boundary between the non-exponential and exponential decay portions can be marked by a characteristic time t^* .

Based on the above definitions, it can be deduced that the decay of $^2F_{5/2}$ of Yb^{3+} in $\text{Rb}_3\text{InCl}_6: x\%\text{Yb}^{3+}$ MCs obeyed the diffusion-limited migration mode (Fig. 3c). According to M. Yokota and O. Tanimoto (*J. Phys. Soc. Jpn.* **1967**, 22, 779-784), the donor decay function for diffusion-limited migration in a three-dimensional sublattice can be expressed as:

$$I(t) = I_0 \exp \left[-\frac{t}{\tau_0} - \frac{4}{3} \pi^{\frac{3}{2}} C_A (Ct)^{\frac{1}{2}} \left(\frac{1 + 10.87x + 15.50x^2}{1 + 8.743x} \right)^{\frac{3}{4}} \right] \quad (1)$$

where $x = DC^{-1/3}t^{2/3}$, D is the diffusion constant, C is the interaction parameter for donor-acceptor, and C_A is the concentration of acceptors. At early times with $t \ll t^*$ ($t^* = C^{1/2}/D^{2/3}$), the time is not sufficient for the excitation energy to diffuse among the donors before being transferred to nearby acceptors, therefore, energy diffusion (migration) is negligible. At $t \gg t^*$, Eq. (1) can be reduced to (*J. Phys. Chem. Solids* **1982**, *43*, 481-490):

$$I(t) = I_0 \exp \left(-\frac{t}{\tau_0} - K_D t \right) = I_0 \exp \left(-\frac{t}{\tau} \right) \quad (2)$$

where

$$K_D = 4\pi DC_A R_D \quad (3)$$

$$R_D = 0.91 \left(\frac{C}{D} \right)^{1/4} \quad (4)$$

$$\frac{1}{\tau} = \frac{1}{\tau_0} + K_D \quad (5)$$

Eq. (2) verifies that the PL decay of diffusion-limited migration becomes exponential in long times after pulse excitation, where the decay rate of the slow exponential portion is determined by the radiation and migration rate. The decay rate due to migration ($K_D = 1/\tau_D$) can thus be obtained from Eq. (5), where $1/\tau$ is the observed decay rate derived from the exponential portion of the decay curve and $1/\tau_0$ is the radiative rate obtained from the low-doping (1 mol% Yb^{3+}) sample.

Text discussion newly added in the main manuscript:

Lines 1-17 of page 12: “According to G. Blasse *et al.*,⁴¹ there are three modes of ET between Ln^{3+} ions when the back ET is negligible, namely, direct ET without diffusion, diffusion-limited and fast-diffusion modes, which can be distinguished from the shapes of the decay curves of their excited states (Supplementary Fig. 23). In the case of direct ET without diffusion ($P_{DD} = 0$), the decay curve is characterized by an initially non-exponential portion which reflects the transfer to acceptors located at various distances from the donors, followed by an exponential portion with a decay rate equal to the radiative rate. In the case of fast diffusion, the rate of ET among donors (namely, EM) is much larger than that of ET to acceptors ($P_{DD} > P_{DA}$), and the decay curve is exponential because the rapid migration has the effect of averaging the environments of the donors. In the case of diffusion-limited migration, the rate of ET among donors is lower than that from donors to acceptors ($P_{DD} < P_{DA}$). In this situation, the decay curve of the donors is characterized by an initially fast non-exponential portion followed by a slow exponential portion with the decay time (τ) shorter than the radiative decay time (τ_0) due to the influence of EM. The boundary between the non-exponential and exponential decay portions can be marked by a characteristic time t^* . Based on these definitions, it can be deduced that the decay of ${}^2F_{5/2}$ of Yb^{3+} in $\text{Rb}_3\text{InCl}_6: x\% \text{Yb}^{3+}$ MCs obeyed the diffusion-limited migration mode (Fig. 3c).”

Revised Fig. 3 in the main manuscript:

Fig. 3. Confined ET between Yb³⁺ and Er³⁺ in Rb₃InCl₆: Yb³⁺/Er³⁺. (a) Crystal structures of monoclinic Rb_3InCl_6 and hexagonal NaYF_4 viewed along the c and b axes, respectively, showing the shortest In-In and Y-Y distances. (b) Schematic of the confined ET and long-range EM among the sensitizers and activators with long and short interionic distances, respectively. (c) PL decay curve of Yb³⁺ ($\lambda_{\text{em}} = 994$ nm) in $\text{Rb}_3\text{InCl}_6: 50\% \text{Yb}^{3+}$ MCs, which can be fitted well with the diffusion-limited mode. PL decay curves of Yb³⁺ in (d) $\text{Rb}_3\text{InCl}_6: x\% \text{Yb}^{3+}$ and (e) $\text{NaYF}_4: x\% \text{Yb}^{3+}$ MCs with different Yb³⁺ concentrations by monitoring the Yb³⁺ emission at 994 nm. (f) EM rates among Yb³⁺ in $\text{Rb}_3\text{InCl}_6: x\% \text{Yb}^{3+}$ and $\text{NaYF}_4: x\% \text{Yb}^{3+}$ MCs with different Yb³⁺ concentrations.

Text discussion and experimental data newly added in the Supplementary Information:

Supplementary Figure 23. Representative PL decay curves of the donor within the framework of energy diffusion modes: direct ET without diffusion ($P_{DD} = 0$), diffusion-limited ($P_{DD} < P_{DA}$), and fast-diffusion ($P_{DD} > P_{DA}$). The boundary between the non-exponential and exponential decay portions in the diffusion-limited mode can be marked by a characteristic time t^* . According to M. Yokota and O. Tanimoto,¹ the donor decay function for diffusion-limited migration in a three-dimensional sublattice can be expressed as:

$$I(t) = I_0 \exp \left[-\frac{t}{\tau_0} - \frac{4}{3} \pi^2 C_A (Ct)^{\frac{1}{2}} \left(\frac{1 + 10.87x + 15.50x^2}{1 + 8.743x} \right)^{3/4} \right] \quad (1)$$

where $x = DC^{-1/3}t^{2/3}$, D is the diffusion constant, C is the interaction parameter for donor-acceptor, and C_A is the concentration of acceptors. At early times with $t \ll t^*$ ($t^* = C^{1/2}/D^{2/3}$), the time is not sufficient for the excitation energy to diffuse among the donors before being transferred to nearby acceptors, therefore, energy diffusion (migration) is negligible. At $t \gg t^*$, Eq. (1) can be reduced to:

$$I(t) = I_0 \exp \left(-\frac{t}{\tau_0} - K_D t \right) = I_0 \exp \left(-\frac{t}{\tau} \right) \quad (2)$$

where

$$K_D = 4\pi DC_A R_D \quad (3)$$

$$R_D = 0.91 \left(\frac{C}{D} \right)^{1/4} \quad (4)$$

$$\frac{1}{\tau} = \frac{1}{\tau_0} + K_D \quad (5)$$

Eq. (2) verifies that the PL decay of diffusion-limited migration becomes exponential in long times after pulse excitation, where the decay rate of the slow exponential portion is determined by the radiation and migration rate. The decay rate due to migration ($K_D = 1/\tau_D$) can thus be obtained from Eq. (5), where $1/\tau$ is the observed decay rate derived from the exponential portion of the decay curve and $1/\tau_0$ is the radiative rate obtained from the low-doping (1 mol% Yb^{3+}) sample.

中国科学院福建物质结构研究所

Fujian Institute of Research on the Structure of Matter, Chinese Academy of Sciences

Supplementary Figure 24. PL decay curves of Yb^{3+} in $\text{Rb}_3\text{InCl}_6: x\%\text{Yb}^{3+}$ MCs with different Yb^{3+} concentrations by monitoring the Yb^{3+} emission at 994 nm upon pulsed laser excitation at 930 nm. The observed decay times (τ) of Yb^{3+} and the decay rates due to migration ($1/\tau_D$) were derived based on the diffusion-limited mode by single-exponential fitting to the exponential portion of the decay curves. The boundary between the non-exponential and exponential decay portions is marked by the characteristic time t^* . The goodness-of-fit parameters R^2 are higher than 0.99, demonstrating the reliability of the fitting.

Supplementary Figure 25. PL decay curves of Yb^{3+} in $\text{NaYF}_4: x\%\text{Yb}^{3+}$ MCs with different Yb^{3+} concentrations by monitoring the Yb^{3+} emission at 994 nm upon pulsed laser excitation at 930 nm. The observed decay times (τ) of Yb^{3+} and the decay rates due to migration ($1/\tau_D$) were derived based on the diffusion-limited mode by single-exponential fitting to the exponential portion of the decay curves. The boundary between the non-exponential and exponential decay portions is marked by the characteristic time t^* . The goodness-of-fit parameters R^2 are higher than 0.99, demonstrating the reliability of the fitting.

Reply to the Comments of Reviewer #2

General Comment:

This manuscript describes a relevant finding in the field of luminescence up-conversion materials, such as the observation of an unprecedentedly efficient conversion of near infrared (NIR, $\lambda=980$ nm) to near ultraviolet (UV, $\lambda=384$ nm) light. To the best of my knowledge, there are no precedents of a result like this in the related literature. This achievement is possible by co-doping with Yb^{3+} and Er^{3+} cations a 0D dimensional network of Rb_3InCl_6 , rationally chosen as a host matrix due to its low phonon energies, which in turn allows foreseeing a mitigated non-radiative multiphonon relaxation. The effect is adequately quantified by comparing the UV up-conversion process described to others well studied, like the NIR-to-green light conversion both in the targeted matrix and others commonly employed in the field (such as NaYF_4 , for instance). The authors also carefully addressed the analysis of the potential mechanisms involved in the observed effect, identifying the much longer lifetime of the $^4\text{G}_{11/2}$ state and, most interestingly, the much larger Yb^{3+} interionic distance (both compared to the values reported for other standard matrices used in the field of up-conversion) as the key factors responsible for its magnitude. The discussion on the inhibition of long range energy migration processes in the proposed 0-dimensional matrix, as opposed to what has been previously observed in other hosts, is particularly clarifying of the surprising absence of quenching of the UV emission even at very high Yb^{3+} concentrations (50%). In what follows, the authors will find some specific questions or comments that should be addressed in a revised version of the manuscript:

Response:

We greatly appreciate the reviewer for his/her positive comments on our manuscript and the efforts to improve the quality of our manuscript. We have carefully looked into all the helpful suggestions by the reviewer and made all the requested changes that have been reflected either in the text or in the Supplementary Information. A point-by-point response is noted below.

Comment #1:

I agree with the authors when they expect the reported effect to have an impact in the field of biomedicine, as a significant conversion of light within the near infrared transparency window of biological tissue into potentially therapeutic higher energy radiation in targeted regions might be of great interest for certain tumor treatments. In this context, my only remark would be that the application of this effect to the anion exchange between lead halide perovskite nanocrystals, given at the end of the manuscript as a kind of bonus, may be better suited to a separate analysis and publication, as it is too specific and lacks the general relevance of the rest of the information provided in the paper.

Response:

Thank you for pointing out this important issue. Our motivation of using Rb_3InCl_6 : $\text{Yb}^{3+}/\text{Er}^{3+}$ NCs for NIR-triggered anion exchange of CsPbX_3 PeNCs is to demonstrate the potential of the $\text{Yb}^{3+}/\text{Er}^{3+}$ couple for NIR-to-UV utilization in areas such as photochemical reaction, which had not been achieved before.

It is closely related to the main finding of our manuscript that $\text{Rb}_3\text{InCl}_6: \text{Yb}^{3+}/\text{Er}^{3+}$ exhibit unusually intense upconverted UV emission of Er^{3+} at 384 nm, which matches well with the absorption of CsPbX_3 PeNCs. The strategy of NIR-triggered anion exchange of CsPbX_3 PeNCs provides significant advantages as compared to that using UV light directly, because it can not only avoid the photodamage of PeNCs upon prolonged UV exposure, but also offer a high degree of remote control by using the NIR laser as the excitation source.

To rationalize this motivation, we have supplemented additional control experiments by using conventional $\text{NaYF}_4: \text{Yb}^{3+}/\text{Er}^{3+}$ MCs for the NIR-triggered anion exchange of CsPbX_3 PeNCs in haloalkanes under otherwise identical conditions. Although $\text{NaYF}_4: \text{Yb}^{3+}/\text{Er}^{3+}$ MCs can also be applied for the NIR-triggered anion exchange of CsPbX_3 PeNCs, they are less efficient with a significantly reduced exchange rate, due to the much weaker UV-UCL as compared to that of $\text{Rb}_3\text{InCl}_6: \text{Yb}^{3+}/\text{Er}^{3+}$. The reaction times required for completing the anion exchange from CsPbCl_3 to CsPbBr_3 and from CsPbBr_3 to CsPbI_3 were 150 and 32 minutes, respectively, by using $\text{NaYF}_4: \text{Yb}^{3+}/\text{Er}^{3+}$ as the NIR-to-UV transducer (Supplementary Fig. 44), which are longer than those based on $\text{Rb}_3\text{InCl}_6: \text{Yb}^{3+}/\text{Er}^{3+}$ (110 and 22 minutes) (Figs. 4d and e). The PLQYs of PeNCs obtained via the NIR-triggered anion exchange using $\text{NaYF}_4: \text{Yb}^{3+}/\text{Er}^{3+}$ (50.8–95.5%) were somewhat lower than those using $\text{Rb}_3\text{InCl}_6: \text{Yb}^{3+}/\text{Er}^{3+}$ (52.3–98.1%), probably due to the prolonged NIR illumination, but were remarkably higher than those of PeNCs obtained via UV-triggered anion exchange (10.0–88.7%) (Supplementary Tables 15-20). See also text discussion newly added in lines 4-8 of page 19 in the revised manuscript, and experimental data newly added in Supplementary Tables 19 and 20 and Supplementary Fig. 44 in the revised Supplementary Information.

Text discussion newly added in the main manuscript:

Lines 4-8 of page 19: “Besides, although conventional UC materials such as $\text{NaYF}_4: \text{Yb}^{3+}/\text{Er}^{3+}$ may also be applied for the NIR-triggered anion exchange of CsPbX_3 PeNCs (Supplementary Fig. 44 and Supplementary Tables 19 and 20), they are less efficient with a significantly reduced reaction rate, due to the much weaker UV-UCL than that of $\text{Rb}_3\text{InCl}_6: \text{Yb}^{3+}/\text{Er}^{3+}$.”

Experimental data newly added in the Supplementary Information:

Supplementary Table 19. Time dependence of the PL emission peak (λ_{em}) and PLQY for PeNCs derived from the mixture of $\text{NaYF}_4: \text{Yb}^{3+}/\text{Er}^{3+}$ (NYF), DBM, and CsPbCl_3 (NYF/DBM/ CsPbCl_3) upon 980-nm irradiation at a power density of 60 W cm^{-2} .

Time (min)	λ_{em} (nm)	PLQY (%)
0	406	50.8
10	428	62.5

25	441	69.1
40	457	72.6
65	468	79.2
90	487	86.1
120	499	73.9
150	515	59.1

Supplementary Table 20. Time dependence of the PL emission peak (λ_{em}) and PLQY for PeNCs derived from the mixture of NaYF₄: Yb³⁺/Er³⁺, IDP, and CsPbBr₃ (NYF/IDP/CsPbBr₃) upon 980-nm irradiation at a power density of 60 W cm⁻².

Time (min)	λ_{em} (nm)	PLQY (%)
0	512	88.7
2	538	90.1
5	570	93.3
10	601	95.5
15	630	87.9
20	665	79.6
25	683	70.4
32	698	56.2

Supplementary Figure 44. Time-dependent PL emission spectra ($\lambda_{\text{ex}} = 365 \text{ nm}$) for (a) $\text{CsPbCl}_3 \rightarrow \text{CsPbBr}_3$ in NYF/DBM/ CsPbCl_3 and (c) $\text{CsPbBr}_3 \rightarrow \text{CsPbI}_3$ in NYF/IDP/ CsPbBr_3 upon 980-nm laser irradiation at a power density of 60 W/cm^2 . Time dependence of the PL emission peaks ($\lambda_{\text{ex}} = 365 \text{ nm}$) for (b) $\text{CsPbCl}_3 \rightarrow \text{CsPbBr}_3$ in NYF/DBM/ CsPbCl_3 and (d) $\text{CsPbBr}_3 \rightarrow \text{CsPbI}_3$ in NYF/IDP/ CsPbBr_3 upon 980-nm laser irradiation at a power density of 60 W/cm^2 . The insets in (b) and (d) show the corresponding PLQYs of PeNCs. The reaction times required for completing the anion exchange from CsPbCl_3 to CsPbBr_3 and from CsPbBr_3 to CsPbI_3 were 150 and 32 minutes, respectively, by using $\text{NaYF}_4: \text{Yb}^{3+}/\text{Er}^{3+}$ as the NIR-to-UV transducer, which are longer than those based on $\text{Rb}_3\text{InCl}_6: \text{Yb}^{3+}/\text{Er}^{3+}$ (110 and 22 minutes). The PLQYs of PeNCs obtained via the NIR-triggered anion exchange using $\text{NaYF}_4: \text{Yb}^{3+}/\text{Er}^{3+}$ (50.8–95.5%) were somewhat lower than those using $\text{Rb}_3\text{InCl}_6: \text{Yb}^{3+}/\text{Er}^{3+}$ (52.3–98.1%), probably due to the prolonged NIR illumination.

Comment #2:

0D materials may also refer to nanomaterials, typically spheres, with all dimensions below 100 nm. A sentence may be added in introduction to avoid any misunderstanding.

Response:

Thank you for your kind reminding. The term “0D” in our manuscript is used to define the crystal structure with isolated structural units at the molecular level rather than the spherical nanoparticles with small sizes. To avoid any misunderstanding, we have added a sentence to define the “0D structure” in the revised Introduction. See also text discussion newly added in line 1 from the bottom of page 3 and line 1 of page 4 in the revised manuscript.

Text discussion newly added in the main manuscript:

Line 1 from the bottom of page 3 and line 1 of page 4: “Note that the term “0D” herein is used to define the crystal structure with isolated structural units at the molecular level rather than the spherical nanoparticles with small sizes.”

Comment #3:

Knowing the stringent effect of doping content on UV UC property, the real concentrations of dopants in NCs should be measured from ICP-AES, just as the authors did with the MCs, to allow a more accurate comparison between MCs and NCs. In general I miss some characterizations of the NCs (SEM, EDS, XPS...)

Response:

Many thanks for this valuable suggestion. To compare the UCL properties between $\text{Rb}_3\text{InCl}_6: \text{Yb}^{3+}/\text{Er}^{3+}$ MCs and NCs, we have synthesized $\text{Rb}_3\text{InCl}_6: x\%\text{Yb}^{3+}/1\%\text{Er}^{3+}$ ($x = 10, 30, 50$) and $\text{Rb}_3\text{InCl}_6: 50\%\text{Yb}^{3+}/y\%\text{Er}^{3+}$ ($y = 1, 4, 7$) NCs with different Yb^{3+} and Er^{3+} concentrations. The doping concentrations of Yb^{3+} and Er^{3+} and the structures and morphologies of $\text{Rb}_3\text{InCl}_6: \text{Yb}^{3+}/\text{Er}^{3+}$ NCs with different Yb^{3+} and Er^{3+} concentrations were characterized by Inductively coupled plasma-atomic emission spectroscopy (ICP-AES), XRD, transmission electron microscopy (TEM), selected area electron diffraction (SAED), energy dispersive X-ray spectra (EDS), X-ray photoelectron spectroscopy (XPS), and Raman spectra.

ICP-AES identified the actual doping concentrations of Yb^{3+} and Er^{3+} ranging from 2.5 and 0.8 mol% to 43.2 and 5.3 mol%, respectively (Supplementary Tables 11 and 12), slightly lower than their feeding concentrations. Structure and morphology characterizations through XRD, TEM, SAED, EDS, and XPS confirmed the pure phase and high crystallinity of the as-synthesized $\text{Rb}_3\text{InCl}_6: \text{Yb}^{3+}/\text{Er}^{3+}$ NCs with particle sizes in the range of 48.8–52.8 nm (Supplementary Figs. 27 and 28). The doping concentrations of Yb^{3+} and Er^{3+} had no significant influence on the size and morphology of the resulting NCs. Raman spectra of the NCs displayed characteristic vibration peaks of Rb_3InCl_6 with a cutoff phonon energy of 282 cm^{-1} (Supplementary Fig. 29), which is in good agreement with that of the MCs (Fig. 1j). See also text discussion newly added in lines 13-18 of page 14 in the revised manuscript, and experimental data newly added in Supplementary Tables 11 and 12 and Supplementary Figs. 27-29 in the revised Supplementary Information.

Text discussion newly added in the main manuscript:

Lines 13-18 of page 14: “Furthermore, we synthesized $\text{Rb}_3\text{InCl}_6: \text{Yb}^{3+}/\text{Er}^{3+}$ NCs with different Yb^{3+} and Er^{3+} concentrations to investigate the effect of particle size on the upconverted UV emission of Er^{3+} . Structural and compositional analyses through XRD, TEM, EDS, ICP-AES, XPS, and Raman spectra confirmed the pure phase and high crystallinity of the resulting $\text{Rb}_3\text{InCl}_6: \text{Yb}^{3+}/\text{Er}^{3+}$ NCs with particle

sizes in the range of 48.8–52.8 nm (Supplementary Figs. 27-29 and Supplementary Tables 11 and 12).”

Experimental data newly added in the Supplementary Information:

Supplementary Table 11. Nominal and actual doping concentrations of Yb^{3+} in $\text{Rb}_3\text{InCl}_6: x\%\text{Yb}^{3+}/1\%\text{Er}^{3+}$ nanocrystals (NCs). The nominal doping concentration of Yb^{3+} was defined by the molar ratio of Yb to (Yb + In + Er) in the precursor solution, and the actual doping concentrations of Yb^{3+} were identified by ICP-AES.

Sample	In : Yb : Er	Nominal	Actual
		Yb^{3+} / mol%	Yb^{3+} / mol%
1	89 : 10 : 1	10	2.5
2	69 : 30 : 1	30	18.3
3	49 : 50 : 1	50	43.2

Supplementary Table 12. Nominal and actual doping concentrations of Er^{3+} in $\text{Rb}_3\text{InCl}_6: 50\%\text{Yb}^{3+}/y\%\text{Er}^{3+}$ NCs. The nominal doping concentration of Er^{3+} was defined by the molar ratio of Er to (Yb + In + Er) in the precursor solution, and the actual doping concentrations of Er^{3+} were identified by ICP-AES.

Sample	In : Yb : Er	Nominal	Actual
		Er^{3+} / mol%	Er^{3+} / mol%
1	49 : 50 : 1	1	0.8
2	46 : 50 : 4	4	3.6
3	43 : 50 : 7	7	5.3

Supplementary Figure 27. (a) XRD patterns of Rb₃InCl₆: x%Yb³⁺/y%Er³⁺ NCs with different Yb³⁺ and Er³⁺ concentrations. The bottom lines represent the standard XRD pattern of monoclinic Rb₃InCl₆ (CCDC No. 2018909). (b) Energy dispersive X-ray spectra of Rb₃InCl₆: 50%Yb³⁺/1%Er³⁺ NCs, showing the elements of Rb, In, Cl, Yb, and Er in the NCs. Transmission electron microscopy (TEM) images and size distribution histograms of (c,d) Rb₃InCl₆: 10%Yb³⁺/1%Er³⁺, (e,f) Rb₃InCl₆: 30%Yb³⁺/1%Er³⁺, and (g,h) Rb₃InCl₆: 50%Yb³⁺/1%Er³⁺ NCs. The size distributions of the NCs were obtained by randomly calculating 200 particles in the TEM image. The inset in (g) shows the selected area electron diffraction (SAED) pattern of the NCs. All diffraction peaks of the NCs can be well indexed into monoclinic Rb₃InCl₆, indicating high crystallinity and phase purity of the as-synthesized Rb₃InCl₆: x%Yb³⁺/y%Er³⁺ NCs. The diffraction peaks of the NCs shifted towards lower angles with the increasing Yb³⁺ and Er³⁺ concentrations, as a result of lattice expansion induced by the substitution of In³⁺ (r = 0.81 Å, CN = 6) by larger Yb³⁺ (r = 0.86 Å, CN = 6) and Er³⁺ (r = 0.88 Å, CN = 6). TEM images showed that the NCs were roughly monodispersed with mean sizes in the range of 48.8–52.8 nm. The doping concentrations of Yb³⁺ and Er³⁺ had no significant influence on the size and morphology of the resulting NCs.

Supplementary Figure 28. XPS spectra of $\text{Rb}_3\text{InCl}_6: 10\%\text{Yb}^{3+}/1\%\text{Er}^{3+}$, $\text{Rb}_3\text{InCl}_6: 50\%\text{Yb}^{3+}/1\%\text{Er}^{3+}$, and $\text{Rb}_3\text{InCl}_6: 50\%\text{Yb}^{3+}/7\%\text{Er}^{3+}$ NCs. The spectra are shown over the energy regions typical for (a) Rb 3d, (b) In 3d, (c) Yb 4d, (d) Er 4d, and (e) Cl 2p peaks. The bimodal peaks at the binding energies of 109.7 and 111.1 eV, 445.0 and 452.3 eV, 184.7 and 192.2 eV, 169.1 eV and 170.4 eV, and 198.3 and 199.9 eV can be assigned to the $\text{Rb}^+ 3d_{5/2}$ and $3d_{3/2}$, $\text{In}^{3+} 3d_{5/2}$ and $3d_{3/2}$, $\text{Yb}^{3+} 4d_{5/2}$ and $4d_{3/2}$, $\text{Er}^{3+} 4d_{5/2}$ and $4d_{3/2}$, and $\text{Cl}^- 2p_{1/2}$ and $2p_{3/2}$, respectively. The $\text{Rb}^+ 3d$, $\text{Cl}^- 2p$, and $\text{In}^{3+} 3d$ peaks of the NCs shifted towards lower energies with the increasing Yb^{3+} or Er^{3+} concentration, ascribing to the strengthening of electron densities around Rb^+ , In^{3+} , and Cl^- . This implies that Yb^{3+} and Er^{3+} ions replace the octahedral In^{3+} site in the Rb_3InCl_6 lattice.

Supplementary Figure 29. Raman spectra of $\text{Rb}_3\text{InCl}_6: 10\%\text{Yb}^{3+}/1\%\text{Er}^{3+}$, $\text{Rb}_3\text{InCl}_6: 50\%\text{Yb}^{3+}/1\%\text{Er}^{3+}$, and $\text{Rb}_3\text{InCl}_6: 50\%\text{Yb}^{3+}/7\%\text{Er}^{3+}$ NCs. The vibrational peaks at 142, 169, and 282 cm^{-1} can be ascribed to the t_{2g} , e_g , and a_{1g} vibrational modes of the $[\text{InCl}_6]^{3-}$ octahedra, respectively. The Raman peaks of the NCs were broadened in comparison with those of the MCs, due to the influence of surface ligands and size effect of the NCs.

Comment #4:

Authors give a complete analysis of UV UC lifetime measurements to explain the abnormal high efficiency of UV UC in their material. For instance, they observe that increasing Er^{3+} content decreases the $^4\text{G}_{11/2}$ UCL lifetime as a result of back ET (= bad thing) while increasing Yb^{3+} content decreases the $^4\text{G}_{11/2}$ UCL lifetime as a result of improved ETU (= good thing). As the UV UC efficiency depends on several competing processes (EM, ET, back ET...), I miss a general conclusion as a rule of thumb here on the UV UC lifetime. Can the obtained results pave the way on how to choose the best UV UC phosphor based on its UV UC lifetime?

Response:

Yes, they can. Generally, a long lifetime of $^4\text{G}_{11/2}$ of Er^{3+} indicates a small probability of nonradiative relaxation from this level, which is conducive to the upconverted UV emission ($^4\text{G}_{11/2} \rightarrow ^4\text{I}_{15/2}$) of Er^{3+} at ~ 384 nm. Owing to the small energy gap between $^4\text{G}_{11/2}$ and $^2\text{H}_{9/2}$ (~ 1700 cm^{-1}) of Er^{3+} , the $^4\text{G}_{11/2}$ level is normally short-lived (< 50 μs) in conventional UC materials such as $\text{NaYF}_4: \text{Yb}^{3+}/\text{Er}^{3+}$ and $\text{Gd}_2\text{O}_3: \text{Yb}^{3+}/\text{Er}^{3+}$ with relatively large phonon energies (> 500 cm^{-1}), because of the fast multiphonon relaxation from $^4\text{G}_{11/2}$ to $^2\text{H}_{9/2}$ (*J. Phys. Chem. C* **2007**, *111*, 683-687; *Adv. Funct. Mater.* **2010**, *20*, 624-634). In this sense, a low phonon energy of the host (< 340 cm^{-1}) is the prerequisite for realizing intense UV-UCL from $^4\text{G}_{11/2}$ of Er^{3+} . This does not mean that any materials with low cutoff photon energies are appropriate for $\text{Yb}^{3+}/\text{Er}^{3+}$ doping to produce efficient UV-UCL of Er^{3+} . The inhibition of EM among Yb^{3+} to the lattice and surface defects and back ET from Er^{3+} to Yb^{3+} ($\text{Er}^{3+}: ^4\text{G}_{11/2} + \text{Yb}^{3+}: ^2\text{F}_{7/2} \rightarrow \text{Er}^{3+}: ^4\text{F}_{9/2} + \text{Yb}^{3+}: ^2\text{F}_{5/2}$), which facilitates the multiphoton UC processes from Yb^{3+} to Er^{3+} , plays a crucial role in the intense UV-UCL of Er^{3+} , as we demonstrated in $\text{Rb}_3\text{InCl}_6: \text{Yb}^{3+}/\text{Er}^{3+}$. Therefore, host materials with low cutoff photon energies and large interionic distances, which favor the multiphoton UC processes with a long lifetime of $^4\text{G}_{11/2}$ of Er^{3+} , are ideal candidates for $\text{Yb}^{3+}/\text{Er}^{3+}$ doping to achieve efficient UV-UCL of Er^{3+} under 980-nm excitation. See also text discussion newly added in lines 10-13 of page 15 in the revised manuscript.

Text discussion newly added in the main manuscript:

Lines 10-13 of page 15: “Our findings may pave the way for exploring efficient UV-UCL from the $\text{Yb}^{3+}/\text{Er}^{3+}$ couple based on the host materials with low photon energies and large interionic distances, which favor the multiphoton UC processes with a long lifetime of $^4\text{G}_{11/2}$ of Er^{3+} .”

Comment #5:

Results demonstrate that a close vicinity between Yb^{3+} - Er^{3+} is necessary while long Yb^{3+} - Yb^{3+} distances are favorable to decrease the EM. Can authors comment on the accuracy of the VASP calculations compared to real materials? Especially in the context of the 2 different synthesis methods used. Indeed MCs synthesis may be thermodynamically driven (12h at 180 °C and slow cooling 3 °C h⁻¹) while the synthesis of NCs may be kinetically driven (30 s at 200 °C with instant temperature quenching).

Response:

Yes, we have added some comments on the accuracy of the DFT calculations with VASP. Since the physical experiments, especially the chemical synthesis of the NCs, are intricate and multifaceted, the theoretical calculations represent only idealized outcomes. Nonetheless, they can provide a general picture of preferential distributions of the Yb^{3+} and Er^{3+} dopants in the Rb_3InCl_6 lattice. See also text discussion newly added in the caption of Supplementary Figure 26 in the revised Supplementary Information.

Text discussion newly added in the Supplementary Information:

Supplementary Figure 26. ... “Note that the theoretical calculations represent only idealized outcomes, because the physical experiments, especially the chemical synthesis of the NCs, are intricate and multifaceted. Nonetheless, they can provide a general picture of preferential distributions of the Yb^{3+} and Er^{3+} dopants in the Rb_3InCl_6 lattice.”

中国科学院福建物质结构研究所
Fujian Institute of Research on the Structure of Matter, Chinese Academy of Sciences

Reply to the Comments of Reviewer #3

General Comment:

Response:

We greatly appreciate the reviewer for his/her comments on our manuscript and the efforts to improve the quality of our manuscript.

Reply to the Comments of Reviewer #4

General Comment:

The authors display a work in which lanthanide ions are incorporated into an oxide perovskite matrix. An overall enhancement of the UV emission of Er is observed, which is interesting, as typically the green and red emission is stronger. Despite this being an interesting result, this work seems like just another article of lanthanide doping in an inert matrix. It is not very clear why and how this particular matrix was chosen and why it has this impact on the UV Er emission. The rationale behind it as well as the mechanism is not elaborated. I am doubtful about the results on the NIR triggered halide cation exchange. The term abnormal in the title is not scientific. It is a nice article, well done, but in my view not bringing sufficient new insight for the readership of Nature Commun.

Response:

We greatly appreciate the reviewer for his/her critical reading and helpful suggestions to improve the quality of our manuscript. We have carefully considered the reviewer's comments and revised our manuscript to address his/her concerns about the new insights of our manuscript. The significance of our work was not explicitly stressed out perhaps because of our unclear writing and data presentation in the original manuscript. In the revised manuscript, we have modified the text discussion and supplemented more experimental data, including the mechanism behind the unusually intense UV emission of Er^{3+} in Rb_3InCl_6 : $\text{Yb}^{3+}/\text{Er}^{3+}$, the validity and advantages of the NIR-triggered anion exchange of PeNCs in haloalkanes, and the reasons for the host selection, in an effort to strengthen the significance of our work. The following is the point-to-point response to the reviewer's comments.

Comment #1:

It is not clear which is the novelty of this specific approach of implementing Rb_3InCl_6 as a matrix for the lanthanide doping. It is not clear if the authors were the first to display lanthanide doping in perovskite halides, if so the question arises about the rationale behind this design. If they are not the first then the literature review needs to become more exhaustive.

Response:

Thank you for this valuable suggestion. It is the first report about lanthanide (Ln^{3+}) doping in 0D Rb_3InCl_6 , although other perovskite-derivative lead-free metal halides such as $\text{Cs}_2\text{NaInCl}_6$, $\text{Cs}_2\text{AgInCl}_6$, $\text{Cs}_2\text{AgBiBr}_6$, and Cs_2ZrCl_6 with Ln^{3+} doping have been widely reported before (*Angew. Chem. Int. Ed.* **2020**, *59*, 11307-11311; *Angew. Chem. Int. Ed.* **2022**, *61*, e202201993; *Angew. Chem. Int. Ed.* **2022**, *61*, e202201628; *Chem. Mater.* **2024**, *36*, 2857-2866). We chose Rb_3InCl_6 as the matrix for $\text{Yb}^{3+}/\text{Er}^{3+}$ doping based on the following considerations: (1) the low phonon energy of Rb_3InCl_6 (282 cm^{-1}) can effectively suppress the nonradiative multiphonon relaxation of Er^{3+} from $^4\text{G}_{11/2}$ to $^2\text{H}_{9/2}$ (energy gap: $\sim 1700\text{ cm}^{-1}$), which is a prerequisite for realizing intense UV upconversion luminescence (UCL) from $^4\text{G}_{11/2}$ ($\sim 384\text{ nm}$) of Er^{3+} ; and (2) the spatially confined 0D structure of Rb_3InCl_6 may mitigate the nonradiative energy losses associated with energy migration (EM) of Yb^{3+} , which is favorable for the multiphoton energy

transfer (ET) upconversion (UC) processes from Yb^{3+} to Er^{3+} . These features make 0D Rb_3InCl_6 appealing as an ideal candidate for $\text{Yb}^{3+}/\text{Er}^{3+}$ doping to achieve efficient UV-UCL of Er^{3+} , which, however, remains unexplored so far. For better clarity, we have added more text discussion about the reasons for the host material design. Some relevant papers about Ln^{3+} -doped metal halides have also been included as new references. See also text discussion newly added in lines 13-16 of page 3, lines 1-3 of page 4, and Refs. 18, 19, and 23-26 newly added in the revised manuscript.

Text discussion newly added in the main manuscript:

Lines 13-16 of page 3: “To unlock the UV-UCL of Er^{3+} , the search for new host materials with low phonon energies that can mitigate the nonradiative multiphonon relaxation (MPR) from $^4\text{G}_{11/2}$ to $^2\text{H}_{9/2}$ of Er^{3+} is of utmost importance.^{18, 19} In this regard, all-inorganic lead-free metal halides with cutoff phonon energies smaller than 300 cm^{-1} could be ideal candidates for this purpose.”

Lines 1-3 of page 4: “These features make 0D Rb_3InCl_6 appealing as a distinctive host material for Ln^{3+} doping to achieve desirable UCL properties, which, however, remains unexplored so far.”

References newly added in the main manuscript:

- 18 Zhou, X., Tanner, P. A. & Faucher, M. D. Electronic spectra and crystal field analysis of Er^{3+} in $\text{Cs}_2\text{NaErF}_6$. *J. Phys. Chem. C* **111**, 683-687 (2007).
- 19 Macedo, A. G. *et al.* Effects of phonon confinement on anomalous thermalization, energy transfer, and upconversion in Ln^{3+} -Doped Gd_2O_3 Nanotubes. *Adv. Funct. Mater.* **20**, 624-634 (2010).
- 23 Arfin, H., Kaur, J., Sheikh, T., Chakraborty, S. & Nag, A. Bi^{3+} - Er^{3+} and Bi^{3+} - Yb^{3+} codoped $\text{Cs}_2\text{AgInCl}_6$ double perovskite near infrared emitters. *Angew. Chem. Int. Ed.* **59**, 11307-11311 (2020).
- 24 Sun, J. Y. *et al.* Efficient near-infrared luminescence in lanthanide-doped vacancy-ordered double perovskite Cs_2ZrCl_6 phosphors via Te^{4+} sensitization. *Angew. Chem. Int. Ed.* **61**, e202201993 (2022).
- 25 Saikia, S. *et al.* Sb^{3+} - Er^{3+} -codoped $\text{Cs}_2\text{NaInCl}_6$ for emitting blue and short-wave infrared radiation. *Angew. Chem. Int. Ed.* **61**, e202201628 (2022).
- 26 de Wit, J. W., Sonneveld, L. L. & Meijerink, A. Shedding light on host-to- Yb^{3+} energy transfer in $\text{Cs}_2\text{AgBiBr}_6$: Yb^{3+} (nano)crystals. *Chem. Mater.* **36**, 2857-2866 (2024).

Comment #2:

The authors use the term abnormal which is not very scientific. What exactly does it mean. If something is abnormal then I expect something else to be normal. Which is the normal part that makes this PL abnormal?

Response:

Thank you for your kind reminding. We intended to use the term “abnormal” to describe the intense UV-UCL of Er^{3+} at 384 nm observed in Rb_3InCl_6 : $\text{Yb}^{3+}/\text{Er}^{3+}$, which is unusual in comparison with that of

conventional $\text{Yb}^{3+}/\text{Er}^{3+}$ co-doped UC materials. In conventional $\text{Yb}^{3+}/\text{Er}^{3+}$ co-doped UC materials, the UCL is dominated by the green (~ 540 nm) and red (~ 650 nm) emissions originating from the ${}^2\text{H}_{11/2}/{}^4\text{S}_{3/2} \rightarrow {}^4\text{I}_{15/2}$ and ${}^4\text{F}_{9/2} \rightarrow {}^4\text{I}_{15/2}$ transitions of Er^{3+} , respectively, while the UV-UCL (~ 380 nm) from the ${}^4\text{G}_{11/2} \rightarrow {}^4\text{I}_{15/2}$ transition of Er^{3+} is very weak and negligible, because of the dense energy levels of Er^{3+} that aggravate the nonradiative energy losses through cross relaxation between adjacent Er^{3+} , back ET from Er^{3+} to Yb^{3+} , and EM among Yb^{3+} to the surface and lattice defects. To avoid any misunderstanding, we have rephrased the term “abnormal” as “unusual” or “unusually strong” throughout the manuscript. Specifically, the title “Abnormal upconverted ultraviolet emission of Er^{3+} through confined energy transfer in $\text{Yb}^{3+}/\text{Er}^{3+}$ co-doped Rb_3InCl_6 ” has been revised as “Unusually intense upconverted ultraviolet emission of Er^{3+} through confined energy transfer in $\text{Yb}^{3+}/\text{Er}^{3+}$ co-doped Rb_3InCl_6 ”.

Comment #3:

I understand that the PL in the UV is increased with respect to other examples of Er doped nanocrystals. The authors try to give an explanation of the mechanism, but it is very briefly discussed only. The mechanistic insight in my view however is very important for a journal of this level and the authors would have needed to show this more in detail and with a more exhaustive discussion.

Response:

Thank you for pointing out this important issue. As per your suggestion, we have supplemented additional experimental data and added more text discussion about the mechanism underlying the unusually intense upconverted UV emission of Er^{3+} in $\text{Rb}_3\text{InCl}_6: \text{Yb}^{3+}/\text{Er}^{3+}$. The unusual upconverted UV emission of Er^{3+} , characterized by a large intensity ratio of ${}^4\text{G}_{11/2}/{}^4\text{S}_{3/2}$ ($I_{384/554}$), was observed not only in $\text{Rb}_3\text{InCl}_6: \text{Yb}^{3+}/\text{Er}^{3+}$ MCs but also in their NC counterparts (Supplementary Figs. 30 and 32). Specifically, the intensity ratio of ${}^4\text{G}_{11/2}/{}^4\text{S}_{3/2}$ ($I_{384/554}$) in $\text{Rb}_3\text{InCl}_6: \text{Yb}^{3+}/\text{Er}^{3+}$ MCs remains abnormally large (>0.4) as compared to that of conventional UC materials, even under excitation with a low power density of 1 W cm^{-2} and at a low temperature of 77 K , wherein the laser-induced heating effect is negligible (Supplementary Figs. 8, 15, and 16). These results demonstrate that the unusually intense upconverted UV emission of Er^{3+} is the intrinsic properties of $\text{Rb}_3\text{InCl}_6: \text{Yb}^{3+}/\text{Er}^{3+}$ and independent of the particle size and the laser-induced heating effect.

Mechanistic investigation through concentration- and power-dependent UCL spectroscopic analyses unraveled that the unusual upconverted UV emission of Er^{3+} is dictated by the spatially confined 0D structure of Rb_3InCl_6 with a large interionic distance ($>7.14 \text{ \AA}$) and low phonon energies ($<282 \text{ cm}^{-1}$), which facilitates the confined ET between Yb^{3+} and Er^{3+} within a short range while suppressing the energy losses through the nonradiative multiphonon relaxation and long-range EM processes. This promotes the population of Er^{3+} at the ${}^4\text{G}_{11/2}$ state, resulting in the intense UV emission at 384 nm with a large UV-to-green ratio (I_{384}/I_{554}). Such confined ET and suppressed EM processes in 0D $\text{Rb}_3\text{InCl}_6: \text{Yb}^{3+}/\text{Er}^{3+}$ can be validated by comparing the PL decays of $\text{Rb}_3\text{InCl}_6: x\% \text{Yb}^{3+}$ MCs with those of $\text{NaYF}_4: x\% \text{Yb}^{3+}$ MCs on the basis of the energy diffusion model, whereby the EM rate of Yb^{3+} in $\text{Rb}_3\text{InCl}_6: \text{Yb}^{3+}$ ($2.10 \times 10^2 \text{ s}^{-1}$ for $50\% \text{Yb}^{3+}$) was found to be about one order of magnitude slower than that in $\text{NaYF}_4: \text{Yb}^{3+}$ ($2.17 \times 10^3 \text{ s}^{-1}$ for $50\% \text{Yb}^{3+}$) (Fig. 3 and Supplementary Table 10).

Particularly, for better clarity to the general reader, we have added more text discussion about the EM model. According to G. Blasse *et al.* (*J. Chem. Phys.* **1981**, 75, 561-571), there are three modes of ET between Ln³⁺ ions when the back ET is negligible, namely, direct ET without diffusion (or migration), diffusion-limited, and fast-diffusion modes, which can be distinguished from the shapes of the decay curves of their excited states (Supplementary Fig. 23). In the case of direct ET without diffusion ($P_{DD} = 0$), the decay curve is characterized by an initially non-exponential portion which reflects the transfer to acceptors located at various distances from the donors, followed by an exponential portion with a decay rate equal to the radiative rate. In the case of fast diffusion, the rate of ET among donors (namely, EM) is much larger than that of ET to acceptors ($P_{DD} > P_{DA}$), and the decay curve is exponential because the rapid migration has the effect of averaging the environments of the donors. In the case of diffusion-limited migration, the rate of ET among donors is lower than that from donors to acceptors ($P_{DD} < P_{DA}$). In this situation, the decay curve of the donors is characterized by an initially fast non-exponential portion followed by a slow exponential portion with the decay time (τ) shorter than the radiative decay time (τ_0) due to the influence of EM. The boundary between the non-exponential and exponential decay portions can be marked by a characteristic time t^* .

Based on the above definitions, it can be deduced that the decay of ²F_{5/2} of Yb³⁺ in Rb₃InCl₆: x%Yb³⁺ MCs obeyed the diffusion-limited migration mode (Fig. 3c). According to M. Yokota and O. Tanimoto (*J. Phys. Soc. Jpn.* **1967**, 22, 779-784), the donor decay function for diffusion-limited migration in a three-dimensional sublattice can be expressed as:

$$I(t) = I_0 \exp \left[-\frac{t}{\tau_0} - \frac{4}{3} \pi^{\frac{3}{2}} C_A (Ct)^{\frac{1}{2}} \left(\frac{1 + 10.87x + 15.50x^2}{1 + 8.743x} \right)^{\frac{3}{4}} \right] \quad (1)$$

where $x = DC^{-1/3}t^{2/3}$, D is the diffusion constant, C is the interaction parameter for donor-acceptor, and C_A is the concentration of acceptors. At early times with $t \ll t^*$ ($t^* = C^{1/2}/D^{2/3}$), the time is not sufficient for the excitation energy to diffuse among the donors before being transferred to nearby acceptors, therefore, energy diffusion (migration) is negligible. At $t \gg t^*$, Eq. (1) can be reduced to (*J. Phys. Chem. Solids* **1982**, 43, 481-490):

$$I(t) = I_0 \exp \left(-\frac{t}{\tau_0} - K_D t \right) = I_0 \exp \left(-\frac{t}{\tau} \right) \quad (2)$$

where

$$K_D = 4\pi DC_A R_D \quad (3)$$

$$R_D = 0.91 \left(\frac{C}{D} \right)^{1/4} \quad (4)$$

$$\frac{1}{\tau} = \frac{1}{\tau_0} + K_D \quad (5)$$

Eq. (2) verifies that the PL decay of diffusion-limited migration becomes exponential in long times after pulse excitation, where the decay rate of the slow exponential portion is determined by the radiation and migration rate. The decay rate due to migration ($K_D = 1/\tau_D$) can thus be obtained from Eq. (5), where $1/\tau$ is the observed decay rate derived from the exponential portion of the decay curve and $1/\tau_0$ is the radiative rate obtained from the low-doping (1 mol% Yb³⁺) sample.

Text discussion newly added in the main manuscript:

Lines 1-17 of page 12: “According to G. Blasse *et al.*,⁴¹ there are three modes of ET between Ln^{3+} ions when the back ET is negligible, namely, direct ET without diffusion, diffusion-limited and fast-diffusion modes, which can be distinguished from the shapes of the decay curves of their excited states (Supplementary Fig. 23). In the case of direct ET without diffusion ($P_{\text{DD}} = 0$), the decay curve is characterized by an initially non-exponential portion which reflects the transfer to acceptors located at various distances from the donors, followed by an exponential portion with a decay rate equal to the radiative rate. In the case of fast diffusion, the rate of ET among donors (namely, EM) is much larger than that of ET to acceptors ($P_{\text{DD}} > P_{\text{DA}}$), and the decay curve is exponential because the rapid migration has the effect of averaging the environments of the donors. In the case of diffusion-limited migration, the rate of ET among donors is lower than that from donors to acceptors ($P_{\text{DD}} < P_{\text{DA}}$). In this situation, the decay curve of the donors is characterized by an initially fast non-exponential portion followed by a slow exponential portion with the decay time (τ) shorter than the radiative decay time (τ_0) due to the influence of EM. The boundary between the non-exponential and exponential decay portions can be marked by a characteristic time t^* . Based on these definitions, it can be deduced that the decay of $^2F_{5/2}$ of Yb^{3+} in $\text{Rb}_3\text{InCl}_6: x\%\text{Yb}^{3+}$ MCs obeyed the diffusion-limited migration mode (Fig. 3c).”

Text discussion and experimental data newly added in the Supplementary Information:

Supplementary Figure 23. Representative PL decay curves of the donor within the framework of energy diffusion modes: direct ET without diffusion ($P_{\text{DD}} = 0$), diffusion-limited ($P_{\text{DD}} < P_{\text{DA}}$), and fast-diffusion ($P_{\text{DD}} > P_{\text{DA}}$). The boundary between the non-exponential and exponential decay portions in the diffusion-limited mode can be marked by a characteristic time t^* . According to M. Yokota and O. Tanimoto,¹ the donor decay function for diffusion-limited migration in a three-dimensional sublattice can be expressed as:

中国科学院福建物质结构研究所

Fujian Institute of Research on the Structure of Matter, Chinese Academy of Sciences

$$I(t) = I_0 \exp \left[-\frac{t}{\tau_0} - \frac{4}{3} \pi^2 C_A (Ct)^{\frac{1}{2}} \left(\frac{1 + 10.87x + 15.50x^2}{1 + 8.743x} \right)^{3/4} \right] \quad (1)$$

where $x = DC^{-1/3}t^{2/3}$, D is the diffusion constant, C is the interaction parameter for donor-acceptor, and C_A is the concentration of acceptors. At early times with $t \ll t^*$ ($t^* = C^{1/2}/D^{2/3}$), the time is not sufficient for the excitation energy to diffuse among the donors before being transferred to nearby acceptors, therefore, energy diffusion (migration) is negligible. At $t \gg t^*$, Eq. (1) can be reduced to:

$$I(t) = I_0 \exp \left(-\frac{t}{\tau_0} - K_D t \right) = I_0 \exp \left(-\frac{t}{\tau} \right) \quad (2)$$

where

$$K_D = 4\pi DC_A R_D \quad (3)$$

$$R_D = 0.91 \left(\frac{C}{D} \right)^{1/4} \quad (4)$$

$$\frac{1}{\tau} = \frac{1}{\tau_0} + K_D \quad (5)$$

Eq. (2) verifies that the PL decay of diffusion-limited migration becomes exponential in long times after pulse excitation, where the decay rate of the slow exponential portion is determined by the radiation and migration rate. The decay rate due to migration ($K_D = 1/\tau_D$) can thus be obtained from Eq. (5), where $1/\tau$ is the observed decay rate derived from the exponential portion of the decay curve and $1/\tau_0$ is the radiative rate obtained from the low-doping (1 mol% Yb^{3+}) sample.

Comment #4:

The application of using NIR light to trigger ion exchange in perovskite nanocrystal is not really convincing. First of all: what is the advantage of UV light. It's way more accessible than IR. Did the authors compare the efficiency with respect to pure NIR light? What about multi photon processes or heat as a result of the NIR excitation of the halide perovskites as a reason for the enhanced ion exchange. The reference experiments of a solution of perovskite nanocrystals without up-converting nanocrystals but with the exposure to NIR light needs to be performed.

Response:

Many thanks for this valuable suggestion. As per your request, we have supplemented the control experiments by using 980-nm NIR laser as the excitation source for the anion exchange in the absence of $\text{Rb}_3\text{InCl}_6: \text{Yb}^{3+}/\text{Er}^{3+}$ (RIC) NCs. The emission peaks of PeNCs derived from the mixture of dibromomethane (DBM)/ CsPbCl_3 and 2-iodopropane (IDP)/ CsPbBr_3 without RIC remained essentially unchanged upon 980-nm excitation for 3 hours (Supplementary Fig. 39). This is in marked contrast to that of RIC/DBM/ CsPbCl_3 and RIC/IDP/ CsPbBr_3 , whereby the 980-nm laser irradiation induced a continuous red-shift of the emission peaks from 408 nm and 510 nm to 511 nm and 696 nm, respectively (Figs. 4d and e), underscoring the key role of NIR-to-UV UC processes in NIR-triggered anion exchange of CsPbX_3 PeNCs.

To assess the laser-induced heating effect on the anion exchange of CsPbX_3 PeNCs, we have recorded

the temperature of the RIC/IDP/CsPbBr₃ solution upon 980-nm excitation at power densities of 30, 60, and 100 W cm⁻² for 10 min. The temperature of the solution increased from room temperature (25.0 °C) to 29.2, 32.9, and 38.2 °C upon 980-nm laser excitation with power densities of 30, 60, and 100 W cm⁻² for 10 min, respectively (Supplementary Fig. 42). Such laser-induced heating effect may also contribute to the enhanced anion exchange rate at higher excitation power due to the increased motion of the anions with the temperature rise.

The strategy of NIR-triggered anion exchange of CsPbX₃ PeNCs provides significant advantages as compared to that using UV light directly, because it can not only avoid the photodamage of PeNCs upon prolonged UV exposure, but also offer a high degree of remote control by using the NIR laser as the excitation source. To this regard, we have supplemented additional control experiments by using a 365-nm UV-LED (3 W) instead of the 980-nm NIR laser as the excitation source for the photoinduced anion exchange of CsPbX₃ PeNCs in haloalkanes without RIC under otherwise identical conditions. The results showed that 365-nm UV-LED can trigger the anion exchange of CsPbX₃ PeNCs with an accelerated reaction rate as compared to that upon 980-nm NIR laser excitation, because of the higher power and larger area of UV light acting on the reaction system upon direct UV illumination. The reaction times required for completing the anion exchange from CsPbCl₃ to CsPbBr₃ and from CsPbBr₃ to CsPbI₃ were 17 and 8 minutes, respectively, under UV illumination (Supplementary Fig. 43), much shorter than those upon NIR irradiation (110 and 22 minutes). However, the PLQYs of PeNCs after UV-triggered anion exchange decreased significantly from 50.8% for CsPbCl₃ to 43.2% for CsPbBr₃ and from 88.7% for CsPbBr₃ to 10.0% for CsPbI₃ (Figs. 4f and g and Supplementary Tables 17 and 18), due to the serious photodamage of PeNCs especially CsPbI₃ upon prolonged UV exposure. Such UV-induced photodamage of PeNCs can be avoided by the strategy of NIR-triggered anion exchange, yielding significantly improved PLQYs in the range of 52.3–98.1% (Supplementary Tables 15 and 16). Additionally, the use of NIR laser enables the remote control of PeNCs with desired PL emissions by tuning the on/off state and the power density of laser as well as the irradiation time (Supplementary Figs. 40 and 41). These results demonstrate the remarkable advantages of our strategy of NIR-triggered anion exchange of CsPbX₃ PeNCs, which offers a new way for the post-synthesis modification of PeNCs towards various optoelectronic applications. See also text discussion newly added in lines 12-15 of page 18, lines 1-3 from the bottom of page 18 and lines 1-4 of page 19, and Fig. 4 in the revised manuscript, and experimental data newly added in Supplementary Tables 17 and 18 and Supplementary Figs. 39 and 43 in the revised Supplementary Information.

Revised Fig. 4 in the main manuscript:

Fig. 4. NIR-triggered anion exchange of CsPbX₃ PeNCs using Rb₃InCl₆: Yb³⁺/Er³⁺. (a) Schematic of NIR-triggered anion exchange of CsPbX₃ PeNCs in haloalkanes by using Rb₃InCl₆: Yb³⁺/Er³⁺ NCs as the NIR-to-UV transducer. (b) PL emission spectra ($\lambda_{\text{ex}} = 365 \text{ nm}$) and (c) PL decay curves of CsPbX₃ PeNCs with different halide compositions, derived from anion exchange based on the mixtures of RIC/DBM/CsPbCl₃ and RIC/IDP/CsPbBr₃ upon 980-nm NIR laser exposure. The insets in (b) show the PL photographs of the resulting PeNCs under 365-nm irradiation. Time dependence of the PL emission peaks ($\lambda_{\text{ex}} = 365 \text{ nm}$) for (d) CsPbCl₃ \rightarrow CsPbBr₃ in DBM/CsPbCl₃ and RIC/DBM/CsPbCl₃ and (e) CsPbBr₃ \rightarrow CsPbI₃ in IDP/CsPbBr₃ and RIC/IDP/CsPbBr₃ upon 980-nm irradiation at a power density of 60 W cm^{-2} . PLQYs for PeNCs derived from the photoinduced anion exchange (f) from CsPbCl₃ to CsPbBr₃ in DBM/CsPbCl₃ ($\lambda_{\text{ex}} = 365 \text{ nm}$) and RIC/DBM/CsPbCl₃ ($\lambda_{\text{ex}} = 980 \text{ nm}$) and (g) from CsPbBr₃ to CsPbI₃ in IDP/CsPbBr₃ ($\lambda_{\text{ex}} = 365 \text{ nm}$) and RIC/IDP/CsPbBr₃ ($\lambda_{\text{ex}} = 980 \text{ nm}$), respectively.

Text discussion newly added in the main manuscript:

Lines 12-15 of page 18: “By contrast, the emission peaks of PeNCs derived from DBM/CsPbCl₃ and IDP/CsPbBr₃ without RIC remained essentially unchanged upon 980-nm excitation for 3 hours (Supplementary Fig. 39), underscoring the key role of NIR-to-UV UC processes in NIR-triggered anion exchange of CsPbX₃ PeNCs.”

Lines 1-3 from the bottom of page 18 and lines 1-4 of page 19: “For comparison, we also performed UV-triggered anion exchange of CsPbX₃ PeNCs in haloalkanes without RIC NCs by using a 365-nm light-

emitting diode (LED) as the excitation source under otherwise identical conditions. The results showed that 365-nm UV-LED can trigger the anion exchange of CsPbX₃ PeNCs with an accelerated reaction rate (Supplementary Fig. 43), but it is detrimental to the PLQYs (decreased to 10.0% for CsPbI₃) due to the serious photodamage of PeNCs upon direct UV exposure (Figs. 4f and g and Supplementary Tables 17 and 18).”

Experimental data newly added in the Supplementary Information:

Supplementary Table 17. Time dependence of the PL emission peak (λ_{em}) and PLQY for PeNCs derived from the mixture of DBM/CsPbCl₃ upon 365-nm UV-LED (3 W) irradiation.

Time (min)	λ_{em} (nm)	PLQY (%)
0	406	50.8
1	430	60.3
2	446	68.0
3	456	74.7
5	468	79.2
8	485	70.1
12	497	58.9
17	510	43.2

Supplementary Table 18. Time dependence of the PL emission peak (λ_{em}) and PLQY for PeNCs derived from the mixture of IDP/CsPbBr₃ upon 365-nm UV-LED (3 W) irradiation.

Time (min)	λ_{em} (nm)	PLQY (%)
0	512	88.7
1	541	80.1
2	575	71.3
3	608	59.5
5	639	45.8

6	670	33.9
7	680	22.1
8	695	10.0

Supplementary Figure 39. PL emission spectra ($\lambda_{\text{ex}} = 365 \text{ nm}$) of the mixtures of (a) DBM/CsPbCl₃ and (b) IDP/CsPbBr₃ before and after irradiation with the 980-nm laser at a power density of 60 W cm^{-2} for 3 hours.

Supplementary Figure 43. Time-dependent PL emission spectra ($\lambda_{\text{ex}} = 365 \text{ nm}$) for (a) CsPbCl₃ \rightarrow CsPbBr₃ in DBM/CsPbCl₃ and (c) CsPbBr₃ \rightarrow CsPbI₃ in IDP/CsPbBr₃ upon 365-nm UV-LED (3 W) irradiation. Time dependence of the PL emission peaks ($\lambda_{\text{ex}} = 365 \text{ nm}$) for (b) CsPbCl₃ \rightarrow CsPbBr₃ in

DBM/CsPbCl₃ and (d) CsPbBr₃ → CsPbI₃ in IDP/CsPbBr₃ upon 365-nm UV-LED (3 W) irradiation. The insets in (b) and (d) show the corresponding PLQYs of PeNCs. The reaction times required for completing the anion exchange from CsPbCl₃ to CsPbBr₃ and from CsPbBr₃ to CsPbI₃ were 17 and 8 minutes, respectively, under UV illumination, much shorter than those upon NIR irradiation using Rb₃InCl₆: Yb³⁺/Er³⁺ (110 and 22 minutes). However, the PLQYs of PeNCs after UV-triggered anion exchange decreased significantly from 50.8% for CsPbCl₃ to 43.2% for CsPbBr₃ and from 88.7% for CsPbBr₃ to 10.0% for CsPbI₃, due to the serious photodamage of PeNCs especially CsPbI₃ upon prolonged UV exposure.

Comment #5:

What is the difference between the micro and the nano-crystals. The authors seem to switch between one or the other depending on the experiment they are performing. But is this valid? What is the difference between them, how does their size impact the specific property they are looking at? More insights into this must be given.

Response:

Many thanks for this valuable suggestion. We have supplemented additional experiments and reorganized the part about Rb₃InCl₆: Yb³⁺/Er³⁺ NCs. To investigate the effect of particle size on the upconverted UV emission of Er³⁺, we have synthesized Rb₃InCl₆: x%Yb³⁺/1%Er³⁺ (x = 10, 30, 50) and Rb₃InCl₆: 50%Yb³⁺/y%Er³⁺ (y = 1, 4, 7) NCs with different Yb³⁺ and Er³⁺ concentrations. ICP-AES identified the actual doping concentrations of Yb³⁺ and Er³⁺ ranging from 2.5 and 0.8 mol% to 43.2 and 5.3 mol%, respectively (Supplementary Tables 11 and 12), slightly lower than their feeding concentrations. Structure and morphology characterizations through XRD, TEM, SAED, EDS, and XPS confirmed the pure phase and high crystallinity of the as-synthesized Rb₃InCl₆: Yb³⁺/Er³⁺ NCs with particle sizes in the range of 48.8–52.8 nm (Supplementary Figs. 27 and 28). The doping concentrations of Yb³⁺ and Er³⁺ had no significant influence on the size and morphology of the resulting NCs. Raman spectra of the NCs displayed characteristic vibration peaks of Rb₃InCl₆ with a cutoff phonon energy of 282 cm⁻¹ (Supplementary Fig. 29), which is in line with that of the MCs (Fig. 1j).

Similar to their MC counterparts, Rb₃InCl₆: Yb³⁺/Er³⁺ NCs displayed unusually strong UV emission of Er³⁺ at 384 nm (⁴G_{11/2} → ⁴I_{15/2}), with the UV-to-green ratio (I₃₈₄/I₅₅₄) of up to 0.702 for Rb₃InCl₆: 50%Yb³⁺/1%Er³⁺ NCs, upon 980-nm excitation at a power density of 60 W/cm⁻² (Supplementary Fig. 30). The corresponding UCQYs for the UV emission (~384 nm) and overall emissions (360–720 nm) of Er³⁺ in Rb₃InCl₆: 50%Yb³⁺/1%Er³⁺ NCs were determined to be 0.08% and 0.37%, respectively, slightly lower than those of their MC counterparts (0.12% and 0.42%). It is noteworthy that the red emission of Er³⁺ at 659 nm (⁴F_{9/2} → ⁴I_{15/2}) was enhanced in NCs as compared to that in MCs, due to the influence of high-energy vibrational groups such as OH⁻ on the surface of the NCs, which facilitated the population of Er³⁺ at the ⁴F_{9/2} level via nonradiative relaxation from higher energy levels (e.g., ²H_{11/2} and ⁴S_{3/2}) (*ACS Nano* **2018**, *12*, 4812-4823; *Light-Sci. Appl.* **2021**, *10*, 105).

We have also investigated the effects of the Yb^{3+} and Er^{3+} concentrations on the UCL properties of $\text{Rb}_3\text{InCl}_6: \text{Yb}^{3+}/\text{Er}^{3+}$ NCs. As shown in Supplementary Fig. 30, the integrated UCL intensity of the NCs increased steadily with the increasing Yb^{3+} concentration from 10 mol% to 50 mol%, indicating the absence of concentration quenching effect of Yb^{3+} in $\text{Rb}_3\text{InCl}_6: x\%\text{Yb}^{3+}/1\%\text{Er}^{3+}$ NCs. Specifically, the intensity ratio of UV-to-green (I_{384}/I_{554}) of Er^{3+} was remarkably enhanced from 0.175 (10 mol% of Yb^{3+}) to 0.702 (50 mol% of Yb^{3+}). Correspondingly, the UCL lifetimes of Er^{3+} exhibited a gradual decrease upon increasing the Yb^{3+} concentration (Supplementary Fig. 31 and Supplementary Table 13), as a result of improved ETU processes that accelerated the exhaustion of the excitation energy from Yb^{3+} and consequently the depopulation of Er^{3+} from the emitting levels (*Chem. Soc. Rev.* **2015**, *44*, 1608-1634; *J. Phys. Chem. Lett.* **2020**, *11*, 3672-3680). Further increasing the Er^{3+} concentration on the basis of $\text{Rb}_3\text{InCl}_6: 50\%\text{Yb}^{3+}/1\%\text{Er}^{3+}$ NCs shortened the interionic distance between Yb^{3+} and Er^{3+} , resulting in the enhanced back ET from Er^{3+} to Yb^{3+} ($\text{Er}^{3+}: ^4\text{G}_{11/2} + \text{Yb}^{3+}: ^2\text{F}_{7/2} \rightarrow \text{Er}^{3+}: ^4\text{F}_{9/2} + \text{Yb}^{3+}: ^2\text{F}_{5/2}$) and the decreased I_{384}/I_{554} ratio of Er^{3+} with reduced UCL lifetime of $^4\text{G}_{11/2}$ (Supplementary Figs. 32 and 33 and Supplementary Table 14). These results are generally consistent with those observed in $\text{Rb}_3\text{InCl}_6: \text{Yb}^{3+}/\text{Er}^{3+}$ MCs, unraveling that the unusually intense upconverted UV emission of Er^{3+} is the intrinsic properties of $\text{Rb}_3\text{InCl}_6: \text{Yb}^{3+}/\text{Er}^{3+}$ and independent of the particle size. See also text discussion newly added in lines 13-23 of page 14 and lines 1-10 of page 15 in the revised manuscript, and experimental data newly added in Supplementary Tables 11-14 and Supplementary Figs. 27-33 in the revised Supplementary Information.

Text discussion newly added in the main manuscript:

Lines 13-23 of page 14 and lines 1-10 of page 15: “Furthermore, we synthesized $\text{Rb}_3\text{InCl}_6: \text{Yb}^{3+}/\text{Er}^{3+}$ NCs with different Yb^{3+} and Er^{3+} concentrations to investigate the effect of particle size on the upconverted UV emission of Er^{3+} . Structural and compositional analyses through XRD, TEM, EDS, ICP-AES, XPS, and Raman spectra confirmed the pure phase and high crystallinity of the resulting $\text{Rb}_3\text{InCl}_6: \text{Yb}^{3+}/\text{Er}^{3+}$ NCs with particle sizes in the range of 48.8–52.8 nm (Supplementary Figs. 27-29 and Supplementary Tables 11 and 12). Similar to their MC counterparts, $\text{Rb}_3\text{InCl}_6: \text{Yb}^{3+}/\text{Er}^{3+}$ NCs displayed unusually strong UV emission of Er^{3+} at 384 nm ($^4\text{G}_{11/2} \rightarrow ^4\text{I}_{15/2}$), with the UV-to-green ratio (I_{384}/I_{554}) of up to 0.702 for $\text{Rb}_3\text{InCl}_6: 50\%\text{Yb}^{3+}/1\%\text{Er}^{3+}$ NCs, upon 980-nm excitation at a power density of 60 W/cm^{-2} (Supplementary Fig. 30). The corresponding UCQYs for the UV emission (~ 384 nm) and overall emissions (360–720 nm) of Er^{3+} in $\text{Rb}_3\text{InCl}_6: 50\%\text{Yb}^{3+}/1\%\text{Er}^{3+}$ NCs were determined to be 0.08% and 0.37%, respectively, slightly lower than those of their MC counterparts (0.12% and 0.42%). Concentration-dependent UCL measurements showed the increased UCL intensity and I_{384}/I_{554} ratio with the increasing Yb^{3+} concentration from 10 mol% to 50 mol%, concurrent with the decreased UCL lifetimes (Supplementary Fig. 31 and Supplementary Table 13), confirming the absence of concentration quenching effect of Yb^{3+} in $\text{Rb}_3\text{InCl}_6: \text{Yb}^{3+}/\text{Er}^{3+}$ NCs. Further increasing the Er^{3+} concentration on the basis of $\text{Rb}_3\text{InCl}_6: 50\%\text{Yb}^{3+}/1\%\text{Er}^{3+}$ NCs shortened the interionic distance between Yb^{3+} and Er^{3+} , resulting in the enhanced back ET from Er^{3+} to Yb^{3+} and the decreased I_{384}/I_{554} ratio of Er^{3+} with reduced UCL lifetime of $^4\text{G}_{11/2}$ (Supplementary Figs. 32 and 33 and Supplementary Table 14). These results are generally consistent with those observed in $\text{Rb}_3\text{InCl}_6: \text{Yb}^{3+}/\text{Er}^{3+}$ MCs, unraveling that the unusually intense upconverted UV emission of Er^{3+} is the intrinsic properties of $\text{Rb}_3\text{InCl}_6: \text{Yb}^{3+}/\text{Er}^{3+}$ and

independent of the particle size.”

Experimental data newly added in the Supplementary Information:

Supplementary Table 11. Nominal and actual doping concentrations of Yb³⁺ in Rb₃InCl₆: x%Yb³⁺/1%Er³⁺ nanocrystals (NCs). The nominal doping concentration of Yb³⁺ was defined by the molar ratio of Yb to (Yb + In + Er) in the precursor solution, and the actual doping concentrations of Yb³⁺ were identified by ICP-AES.

Sample	In : Yb : Er	Nominal	Actual
		Yb ³⁺ / mol%	Yb ³⁺ / mol%
1	89 : 10 : 1	10	2.5
2	69 : 30 : 1	30	18.3
3	49 : 50 : 1	50	43.2

Supplementary Table 12. Nominal and actual doping concentrations of Er³⁺ in Rb₃InCl₆: 50%Yb³⁺/y%Er³⁺ NCs. The nominal doping concentration of Er³⁺ was defined by the molar ratio of Er to (Yb + In + Er) in the precursor solution, and the actual doping concentrations of Er³⁺ were identified by ICP-AES.

Sample	In : Yb : Er	Nominal	Actual
		Er ³⁺ / mol%	Er ³⁺ / mol%
1	49 : 50 : 1	1	0.8
2	46 : 50 : 4	4	3.6
3	43 : 50 : 7	7	5.3

Supplementary Table 13. Effective UCL lifetimes of Er³⁺ in Rb₃InCl₆: x%Yb³⁺/1%Er³⁺ NCs with different Yb³⁺ concentrations. The effective UCL lifetimes (τ_{eff}) of the NCs were measured by monitoring the Er³⁺ emissions at 384 nm (⁴G_{11/2}), 524 nm (²H_{11/2}), 554 nm (⁴S_{3/2}), and 659 nm (⁴F_{9/2}), respectively, upon 980-nm pulsed laser excitation, and calculated by

$$\tau_{eff} = \frac{1}{I_{max}} \int_0^{\infty} I(t) dt$$

where $I(t)$ denotes the UCL intensity as a function of time t , and I_{max} is the maximum UCL intensity.

Yb^{3+} / mol%	${}^4G_{11/2}$ (μs)	${}^2H_{11/2}$ (μs)	${}^4S_{3/2}$ (μs)	${}^4F_{9/2}$ (μs)
10	190	662	693	869
30	76	273	240	347
50	59	157	151	207

Supplementary Table 14. Effective UCL lifetimes of Er^{3+} in Rb_3InCl_6 : $50\%Yb^{3+}/y\%Er^{3+}$ NCs with different Er^{3+} concentrations. The effective UCL lifetimes (τ_{eff}) of the NCs were measured by monitoring the Er^{3+} emissions at 384 nm (${}^4G_{11/2}$), 524 nm (${}^2H_{11/2}$), 554 nm (${}^4S_{3/2}$), and 659 nm (${}^4F_{9/2}$), respectively, upon 980-nm pulsed laser excitation, and calculated by

$$\tau_{eff} = \frac{1}{I_{max}} \int_0^{\infty} I(t) dt$$

where $I(t)$ denotes the UCL intensity as a function of time t , and I_{max} is the maximum UCL intensity.

Er^{3+} / mol%	${}^4G_{11/2}$ (μs)	${}^2H_{11/2}$ (μs)	${}^4S_{3/2}$ (μs)	${}^4F_{9/2}$ (μs)
1	59	157	151	207
4	44	109	99	192
7	33	87	78	167

Supplementary Figure 27. (a) XRD patterns of Rb₃InCl₆: x%Yb³⁺/y%Er³⁺ NCs with different Yb³⁺ and Er³⁺ concentrations. The bottom lines represent the standard XRD pattern of monoclinic Rb₃InCl₆ (CCDC No. 2018909). (b) Energy dispersive X-ray spectra of Rb₃InCl₆: 50%Yb³⁺/1%Er³⁺ NCs, showing the elements of Rb, In, Cl, Yb, and Er in the NCs. Transmission electron microscopy (TEM) images and size distribution histograms of (c,d) Rb₃InCl₆: 10%Yb³⁺/1%Er³⁺, (e,f) Rb₃InCl₆: 30%Yb³⁺/1%Er³⁺, and (g,h) Rb₃InCl₆: 50%Yb³⁺/1%Er³⁺ NCs. The size distributions of the NCs were obtained by randomly calculating 200 particles in the TEM image. The inset in (g) shows the selected area electron diffraction (SAED) pattern of the NCs. All diffraction peaks of the NCs can be well indexed into monoclinic Rb₃InCl₆, indicating high crystallinity and phase purity of the as-synthesized Rb₃InCl₆: x%Yb³⁺/y%Er³⁺ NCs. The diffraction peaks of the NCs shifted towards lower angles with the increasing Yb³⁺ and Er³⁺ concentrations, as a result of lattice expansion induced by the substitution of In³⁺ (r = 0.81 Å, CN = 6) by larger Yb³⁺ (r = 0.86 Å, CN = 6) and Er³⁺ (r = 0.88 Å, CN = 6). TEM images showed that the NCs were roughly monodispersed with mean sizes in the range of 48.8–52.8 nm. The doping concentrations of Yb³⁺ and Er³⁺ had no significant influence on the size and morphology of the resulting NCs.

Supplementary Figure 28. XPS spectra of $\text{Rb}_3\text{InCl}_6: 10\%\text{Yb}^{3+}/1\%\text{Er}^{3+}$, $\text{Rb}_3\text{InCl}_6: 50\%\text{Yb}^{3+}/1\%\text{Er}^{3+}$, and $\text{Rb}_3\text{InCl}_6: 50\%\text{Yb}^{3+}/7\%\text{Er}^{3+}$ NCs. The spectra are shown over the energy regions typical for (a) Rb 3d, (b) In 3d, (c) Yb 4d, (d) Er 4d, and (e) Cl 2p peaks. The bimodal peaks at the binding energies of 109.7 and 111.1 eV, 445.0 and 452.3 eV, 184.7 and 192.2 eV, 169.1 eV and 170.4 eV, and 198.3 and 199.9 eV can be assigned to the $\text{Rb}^+ 3d_{5/2}$ and $3d_{3/2}$, $\text{In}^{3+} 3d_{5/2}$ and $3d_{3/2}$, $\text{Yb}^{3+} 4d_{5/2}$ and $4d_{3/2}$, $\text{Er}^{3+} 4d_{5/2}$ and $4d_{3/2}$, and $\text{Cl}^- 2p_{1/2}$ and $2p_{3/2}$, respectively. The $\text{Rb}^+ 3d$, $\text{Cl}^- 2p$, and $\text{In}^{3+} 3d$ peaks of the NCs shifted towards lower energies with the increasing Yb^{3+} or Er^{3+} concentration, ascribing to the strengthening of electron densities around Rb^+ , In^{3+} , and Cl^- . This implies that Yb^{3+} and Er^{3+} ions replace the octahedral In^{3+} site in the Rb_3InCl_6 lattice.

Supplementary Figure 29. Raman spectra of $\text{Rb}_3\text{InCl}_6: 10\%\text{Yb}^{3+}/1\%\text{Er}^{3+}$, $\text{Rb}_3\text{InCl}_6: 50\%\text{Yb}^{3+}/1\%\text{Er}^{3+}$, and $\text{Rb}_3\text{InCl}_6: 50\%\text{Yb}^{3+}/7\%\text{Er}^{3+}$ NCs. The vibrational peaks at 142, 169, and 282 cm^{-1} can be ascribed to the t_{2g} , e_g , and a_{1g} vibrational modes of the $[\text{InCl}_6]^{3-}$ octahedra, respectively. The Raman peaks of the NCs were broadened in comparison with those of the MCs, due to the influence of surface ligands and size effect of the NCs.

Supplementary Figure 30. (a) UCL spectra of $\text{Rb}_3\text{InCl}_6: x\%\text{Yb}^{3+}/1\%\text{Er}^{3+}$ NCs with different Yb^{3+} concentrations under 980-nm excitation at a power density of 60 W cm^{-2} . (b) UCL intensity ratio of ${}^4\text{G}_{11/2}/{}^4\text{S}_{3/2}$ of Er^{3+} and intensities of the upconverted emissions from ${}^4\text{G}_{11/2}$ (384 nm), ${}^2\text{H}_{9/2}$ (409 nm), ${}^2\text{H}_{11/2}$ (524 nm), ${}^4\text{S}_{3/2}$ (554 nm), and ${}^4\text{F}_{9/2}$ (659 nm) of Er^{3+} in $\text{Rb}_3\text{InCl}_6: x\%\text{Yb}^{3+}/1\%\text{Er}^{3+}$ NCs as a function of the Yb^{3+} concentration. The integrated UCL intensity of the NCs increased steadily with the increasing Yb^{3+} concentration from 10 mol% to 50 mol%, indicating the absence of concentration quenching effect of Yb^{3+} in $\text{Rb}_3\text{InCl}_6: x\%\text{Yb}^{3+}/1\%\text{Er}^{3+}$ NCs. Specifically, the intensity ratio of UV-to-green (I_{384}/I_{554}) of Er^{3+} was remarkably enhanced from 0.175 (10 mol% of Yb^{3+}) to 0.702 (50 mol% of Yb^{3+}). These results are generally consistent with those observed in $\text{Rb}_3\text{InCl}_6: x\%\text{Yb}^{3+}/1\%\text{Er}^{3+}$ MCs.

Supplementary Figure 31. UCL decay curves from (a) ${}^4\text{G}_{11/2}$ (384 nm), (b) ${}^2\text{H}_{11/2}$ (524 nm), (c) ${}^4\text{S}_{3/2}$ (554 nm), and (d) ${}^4\text{F}_{9/2}$ (659 nm) of Er^{3+} in $\text{Rb}_3\text{InCl}_6: x\%\text{Yb}^{3+}/1\%\text{Er}^{3+}$ NCs with different Yb^{3+} concentrations. The effective UCL lifetimes of ${}^4\text{G}_{11/2}$, ${}^2\text{H}_{11/2}$, ${}^4\text{S}_{3/2}$, and ${}^4\text{F}_{9/2}$ of Er^{3+} were determined to decrease from 190, 662, 693, and 869 μs to 59, 157, 151, and 207 μs , respectively, as the Yb^{3+} concentration increased from 10 to 50 mol% (see also Supplementary Table 13).

Supplementary Figure 32. (a) UCL spectra of $\text{Rb}_3\text{InCl}_6: 50\%\text{Yb}^{3+}/y\%\text{Er}^{3+}$ NCs with different Er^{3+} concentrations under 980-nm excitation at a power density of 60 W cm^{-2} . (b) UCL intensity ratio of ${}^4\text{G}_{11/2}/{}^4\text{S}_{3/2}$ of Er^{3+} (I_{384}/I_{554}) in $\text{Rb}_3\text{InCl}_6: 50\%\text{Yb}^{3+}/y\%\text{Er}^{3+}$ NCs as a function of the Er^{3+} concentration, showing decreased I_{384}/I_{554} with the increasing Er^{3+} concentration.

Supplementary Figure 33. UCL decay curves from (a) ${}^4\text{G}_{11/2}$ (384 nm), (b) ${}^2\text{H}_{11/2}$ (524 nm), (c) ${}^4\text{S}_{3/2}$ (554 nm), and (d) ${}^4\text{F}_{9/2}$ (659 nm) of Er^{3+} in $\text{Rb}_3\text{InCl}_6: 50\%\text{Yb}^{3+}/y\%\text{Er}^{3+}$ NCs with different Er^{3+} concentrations. The effective UCL lifetimes of ${}^4\text{G}_{11/2}$, ${}^2\text{H}_{11/2}$, ${}^4\text{S}_{3/2}$, and ${}^4\text{F}_{9/2}$ of Er^{3+} were determined to decrease from 59, 157, 151, and 207 μs to 33, 87, 78, and 167 μs , respectively, as the Er^{3+} concentration increased from 1 to 7 mol% (see also Supplementary Table 14).

Reply to the Comments of Reviewer #1

General Comment:

The authors have carefully addressed the referees' concerns and the level of this manuscript has improved a lot. Nonetheless, I fear there are still issues to be discussed.

Response:

We greatly appreciate the reviewer for his/her positive comments on our manuscript and the efforts to improve the quality of our manuscript. We have carefully looked into all the helpful suggestions by the reviewer and made all the requested changes that have been reflected either in the text or in the Supplementary Information. A point-by-point response is noted below.

Comment #1:

The authors used a Yokota-Tanimoto diffusion model to describe the energy transfer mechanistically. However, this does not take into account the discrete nature of the crystal structure with pre-defined cation-cation distances. A shell model-type approach seems more suited to me, see e.g. the works by Rabouw and Meijerink 2014, <https://doi.org/10.1038/ncomms4610>.

Response:

Thank you for pointing out this important issue. We have carefully read the recommended paper about the investigation of photonic effects on the FRET efficiency in $\text{LaPO}_4:\text{Ce}^{3+}/\text{Tb}^{3+}$ nanocrystals (NCs) with an ultrasmall size (~ 4 nm) (*Nat. Commun.* **2014**, 5, 3610). In their work, the “shell model” was used to simulate the dynamics of energy transfer (ET) from Ce^{3+} to Tb^{3+} , which agreed very well with the experimental data. They neglected the possibility of donor-to-donor energy migration (EM), justified by the low donor (Ce^{3+}) concentration of 1 mol%.

In our case, we focused on the donor-to-donor EM processes in Yb^{3+} singly-doped Rb_3InCl_6 and NaYF_4 microcrystals (MCs) with a donor concentration of up to 50 mol%. Although the Rb_3InCl_6 host has a peculiar 0D structure with isolated $[\text{InCl}_6]^{3-}$ octahedra surrounded by Rb^+ cations, the interionic $\text{Yb}-\text{Yb}$ and $\text{Yb}-\text{Er}$ distances were calculated to be approximately 0.7–0.8 nm when Yb^{3+} and Er^{3+} substitute the octahedral In^{3+} sites (Fig. 3a and Supplementary Fig. 22), significantly beneath the critical FRET distance of 10 nm (*J. Chem. Phys.* **1953**, 21, 836-850; *Chem. Soc. Rev.* **2017**, 46, 4150-4167). Therefore, we think that the Yokota-Tanimoto diffusion model is reliable for elucidating the EM process among Yb^{3+} in Yb^{3+} singly-doped Rb_3InCl_6 MCs. However, in the presence of acceptor ions (Er^{3+}), the “shell model” instead of the Yokota-Tanimoto model could be more appropriate for elucidating the dynamics of ET between Yb^{3+} and Er^{3+} , as proposed by the reviewer. To be more rigorous, we have added more text discussion about the validity of ET models used in our work. See also text discussion newly added in lines 7-10 of page 14 and Ref. 52 in the revised manuscript.

Text discussion newly added in the main manuscript:

Lines 7-10 of page 14: “It is worthy of mentioning that the “shell model” developed by F. Rabouw and A. Meijerink *et al.*⁵² instead of the Yokota-Tanimoto model could be more appropriate for elucidating the dynamics of ET from Yb³⁺ to Er³⁺, considering the complex interplay between the donors and acceptors.”

Reference newly added in the main manuscript:

52 Rabouw, F. T., den Hartog, S. A., Senden, T. & Meijerink, A. Photonic effects on the Förster resonance energy transfer efficiency. *Nat. Commun.* **5**, 3610 (2014).

Comment #2:

The authors tested the anion exchange in the perovskite NCs upon NIR irradiation and in the absence of the Rb₃InCl₆: Er, Yb NCs, which is a very important experiment. The incident power density was, however, 60 W cm⁻² (see Fig. 4). At such a high local power density, I would definitely expect local heating effects. How can the authors be sure that the anion exchange is not a consequence of local heating and thus, thermally stimulated reactivity? It may be good to have a control setup here by e.g. performing an experiment under varying temperatures and cross-correlate the results.

Response:

Many thanks for this valuable suggestion. We have conducted additional control experiments by heating (1) the mixed solution containing Rb₃InCl₆: Yb³⁺/Er³⁺ (RIC), dibromomethane (DBM), and CsPbCl₃ perovskite NCs (PeNCs) (RIC/DBM/CsPbCl₃) and (2) the mixed solution containing RIC, 2-iodopropane (IDP), and CsPbBr₃ PeNCs (RIC/IDP/CsPbBr₃) in the dark for 10 minutes at varying temperatures (25, 30, 40, 50, and 60 °C). The solutions were then subjected to PL measurements upon 365 nm excitation. As shown in Supplementary Fig. 43, the emission peak positions of the RIC/DBM/CsPbCl₃ and RIC/IDP/CsPbBr₃ solutions remained nearly unchanged with varying temperatures, despite of the different intensities. These observations demonstrate unambiguously that the local heating effect of laser (60 W cm⁻², 32.9 °C; Supplementary Fig. 42) cannot trigger the anion exchange of PeNCs in the mixed solutions. Instead, only under 980 nm excitation, Rb₃InCl₆: Yb³⁺/Er³⁺ NCs can generate efficient UV emission at 384 nm, which resulted in the cleavage of the C–X bonds of haloalkanes and subsequently drove the anion exchange of PeNCs (Figs. 4d and e). See also text discussion newly added in lines 2-5 from the bottom of page 18 in the revised manuscript and experimental data newly added in Supplementary Fig. 43 in the revised Supplementary Information.

Text discussion newly added in the main manuscript:

Lines 2-5 from the bottom of page 18: “Note that the laser-induced heating effect may also contribute to the enhanced anion exchange rate due to the increased motion of the anions with the temperature rise (Supplementary Fig. 42), but it is not the cause of anion exchange as it cannot lead to the cleavage of the C–X bonds of DBM and IDP (Supplementary Fig. 43).”

Experimental data newly added in the Supplementary Information:

Supplementary Figure 43. PL emission spectra ($\lambda_{\text{ex}} = 365 \text{ nm}$) of (a) the RIC/DBM/CsPbCl₃ and (b) RIC/IDP/CsPbBr₃ solutions after heating in the dark for 10 minutes at different temperatures. The emission peak positions of the RIC/DBM/CsPbCl₃ and RIC/IDP/CsPbBr₃ solutions remained nearly unchanged with varying temperatures, despite of the different intensities. These observations demonstrate unambiguously that the local heating effect of laser (60 W cm^{-2} , $32.9 \text{ }^\circ\text{C}$) cannot trigger the anion exchange of PeNCs in the mixed solutions.

Reply to the Comments of Reviewer #2

General Comment:

I have reviewed the response to the referees' comments and I am satisfied with it. They have satisfactorily addressed my concerns and provided a significant amount of new evidence to support their claims. I have no further suggestions, I believe this work is now suitable for Nature Communications standards and broad audience.

Response:

We greatly appreciate the reviewer for his/her positive comments on our manuscript and the efforts to improve the quality of our manuscript.

中国科学院福建物质结构研究所
Fujian Institute of Research on the Structure of Matter, Chinese Academy of Sciences

Reply to the Comments of Reviewer #4

General Comment:

The authors have responded to my concerns.

Response:

We greatly appreciate the reviewer for his/her comments on our manuscript and the efforts to improve the quality of our manuscript.

中国科学院福建物质结构研究所
Fujian Institute of Research on the Structure of Matter, Chinese Academy of Sciences

Reply to the Comments of Reviewer #1

General Comment:

The authors have clarified my remaining concerns and the manuscript appears publishable to me now. It is good to see how the (already initially high!) quality even improved in each step and I am sure that the study will be beneficial for the community.

Response:

We greatly appreciate the reviewer for his/her positive comments on our manuscript and the efforts to improve the quality of our manuscript.